# An integrated view of the structure and function of the human 4D nucleome

Job Dekker[1,2 ✉], Betul Akgol Oksuz[1], Yang Zhang[3], Ye Wang[4], Miriam K. Minsk[5], Shuzhen Kuang[6], Liyan Yang[1], Johan H. Gibcus[1], Nils Krietenstein[7], Oliver J. Rando[8], Jie Xu[9], Derek H. Janssens[10,11], Steven Henikoff[2,10], Alexander Kukalev[12], Willemin Andréa[12], Warren Winick-Ng[12], Rieke Kempfer[12], Ana Pombo[12], Miao Yu[13,14], Pradeep Kumar[15], Liguo Zhang[15], Andrew S. Belmont[15], Takayo Sasaki[16], Tom van Schaik[17,18], Laura Brueckner[17], Daan Peric-Hupkes[17,18], Bas van Steensel[17,18], Ping Wang[9], Haoxi Chai[19], Minji Kim[20], Yijun Ruan[19], Ran Zhang[21], Sofia A. Quinodoz[22,23], Prashant Bhat[22,24], Mitchell Guttman[22], Wenxin Zhao[25], Shu Chien[25], Yuan Liu[25], Sergey V. Venev[1], Dariusz Plewczynski[26,27], Ibai Irastorza Azcarate[12], Dominik Szabó[12], Christoph J. Thieme[12], Teresa Szczepińska[12,27,28], Mateusz Chiliński[26], Kaustav Sengupta[26], Mattia Conte[29], Andrea Esposito[29], Alex Abraham[29], Ruochi Zhang[3], Yuchuan Wang[3], Xingzhao Wen[30], Qiuyang Wu[25], Yang Yang[3], Jie Liu[20], Lorenzo Boninsegna[4], Asli Yildirim[4], Yuxiang Zhan[4], Andrea Maria Chiariello[29], Simona Bianco[29], Lindsay Lee[31], Ming Hu[31], Yun Li[32], R. Jordan Barnett[5], Ashley L. Cook[5], Daniel J. Emerson[5], Claire Marchal[33], Peiyao Zhao[16], Peter J. Park[34], Burak H. Alver[34], Andrew J. Schroeder[34], Rahi Navelkar[34], Clara Bakker[34], William Ronchetti[34], Shannon Ehmsen[34], Alexander D. Veit[34], Nils Gehlenborg[34], Ting Wang[35], Daofeng Li[35], Xiaotao Wang[9,36 ✉], Mario Nicodemi[29 ✉], Bing Ren[13 ✉], Sheng Zhong[25 ✉], Jennifer E. Phillips-Cremins[5 ✉], David M. Gilbert[16 ✉], Katherine S. Pollard[6 ✉], Frank Alber[4 ✉], Jian Ma[3 ✉], William S. Noble[21 ✉] & Feng Yue[9,37 ✉]

The dynamic three-dimensional (3D) organization of the human genome (the 4D nucleome) is linked to genome function. Here we describe efforts by the 4D Nucleome Project[1] to map and analyse the 4D nucleome in widely used H1 human embryonic stem cells and immortalized fibroblasts (HFFc6). We produced and integrated diverse genomic datasets of the 4D nucleome, each contributing unique observations, which enabled us to assemble extensive catalogues of more than 140,000 looping interactions per cell type, to generate detailed classifications and annotations of chromosomal domain types and their subnuclear positions, and to obtain single-cell 3D models of the nuclear environment of all genes including their long-range interactions with distal elements. Through extensive benchmarking, we describe the unique strengths of different genomic assays for studying the 4D nucleome, providing guidelines for future studies. Three-dimensional models of population-based and individual cell-to-cell variation in genome structure showed connections between chromosome folding, nuclear organization, chromatin looping, gene transcription and DNA replication. Finally, we demonstrate the use of computational methods to predict genome folding from DNA sequence, which will facilitate the discovery of potential effects of genetic variants, including variants associated with disease, on genome structure and function.

Since the publication of the first draft sequence of the human genome more than two decades ago, massive efforts have focused on identifying all genes and functional elements encoded in the genome. The resulting encyclopaedia of annotations has revealed a vast richness of coding and regulatory information, leading to an increased understanding of gene regulation in a multitude of cell types and conditions across human development and physiology[2,3]. Integration of functional annotations with genetic variation is starting to link genetically encoded functional elements and genes to complex traits and human diseases.

The spatial organization of genomes is linked to how genetic information encoded within them is activated, used and expressed in a cell-type- and condition-dependent manner. For example, enhancers functionally interact with specific distal genes, while ignoring others, through a process that can be controlled by genetic sequences such as insulator elements and tethering elements. This may involve biophysical mechanisms including phase separation, direct enhancer–promoter contacts, loop extrusion by cohesin, condensins and possibly other folding machines, as well as possible 'action at a distance' mechanisms involving diffusion and/or DNA-tracking factors[4–9].

The genome is organized at different scales[9–14]. At the local scale of the chromatin fibre, nucleosome positioning and histone modifications influence the structure and accessibility of DNA. At the scale of up to hundreds of kilobases, chromatin loops form in a dynamic manner, sometimes enriched near specific *cis*-elements and in many, but not all, cases such loops are generated through active loop extrusion by cohesin and condensin complexes[15]. The pattern of extrusion along chromosomes is modulated by *cis*-elements such as enhancers, promoters and insulators[16–18]. The process of loop extrusion contributes not only to loops between specific *cis*-elements including CTCF-bound sites, but it also underlies the formation of topologically associating domains (TADs)[19–21]. Loci within TADs interact frequently through cohesin-mediated extrusion[22]. TADs often have CTCF sites at their boundaries that block extrusion[20,23,24], thereby lowering the probability of interactions between loci on either side of the boundary, a phenomenon referred to as insulation[25]. Finally, chromosomal domains that can range in size from several kilobases to megabases cluster together in space to form subnuclear compartments[26–28]. Such associations can involve functionally distinct subnuclear structures and bodies such as nuclear speckles, nucleoli and the nuclear periphery. Many studies over the past several years have started to describe these phenomena, exploring the mechanisms of their formation and their potential roles in genome regulation.

To understand how genomes work to process genetic information into biologically meaningful responses, it is critical to quantitatively map and mechanistically understand the physical organization of the genome relative to itself and to nuclear landmarks and bodies, for example, identifying which distal enhancers contact target genes and how they work together to regulate gene expression.

The goal of the 4D Nucleome project is to gain detailed insights into the 3D folding of the human genome at the resolution of functional elements, in different cell states, over time and in single cells (that is, to map the 4D nucleome) so that links between chromosome folding and genomic function can be derived, mechanisms of folding can be explored and causal relationships between genome structure and function can be deduced[1,29,30].

During its first phase, starting in 2015, a major focus of the project has been the development and benchmarking of complementary experimental approaches for measuring the 4D nucleome, the development of computational and modelling approaches to analyse and interpret 4D nucleome data, and the generation of structural and quantitative models of the folded human genome (Fig. 1). We have collected data on chromatin state, chromosome folding and nuclear organization for two defined human cell types—H1 embryonic stem cells (hereafter, H1 cells) and immortalized foreskin fibroblasts (HFFc6 cells) (Supplementary Note 1). We have benchmarked and validated genomic assays for detecting and quantifying distinct features of chromosome folding, finding that each method contributes different information. This enabled us to put together a user guide with advice for future studies. Datasets were integrated to obtain linear annotations of 4D nucleome features along chromosomes, an extensive annotation of more than 140,000 looping interactions per cell type, and detailed 3D genome models, including models that reflect cell-to-cell variation in genome organization. Genome models and structural features were used to gain insights into how chromosome structure relates to gene expression and DNA replication patterns, and to build predictive models that can infer effects of sequence variants on chromosome folding, for example, in disease. All data described in this work are publicly available at the 4D Nucleome Data Coordination and Integration Center (https://data.4dnucleome.org/). Here we summarize and build on an extensive set of studies produced by the consortium (Supplementary Table 1).

## Benchmarking 3D genomic assays

A growing number of assays are being developed to probe the 3D folding of genomes. Here we present the generation and analysis of data obtained with sequencing-based assays, while ongoing and future analyses of the consortium place emphasis on imaging-based assays. Sequencing-based assays can be divided in two broad classes (Fig. 1). The first relies on chromatin interaction assays that comprehensively detect spatial proximities between loci, that is, the interaction frequencies between pairs or among sets of loci (for example, 3C-based assays and genome architecture mapping (GAM)[31]). The second set of genomic approaches report on physical distances of loci (that is, tyramide signal amplification and sequencing (TSA–seq)[32]) or contact frequencies (that is, DNA adenine methyltransferase identification (DamID)[33], split-pool recognition of interactions by tag extension (SPRITE)[34]) to specific nuclear structures, such as the lamina, nucleoli and nuclear speckles. We started by comparing and benchmarking chromatin interaction assays and integrating the data with independent data obtained with assays that report subnuclear locus positions.

Sequencing-based chromatin interaction detection methods differ in important ways, for example, detecting pairwise contact frequencies as in Hi-C versus sets of spatially proximal loci as in GAM or SPRITE. These methods can be unbiased in that they identify spatially proximal loci independent of specific factors (for example, chromosome conformation capture-based assays[35] such as Hi-C[26] and Micro-C[36,37], SPRITE[34] or GAM[31]), or are tailored to identify interactions between loci associated with specific proteins (for example, chromatin interaction analysis by paired-end tag sequencing (ChIA-PET)[38] and HiChIP/proximity ligation-assisted (PLAC)-seq[39,40]).

Here we compared the different methods for their ability to determine and quantify 4D nucleome features in H1 and HFFc6 cells. Data were compared using two concordant biological replicates obtained using unbiased genome-wide approaches (Hi-C, Micro-C, SPRITE and GAM) and targeted approaches (ChIA-PET for RNA polymerase II (RNA Pol II) and CTCF and PLAC-seq histone H3 trimethylated at Lys4 (H3K4me3)) (Fig. 2a,b and Supplementary Fig. 1a). We also refer to two comprehensive studies benchmarking GAM against Hi-C[41], and Micro-C and Hi-C protocol variants[42]. Those studies showed that GAM and Hi-C, and Micro-C and Hi-C quantitatively differ in compartment detection and loop detection. We also refer to a recent study in which polymer models of chromatin 3D architecture were used to show that Hi-C, GAM and SPRITE bulk data all capture overall reference 3D structures, whereas single-cell data can reflect the strong variability among single DNA molecules[43]. Finally, the consortium has extensively validated data obtained with Hi-C using high-throughput fluorescence in situ hybridization (FISH), finding extensive concordance[44].

One measure of data quality is the fraction of interactions that are intrachromosomal (*cis*) versus interchromosomal (*trans*). For all datasets (except for GAM, where such a metric cannot be directly obtained), the level of *cis* interactions was 70–90% (Supplementary Fig. 1b), indicating high signal-to-noise ratios as an absence of true signal is expected to produce only 2–5% *cis* interactions. Datasets clustered (based on compartment and insulation profiles) first by cell type and then by method. SPRITE and GAM, being the only multiway interaction detection methods, clustered as separate groups (Fig. 2c,e).

## Comparative analysis of contacts

To visualize relative interaction frequencies (contacts) between loci, contact maps were plotted as 2D heat maps at different length scales (Fig. 2a,b). Visual inspection of these contact maps at large genomic distances show that methods capture similar chromatin organization patterns, independently of whether contacts are mapped with Hi-C, Micro-C, GAM or SPRITE, or based on the occupancy of specific proteins (CTCF, RNA Pol II, H3K4Me3). Zooming in on specific genomic regions shows that mapping of chromatin contacts enriched for CTCF, RNA Pol II or the H3K4me3 histone mark captures subsets of contacts detected with untargeted methods (Hi-C, Micro-C). SPRITE detects sets of spatially proximal genomic loci, ranging from clusters of two loci up to thousands of loci. To visualize SPRITE data in Fig. 2, we converted

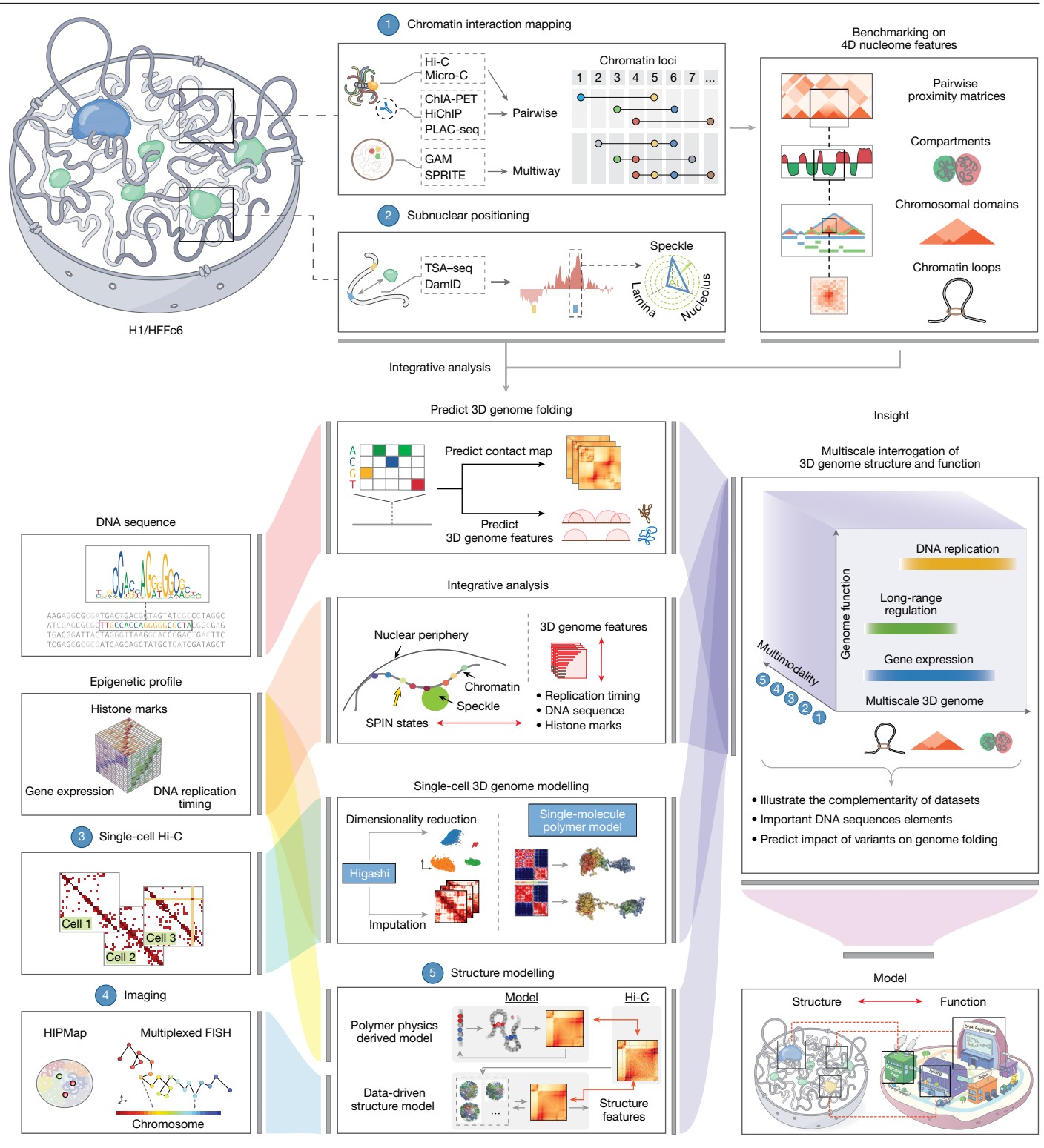

**Fig. 1 | Overview of the approach to generate and integrate genomic data on the 4D nucleome.** Top left, schematic of the two types of complementary genomic assay for mapping 3D genome folding and the relative distances of genomic loci to nuclear bodies in H1 and HFFc6 cells (top left). Top right, different chromatin interaction mapping methods were compared and benchmarked to assess their ability to identify and quantify 3D genome features at scales ranging from chromatin compartments (megabase) to focally enriched chromatin interactions (kilobase). Bottom left, additional multimodal datasets generated or used to facilitate integrative analyses (see below). HIPmap[44], high-throughput imaging position mapping. Bottom middle, multiple integrative modelling and analysis approaches were conducted to reveal the spatial features of chromatin loci by combining 3D genome features and various multimodal datasets. The connections between different input data and integrative analyses is illustrated through colour-coded flow paths. Bottom right, an illustrative cartoon summarizes the overarching aim of the project, which aims to provide insights into structure–function relationships by connecting variable 3D genome features (represented on the $x$ axis) derived from multimodal datasets ($y$ axis) with key cellular functions, such as transcription and replication ($z$ axis). Our models pave the way for identifying the sequence determinants of genome folding and predicting how different variants might influence this folding process. Hi-C contact map examples were drawn using ORCA[126].

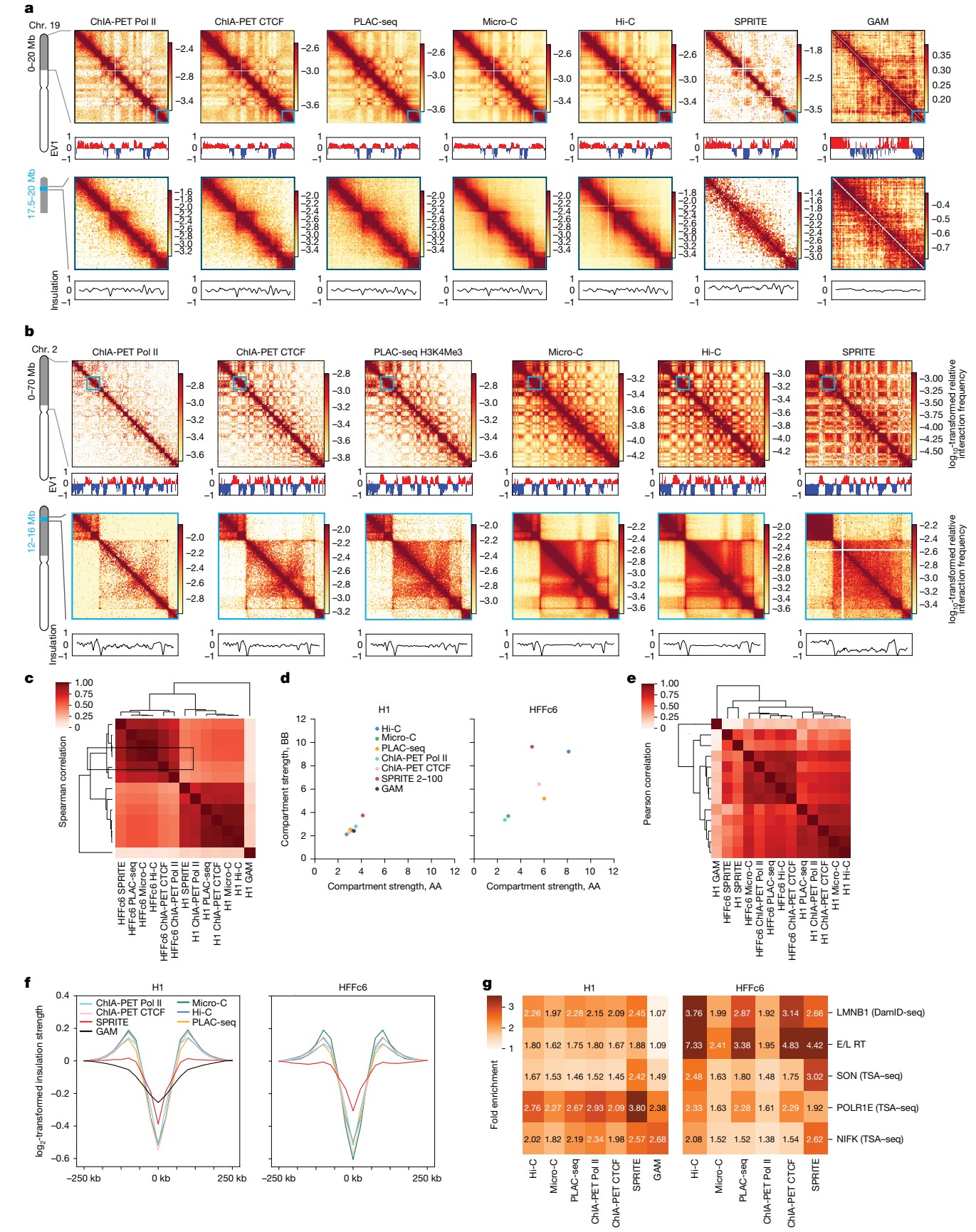

**Fig. 2 | See next page for caption.**

**Fig. 2 | Methods for chromatin interaction mapping differ in quantitative detection of compartmentalization. a**, Contact maps (100-kb bins, chromosome 19: 0–20 Mb) generated using the indicated methods were obtained with H1 cells. The plots below the heat maps show EV1 (compartments: red, A compartment; blue, B compartment). Bottom, magnified contact maps (corresponding to the blue squares in the heat maps in the top panel; 25-kb bins, chromosome 19: 17.5–20 Mb). The plots below the bottom heat maps show insulation profiles. **b**, Contact maps (100-kb bins, chromosome 2: 0–70 Mb) were generated using the indicated methods obtained with HFFc6 cells (top). The plots below the heat maps show EV1. Bottom, magnified contact maps (corresponding to the blue squares in the heat maps in top panel; 25-kb bins, chromosome 2: 12–16 Mb). The plots below the bottom heat maps show insulation profiles. **c**, Spearman correlation of compartment profiles determined by Eigenvector decomposition (Methods). **d**, Compartment strength quantified using eigenvectors from contact data obtained using the corresponding 3D methods. **e**, Pearson correlation of genome-wide insulation scores for all methods. **f**, Aggregated insulation scores at strong boundaries detected in multiple datasets (Methods, Supplementary Note) for all methods. **g**, Preferential interactions quantified in Hi-C, Micro-C, ChIA-PET, PLAC Seq, SPRITE and GAM, using DamID-seq for lamin B1, early and late replication timing (E/L RT) using RepliSeq and TSA–seq for SON to rank loci. The fold enrichment indicates the preference of loci with similar associations with speckles (SON), nucleoli (POLR1E/NIFK), lamina (lamin B), or that display early or late replication, to interact with each other, as detected by the indicated assays.

all clusters into weighted pairwise interactions exactly as described previously[34]. For most subsequent analyses described below, SPRITE data were split in subsets of interactions dependent on cluster size.

We computed interaction frequencies $P$ as a function of genomic distance ($s$) for HFFc6 cells (Supplementary Fig. 1d; similar results were obtained for H1 cells). For all methods, the expected inverse relationship between interaction frequency and genomic distance was observed. The shape of the $P(s)$ plots is comparable for all datasets, as indicated by the derivative of $P(s)$. $P(s)$ of all datasets revealed the presence of loops of around 100 kb as visible by the characteristic 'bump' in the $P(s)$ plot around 100–200 kb (ref. 45) (Supplementary Fig. 1d). This characteristic bump has been ascribed to the presence of cohesin-mediated loops. It is noteworthy that such global features of chromosome folding are also detected with ChIA-PET and PLAC-seq that were targeted to enrich for interactions involving sites occupied by CTCF, RNA Pol II or H3K4me3.

However, the methods differ in dynamic range, with Micro-C having the largest dynamic range and SPRITE (all cluster sizes combined) and GAM[41] the smallest. We further explored SPRITE data split by cluster size. For small clusters (2–100 fragments), $P(s)$ is steeper, and the dynamic range approaches that of Hi-C. For larger cluster sizes (100–1,000 and 1,000–10,000 fragments), $P(s)$ became increasingly flat, and *trans* interactions increased greatly (Supplementary Fig. 1e–g). Thus, SPRITE clusters of increasing sizes represent increasingly larger chromosome structures, ranging from local pairwise structures for small clusters to large subnuclear structures containing sections from multiple chromosomes for the largest clusters.

### Quantification of compartmentalization
Genomes are spatially segregated into active A and inactive B compartments that correlate with euchromatin and heterochromatin, respectively[26,46]. A and B compartments can be further split into subcompartments with distinct chromatin states and interaction profiles[23,47,48]. Compartmentalization is readily visible in all contact maps (Fig. 2a,b) as a plaid pattern of enriched interactions between domains of the same type, and depleted interactions of domains of different types. We used eigenvector decomposition and found that A and B compartmentalization is typically captured in the first eigenvectors (as shown previously[26]), which were highly correlated for most assays (Spearman coefficient > 0.73; Fig. 2c) and clustered according to cell type. GAM eigenvectors correlated with lower Spearman coefficients.

We calculated strength of compartmentalization using saddle plot analyses[42,49]. We found that, in H1 cells, compartmentalization is relatively weak for all methods, in comparison to the terminally differentiated HFFc6 fibroblast cells, as reported previously[42,50] (Fig. 2d). Notably, different compartment strengths were found with each method for HFFc6 cells: the strongest compartmentalization was found with SPRITE data obtained from clusters containing 2–100 fragments, and with Hi-C (Fig. 2d). Compartmentalization detected with GAM, Micro-C and the targeted assays was considerably weaker.

We also explored the contribution of larger SPRITE clusters to the detection of compartmentalization. We found that inclusion of larger SPRITE clusters results in decreased compartmentalization strength in both H1 and HFFc6 cells, and loss of the smaller compartment domains due to becoming absorbed into flanking domains (Extended Data Fig. 1a–c). Comparing the distributions of compartment domain sizes as detected by all methods, we find that GAM and ChIA-PET detect the smallest domains (for GAM: 80% of domains are <1 Mb) whereas, for data obtained with most other assays, only 50% of domains are smaller than 1 Mb (Extended Data Fig. 1e,f). However, compartmentalization is the strongest when calculated with data obtained with relatively small SPRITE clusters (2–100 fragments) (Fig. 2d and Extended Data Fig. 1a,c,d).

Cytologically, compartmentalization is related to the preferential co-localization of sets of loci at preferred subnuclear locations, or around specific subnuclear bodies[51,52]. For example, B compartment domains are often located near the nuclear and/or nucleolar periphery and are late replicating[53]. Such domains include lamin-associated domains (LADs), and these have been shown to colocalize by Hi-C[54]. By contrast, A compartments and gene-dense chromatin in general are located within the nuclear interior, are earlier replicating[53], and are enriched for active genes[55–57]. A subset of A compartment domains contains genomic regions with high gene expression that are preferentially positioned near nuclear speckles[32,34]. This enabled us to assess and validate the performance of each of the chromatin interaction assays to detect compartmentalization by using orthogonal datasets representing independent measures of subnuclear compartments. For H1 and HFFc6 cells, we generated genome-wide maps of LADs using lamin B1 DamID[58], maps of speckle-associated domains using SON TSA–seq[59] and maps of nucleolar-associated domains using POLR1E TSA–seq and NIFK TSA–seq[60], and determined replication timing using Repli-seq[61]. We calculated the extent to which preferential interactions between loci of the same type are detected with each interaction method (Fig. 2g). We find that compartmentalization calculated in this manner is again stronger in HFFc6 cells compared with in H1 cells. SPRITE (2–100 cluster size) and Hi-C generally detect the strongest homotypic associations. Micro-C, ChIA-PET and PLAC-seq also detected such preferentially homotypic interactions, but these preferences appeared weaker. GAM and SPRITE detected relatively strong associations between loci associated with the nucleolus. These assays may capture interactions with a larger contact radius, which may contribute to their ability to detect co-association of loci at and around larger subnuclear structures such as speckles and nucleoli. Notably, interactions between loci with similar replication timing is observed with all assays, consistent with earlier reports that early and late replication domains correlate strongly with A and B compartments detected by Hi-C[53]. In HFFc6 cells, this correlation between replication timing and interaction frequency is much higher than in H1 cells, consistent with the previous demonstration that consolidation of replication domains occurs during human embryonic stem (hES) cell differentiation coincident with a progressive increase in the alignment of replication timing with A/B Hi-C compartments[50,53,62].

Together, these observations show that all methods can be used to qualitatively detect compartmentalization and to identify compartment domains (Extended Data Fig. 1b). However, quantitative

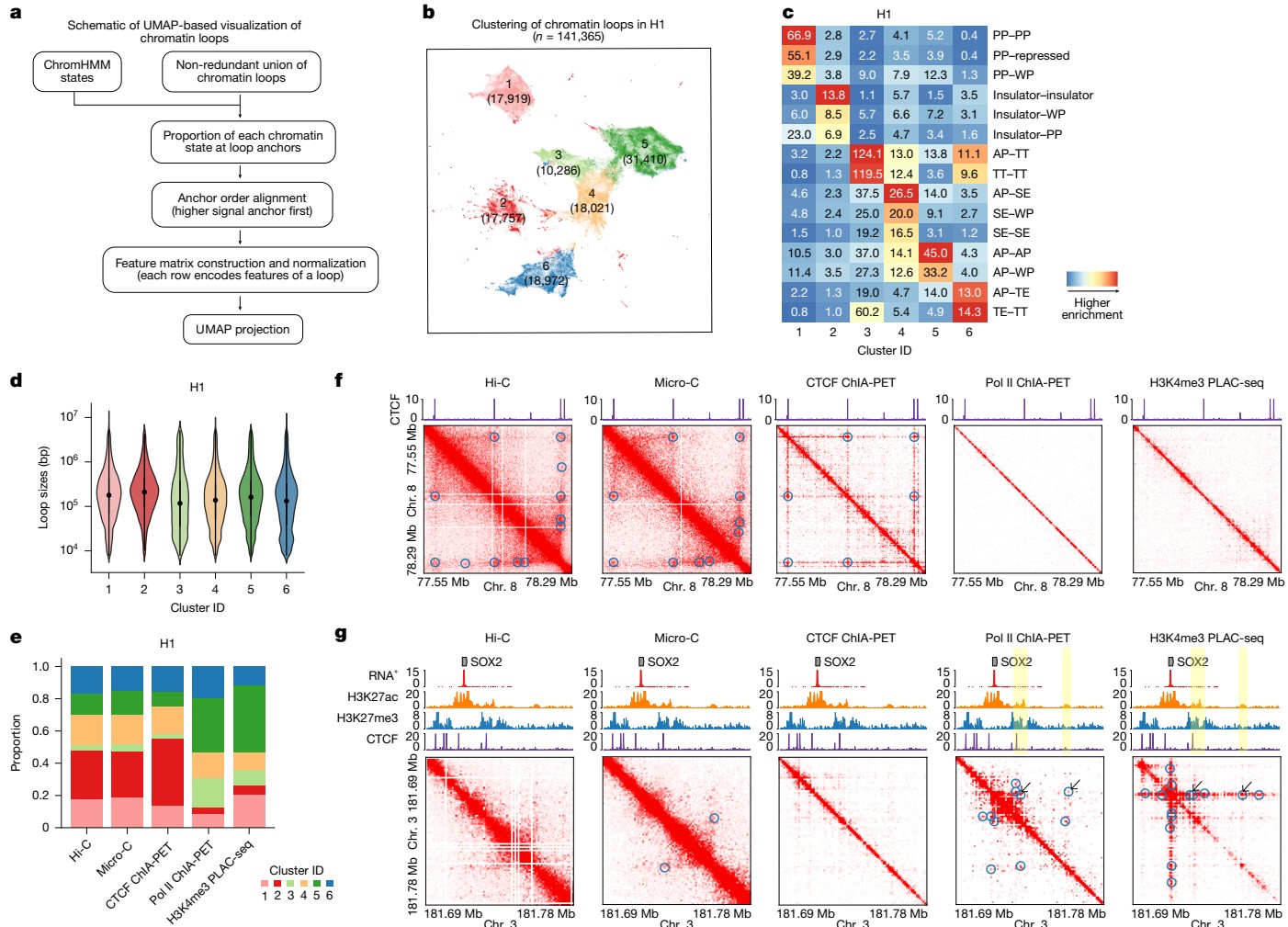

**Fig. 3 | Cross-platform loop comparisons in the H1 cell line. a,** Schematic of the construction of the feature matrix used for UMAP projection of chromatin loops. **b,** UMAP projection and clustering of 141,365 union loops in H1 cells, based on the ChromHMM state composition at their loop anchors. Cluster IDs and the number of loops per cluster are labelled on the UMAP. **c,** Fold enrichment of state pairs in each cluster. ChromHMM States: AP, active promoter; WP, weak promoter; TE, transcriptional elongation; TT, transcriptional transition; SE, strong enhancer; PP, poised promoter; repressed, polycomb repressed. **d,** The size distribution (genomic distance between loop anchors) of chromatin loops in each cluster. Within each violin, the black dot represents the median, and the

vertical line represents $1.5 \times$ the interquartile range (IQR). The number of loops (*n*) in each cluster is indicated on the UMAP in **b**. **e,** The cluster composition of loops detected by each platform. **f,** Example illustrating differences among platforms in detecting insulator-related loops. Contact maps are plotted at 5-kb resolution, with loops detected by each platform marked by blue circles. **g,** Example illustrating differences among platforms in detecting transcription-related loops. Contact maps are plotted at 1-kb resolution; loops detected by each platform are marked by blue circles. Loops linking the *SOX2* gene to distal enhancers are indicated by black arrows, and the interacting enhancers are highlighted with yellow shading.

differences between the methods are large, in terms of the size of compartment domains, the ability to detect smaller compartment domains and the ability to quantify the strength of compartmentalization.

### Detection of TAD boundaries

TAD boundaries reduce the probability of interactions between *cis*-regulatory elements and genes located on either side of the boundary, and there is therefore a great interest in identifying their genomic locations and characteristics. To measure such boundaries, we performed insulation analysis[25]. Insulation score profiles were visually very similar for data obtained with the different methods, although for SPRITE and GAM data, the profile has a reduced dynamic range (examples in Fig. 2a,b,f). This result was confirmed by calculating and clustering genome-wide Pearson correlation values of insulation scores: insulation profiles for data obtained with all methods were highly correlated (for all assays but SPRITE and GAM: $r > 0.76$ for H1 cells and $r > 0.8$ for HFFc6 cells); SPRITE and GAM-derived insulation scores had lower correlation values with other datasets (SPRITE: $r = 0.28–0.75$;

GAM: $r = 0.19–0.36$; Fig. 2e). Insulation profiles are generally clustered by cell type, except for SPRITE and GAM data. Finally, insulation scores aggregated at a set of boundaries identified in multiple datasets (Supplementary Note) showed that all methods detected boundary strength in comparable ways, except for SPRITE and GAM, for which the boundary strength appeared weaker (Fig. 2f). In summary, local domain boundary formation is a robustly detected feature of genome folding that is captured by a variety of chromatin interaction assays.

### Detection of chromatin loops

We next evaluated the ability of different chromatin interaction methods to detect chromatin loops, defined as focally enriched long-range interactions between specific pairs of loci (Fig. 3a). For this analysis, we used a hybrid approach, combining our previously developed platform-agnostic tool Peakachu[63] with platform-specific methods, which enabled us to effectively identify loops for different assays (Methods). We currently do not have tools to detect significant looping interactions with sufficient resolution from GAM data, and the

sequencing depth for SPRITE is not sufficient. However, aggregate peak analysis (APA) revealed that chromatin loops detected with other methods exhibited enriched SPRITE and GAM signals, albeit weaker compared with data obtained from other methods. Notably, smaller SPRITE clusters with 2–10 fragments displayed greater enrichment for such loop signals than larger clusters (Extended Data Fig. 2).

Combining all chromatin loops detected, we defined a union set of loops and loop anchors for each cell type (H1 cells: 141,365 loops and 69,731 anchors; HFFc6 cells: 146,140 loops and 75,305 anchors) (Methods and Extended Data Fig. 3a). This number far exceeds the number of loops detected with any single assay, indicating that each method contributes additional loops. We first focused on loop anchor comparisons and characterized the chromatin features of loop anchors identified by different assays (Extended Data Fig. 3b). Chromatin states were defined by ChromHMM[64] (Methods and Supplementary Fig. 2). In both cell lines, we observed that anchors detected by Hi-C, Micro-C and CTCF ChIA-PET were primarily enriched for insulators. Notably, while anchors detected by both Pol II ChIA-PET and H3K4me3 PLAC-seq were characterized by various active chromatin states, anchors unique to Pol II ChIA-PET showed greater enrichment for strong enhancers and transcriptional transition states, whereas anchors unique to H3K4me3 PLAC-seq were more enriched for poised promoter states. These patterns underscore the complementary nature of different assays in capturing regulatory elements across distinct chromatin states.

On the basis of the composition of chromatin states at loop anchors, we further projected the union set of chromatin loops onto a 2D space using uniform manifold approximation and projection (UMAP) (Fig. 3a and Methods). In both cell lines, six loop clusters were identified (Fig. 3b for H1 cells; Extended Data Fig. 3c for HFFc6). We observed distinct chromatin state compositions across clusters (Fig. 3c, Extended Data Fig. 3d and Supplementary Fig. 3): (1) the first cluster predominantly featured loops associated with poised promoters; (2) the second cluster primarily consisted of loops between insulators; and (3) the remaining four clusters exhibited varying degrees of enrichment for transcription-related chromatin states, including active promoters, weak promoters, strong enhancers, transcriptional elongation and transcriptional transition states. Notably, the enriched chromatin states of some clusters differed markedly between H1 and HFFc6 cells, probably reflecting extensive epigenetic reprogramming and dynamic loop remodelling during the developmental processes that gives rise to these distinct cell types. Moreover, loops in the second (insulator related) cluster tended to span longer genomic distances, whereas transcription-related loops in clusters 3–6 were generally shorter-range—consistent with previous findings that short-range chromatin loops are more closely associated with gene regulation[65–67] (Fig. 3d and Extended Data Fig. 3e).

We further characterized loops using additional transcription factor (TF) and chromatin regulator binding data and found that different loop clusters are enriched for distinct sets of regulatory proteins (Methods and Extended Data Fig. 4a,b). Notably, polycomb-group (PcG) proteins, such as EZH2 and RNF2, are specifically enriched at loop anchors in the first cluster. This is reminiscent of recent studies suggesting that a subset of chromatin loops is mediated by polycomb repressive complex 2 and may contribute to chromatin compaction and gene repression[68,69]. Notably, these PcG proteins consistently co-localized with KDM4A. Although originally characterized as a demethylase targeting H3K36me3 and H3K9me3, recent studies have shown that knockdown or overexpression of *KDM4A* can also significantly alter H3K27me3 levels[70,71]. This suggests that KDM4A may have a role in establishing or modulating a polycomb-type repressive chromatin environment by coordinating with PcG proteins at these specific loops. As expected, insulator-related loops in the second cluster showed the highest enrichment for CTCF and cohesin binding at their anchors but were relatively depleted of most other TFs. By contrast, loops associated with active promoters—primarily in clusters 3

and 5—were characterized by strong binding of RNA polymerase II (POLR2A), chromatin remodelling proteins (for example, CHD1) and transcription initiation factors (for example, TAF1), consistent with their roles in gene activation. Enhancer–promoter loops in cluster 4 and transcriptional-elongation-related loops in cluster 6 showed lower POLR2A enrichment but retained strong cohesin binding, indicating a role for cohesin in mediating these regulatory interactions.

For transcription-related loops in clusters 3–6, we further classified loop anchors into CTCF-bound and CTCF-unbound categories. While these two groups shared broadly similar chromatin states and TF binding profiles, certain TFs exhibited differential enrichment (Extended Data Fig. 4c,d). For example, in cluster 4 of H1 cells—which is predominantly composed of enhancer–promoter loops—TFs such as BCL11A, CHD1, CHD7, POLR2A and POU5F1 were preferentially enriched at CTCF-unbound loop anchors. This suggests that these factors may contribute to the formation or stabilization of enhancer–promoter interactions independently of CTCF in H1 cells.

We next visualized chromatin loops detected by each chromatin interaction assay on the 2D UMAP projection of the union set of loops (Extended Data Fig. 3f). In parallel, we quantified the proportion of loops from each cluster among those detected by each platform (Fig. 3e and Extended Data Fig. 3g). Pull-down-based methods targeting different factors detect distinct subsets of chromatin loops: (1) CTCF ChIA-PET predominantly captures insulator-related loops (cluster 2); (2) both Pol II ChIA-PET and H3K4me3 PLAC-seq are enriched for transcription-related loops, particularly those involving active promoters (clusters 3 and 5); (3) H3K4me3 PLAC-seq shows stronger enrichment for loops involving poised promoters (cluster 1) compared with Pol II ChIA-PET. Hi-C and Micro-C offer a relatively unbiased view of chromatin interactions and detect both insulator- and transcription-related loops (Fig. 3e,f). However, these methods appear less sensitive to certain transcription-related loops compared with Pol II ChIA-PET and H3K4me3 PLAC-seq, as they fail to detect key enhancer–promoter interactions, including those associated with genes essential for maintaining the embryonic stem cell state in H1 cells (Fig. 3g and Supplementary Fig. 4).

Together, these analyses reveal that different chromatin interaction assays preferentially detect distinct types of chromatin loops. Hi-C and Micro-C provide an unbiased view of genome architecture and are particularly effective at capturing CTCF/cohesin-mediated insulator loops. By contrast, targeted methods such as Pol II ChIA-PET and H3K4me3 PLAC-seq are enriched for transcription-related loops, with H3K4me3 PLAC-seq showing additional sensitivity to poised promoter loops. Despite these differences, most loop anchors—regardless of loop type—are associated with cohesin (Extended Data Fig. 4a (rightmost column)), underscoring the general role of this loop-extrusion complex in chromatin looping.

## Annotation through integrative modelling

Previously, we have demonstrated that it is possible to derive linear genome-wide annotations of spatial nuclear compartments by integrating complementary 3D genome mapping data, such as subnuclear spatial localization data obtained with TSA–seq, DamID and Hi-C data into a unified probabilistic model SPIN[72]. The resulting annotations (SPIN states) reveal distinct patterns of spatial localization of loci relative to multiple types of nuclear bodies supported by microscopy-based measurements[72] and show strong connections between large-scale chromosome structure and function, including replication timing and gene expression[53]. Here we applied a further improved SPIN framework with the support of joint modelling on multiple cell types to identify primary SPIN states in H1 and HFFc6 cells relative to nuclear speckles, nucleolus and nuclear lamina. We integrated datasets of TSA–seq and DamID to map proximity to nuclear bodies, and Hi-C to map chromatin interactions (Fig. 4a, Extended Data Fig. 5a and Supplementary Note)

and identified nine SPIN states with distinct patterns of chromatin compartmentalization. These SPIN states are as follows: speckle, interior active 1, 2 and 3 (Interior_Act1, Interior_Act2 and Interior_Act3), interior repressive 1 and 2 (Interior_Repr1 and Interior_Repr2), near lamin 1 and 2 (Near_Lm1 and Near_Lm2) and lamina (Extended Data Fig. 5a). The functional annotations of these nine SPIN states were verified by comparisons with functional genomic data (below) and exhibit high correlation with SPIN states identified in K562 cells (Extended Data Fig. 5b). Overall, different SPIN states have distinct distributions and combinations of TSA–seq and DamID signals, reflecting distinct patterns of spatial compartmentalization (Extended Data Fig. 5c).

## SPIN states and histone modifications

To gain a better understanding of the transcriptional regulatory landscape of these SPIN states, we measured the enrichment of chromatin immunoprecipitation–sequencing (ChIP–seq) signals for a range of histone marks on each SPIN state as compared to the genome-wide average for each mark. For H1 cells, we used ChIP–seq data, and for HFFc6 cells we used ChIP–seq data imputed using Avocado[73]. We found that, as the SPIN state changes from the nuclear periphery to interior (for example, lamina state to speckle state), the enrichment of active histone marks (for example, H3K27ac, H3K4me1, H3K4me3 and H3K9ac) increases, along with gradual depletion of the repressive heterochromatin mark H3K9me3 (Fig. 4a and Extended Data Fig. 6a), consistent with what we reported earlier[72] with additional cross cell-type comparisons. Notably, active histone marks such as H3K4me1, H3K4me2, H3K4me3 and H3K27ac are most prevalent in speckle states ($P < 2.2 \times 10^{-16}$), followed by in the Interior_Act1/2/3 states. Similarly, CTCF is most enriched in speckle-associated and interior active states, consistent with recent studies[74], and shows an overall decrease from interior to peripheral SPIN states (see the HFFc6 analysis in Fig. 4). Although the correlations with histone marks are generally consistent between H1 and HFFc6 cells, certain histone marks exhibit more variable association with SPIN states. In particular, H3K27me3 is more enriched in the Interior_Repr1 states in HFFc6 cells but has a stronger association with the Interior_Act3 and speckle states in H1 cells, indicating the cell-type-dependent and variable nature of the spatial localization of loci associated with specific histone marks (Fig. 4a). The variable distribution of H3K27me3 is also observed in CUT&RUN data (Extended Data Fig. 6a). Previous work also reported a high cell-type-specific distribution of H3K27me3 across human cell lines[75]. Moreover, we found that more interior SPIN states are more ubiquitous across cell types than more peripheral SPIN states, consistent with previous observations of conservation of nuclear speckle associated domains[59]. These results reveal that SPIN states have a generic correlation with active histone marks. However, at least in certain cell types, SPIN states have a cell-type-specific distribution of repressive histone marks such as H3K27me3.

## SPIN states and chromatin-associated RNA

We further compared SPIN states with different types of chromatin-associated RNAs (caRNAs) detected with iMARGI, a genome-wide profiling technology designed to systematically identify and map the physical interactions between RNA molecules and chromatin in their native nuclear environment[76–78]. RNA facilitates spatial compartmentalization in the nucleus[79]. caRNAs can promote or suppress chromatin looping depending on their associated genomic sequences. Loop-anchor associated caRNAs often promote looping, including enhancer–promoter loops[80], whereas between-loop-anchor-associated caRNAs often suppress looping[78]. We examined whether any SPIN state is enriched with caRNAs containing specific types of sequence features, especially repetitive elements. We selected the caRNAs if their RNA ends in iMARGI mapped to repetitive elements and further stratified them into different groups according to SPIN states on their DNA ends (Methods, Supplementary Note). To avoid bias due to nascent RNA transcripts interacting with genomic regions where they are transcribed, we included only

interchromosomal iMARGI pairs, where the transcription and interacting genomic regions are on different chromosomes. We found that the genomic target sequences of different types of repeat sequences associated with different types of caRNAs are enriched for distinct SPIN states (Fig. 4a (right)). The caRNAs that contained Alu, srpRNA, SVA and snRNA repeat elements are mostly enriched in interior SPIN states (for example, speckle and Interior_Act1/2/3), while the caRNAs connected to L1, ERVL, ERV1 and LTR repeats are enriched in SPIN states closer to nuclear lamina (lamina, Near_Lm2). Thus, sequence features of caRNAs are correlated with the 3D compartmentalization of their target genomic regions.

Together, these results demonstrate that the SPIN framework effectively integrates various nuclear organization mapping data to produce genome-wide large-scale compartmentalization patterns relative to multiple nuclear bodies. These SPIN states stratify orthogonal functional genomic data, including histone modification, replication timing and RNA association.

## A 3D view of the genome

Using Hi-C, lamin B1 DamID and SPRITE data, we used the integrative genome modelling platform (IGM)[81] to construct a population of 1,000 single cell 3D genome structures at 200-kb resolution for H1 and HFFc6 cells, ensuring collective consistency with all input datasets. These structures illuminate the folding of chromosomes within the nuclear topography in single cells (Fig. 4c,b and Extended Data Fig. 6b,c). Independent validation using multiplexed FISH imaging[82] and TSA–seq data confirmed the consistency of the predicted spatial organization (Supplementary Fig. 5; further validations were described previously[81,83]). We have shown that such models are instructive when analysing cell-to-cell variation in 3D folding variation to gene expression[84] and compartmentalization[85].

To characterize the nuclear microenvironment of loci, we define 14 structural features that collectively specify their 3D positioning relative to the predicted locations of nuclear bodies, compartments and properties of the chromatin fibre, as described previously[83] (Methods, Extended Data Fig. 6d and Supplementary Fig. 6b–e).

## SPIN states

We analysed the nuclear microenvironment of chromatin across SPIN states (Extended Data Fig. 6b–d and Supplementary Fig. 6a,b), revealing distinct enrichments of 3D structure features (Extended Data Fig. 6d and Supplementary Fig. 6b). Radial positions progressively increase from speckle to lamina states in HFFc6 cells, consistent with expectations (Extended Data Fig. 6e; see Supplementary Fig. 6c for H1 cells). Specific SPIN states show preferred associations with nuclear bodies, including nuclear speckles for the speckle states (see Extended Data Fig. 6d,f for HFFc6 cells, and Supplementary Fig. 6d for H1 cells) and enriched nucleolar associations for the Interior_Act2 and Near_Lm1 chromatin states (Extended Data Fig. 6d and Supplementary Fig. 6b).

## Genome structure differences between cell types

Genes with large expression differences between H1 and HFFc6 cells are often found in distinct nuclear microenvironments. For example, 74% of genes with a $\log_2$-fold expression reduction of >9 in HFFc6 cells (FDR < 0.05) show significantly increased distances to nuclear speckles compared with H1 cells (false discovery rate (FDR) < 0.05; Extended Data Fig. 6g). This spatial difference is further illustrated by contrasting radial positions of genes along the p-arm of chromosome 1 (Fig. 4b). For example, the TF POU3F1 (also known as OCT6) has a pivotal role in cell differentiation and maintenance of the nervous system[86,87]. Expressed in H1 cells, the *POU3F1* gene is predominantly located in the nuclear interior (average radial position, RAD) and shows relatively small speckle distances in a high fraction of cells (Fig. 4b–d and Extended Data Fig. 6h (top)), consistent with the speckle SPIN state. By contrast, in HFFc6

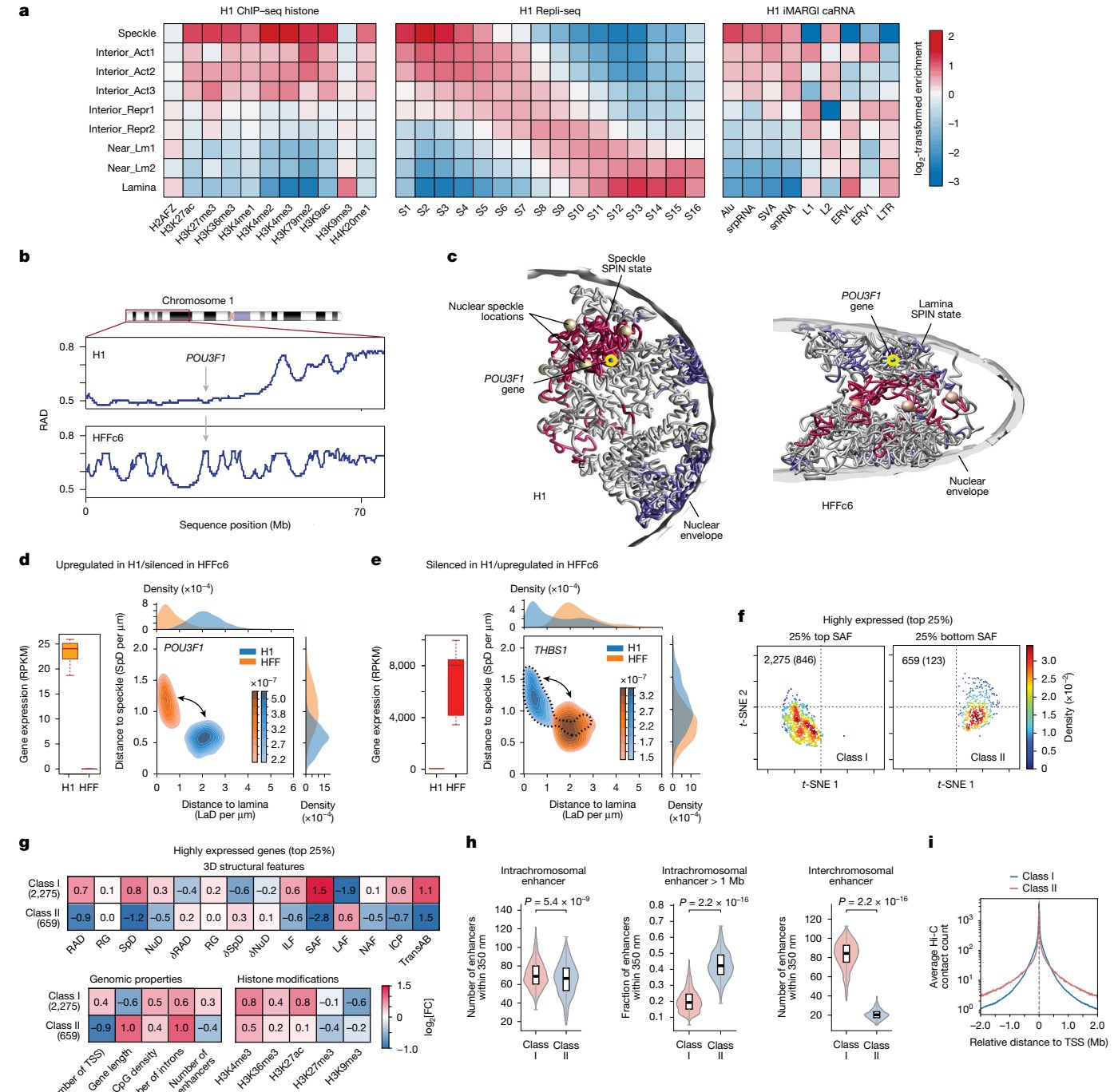

**Fig. 4 | 3D structure–function relationship and its cell-to-cell variability.**
**a**, SPIN states define spatial genome compartments. The heat maps show enrichment of histone marks, Repli-seq and caRNAs (columns) across SPIN states (rows) in H1 cells. The fold change shows the ratio of observed signals over the genome-wide expectation. **b**, The average radial positions for a chromosome 1 segment in H1 (top) and HFFc6 (bottom) cells. **c**, Representative single-cell 3D structures of chromosome 1 in H1 (left) and HFFc6 (right) cells. The yellow circles mark *POU3F1* loci; red and blue shading denotes chromatin in speckle- and lamina-associated SPIN states; spheres indicate predicted speckles. **d**, *POU3F1* expression (RNA-seq) in H1 and HFFc6 cells (left). Right, the joint distribution of the nearest speckle/lamina distances of *POU3F1* across 1,000 structures. The box plot shows the median (centre line), IQR (box) and the whiskers extend to 1.5 × IQR. *n* = 7 (H1) and 5 (HFF). **e**, The same as in **d** but for *THBS1*. **f**, *t*-Distributed stochastic neighbour embedding (*t*-SNE) of transcription start site (TSS)-containing regions for the top 25% expressed genes, based on 3D-structure features, separated by genome-wide top (left) and bottom (right) quartiles of SAF. **g**, The log2-transformed fold enrichment of 14 structural features for highly expressed class I and II genes (top): radial position (RAD, 1 − norm); chromatin decondensation (RG, ±500 kb); distance to nearest speckle/nucleolus (SpD/NuD); interior localization probability (ILF); speckle, lamina and nucleolus association frequencies (SAF, lamina association frequency (LAF) and nucleolus association frequency (NAF)); interchromosomal interaction probability (ICP), trans A/B ratio (TransAB), cell-to-cell variability (δRAD, δRG, δSpd and δNuD). Bottom, log2 enrichment of genomic features (within 200-kb bin) and histone marks (±10 kb from TSS). **h**, Spatial enhancer count within 350 nm from TSS for class I and II genes: intrachromosomal (left), ultra-long-range >1 Mb (middle) and interchromosomal (right). The box plots are as described in **d**. *n* = 2,275 (class I) and 659 (class II). *P* values were calculated using two-sided Mann–Whitney *U*-tests with no adjustment for multiple comparisons. **i**, Pile-up Hi-C contact frequencies (10-kb bins) centred on TSS of class I and II genes showing mean contact decay with sequence distance.

where *POU3F1* is silent, it resides closer to the nuclear periphery in a high fraction of cells (Fig. 4b–d, Extended Data Fig. 6h (top) and Supplementary Fig. 6e), shows increased association with B compartment chromatin (decreased trans A/B ratio (transAB)) and a higher degree of chromatin fibre compaction (lower RG (radius of gyration over a 1 Mb window)) (Extended Data Fig. 6h (top)), consistent with the Interior_Rep1 SPIN state. We found opposing trends in nuclear localization patterns for genes highly expressed in HFFc6 cells but silent in H1 cells, such as *THBS1* (thrombospondin-1, which promotes cell adhesion in connective tissue cells[88,89]) (Fig. 4e and Extended Data Fig. 6h (middle)). Notably, genes silent in H1 cells often show a bimodal distribution of their nuclear locations characterized by a smaller subpopulation of alleles situated in a nuclear microenvironment resembling those of active genes (that is, *THBS* and *CAV1*; dotted line in Fig. 4e and Supplementary Fig. 6f). This may either indicate increased structural heterogeneity between individual H1 cells or the presence of a subpopulation in a different epigenetic state. Genes highly expressed in both cell types typically show similar microenvironments, as shown by overlapping 2D distributions of their joint speckle and lamina distances in the models of both cell types (Extended Data Fig. 6i).

## Gene expression

Although gene expression generally correlates with the nuclear microenvironment, notable exceptions exist (Extended Data Fig. 6j,k). We analysed the nuclear microenvironment of the 25% most highly and lowly expressed genes in our HFFc6 genome structure models. We found that 90% of the most highly expressed genes, including 73% of all housekeeping genes, show high to medium association frequencies with nuclear speckles, consistent with previous observations[32,34,90]. Among these, 2,275 genes with the highest speckle associations (top 25% speckle association frequency (SAF) genome-wide) have predominantly interior radial positions (RAD), relatively low cell-to-cell variability in both radial location (δRAD) and speckle distances (δSpD) and high interchromosomal proximities (ICP) (Fig. 4f,g, Extended Data Fig. 6l and Supplementary Fig. 7a,b). We define these genes as belonging to the class I microenvironment (Fig. 4f (left), Methods and Supplementary Fig. 7a,b). Class I genes tend to be shorter and reside in gene-dense regions (Fig. 4g (bottom left) and Supplementary Fig. 7a,b). By contrast, 10% (659 genes) of the most highly expressed genes are found in a markedly different nuclear microenvironment, termed class II, which is typically associated with low-expressed or silenced genes (Fig. 4f (right) and 4g and Extended Data Fig. 6m). Compared with lower transcribed class II genes, they exhibit notably higher CpG density and H3K4me3 and H3K27ac levels at their promoter sites ($P < 10^{-5}$ for both H3K4me3 and H3K27ac) (Fig. 4g and Supplementary Fig. 7a,b). Class II genes are more uniformly distributed throughout the nucleus and, on average, occupy more peripheral radial positions with greater cell-to-cell variability than class I genes (high SpD, δSpD, δRAD, low SAF) (Fig. 4g, Extended Data Fig. 6l and Supplementary Fig. 7a,b). They feature minimal speckle associations (bottom 25% SAF genome-wide), are predominantly associated with B compartment chromatin (low transAB) and are more spatially confined within their chromosome territory, as indicated by lower ICP (Fig. 4g and Supplementary Fig. 7a,b). These genes tend to be longer and located in regions of low gene density (Fig. 4g). Nineteen percent of these genes are identified as housekeeping genes, highlighting that housekeeping genes can be found in at least two contrasting nuclear microenvironments (Supplementary Fig. 7c,d).

We observed marked differences in the regulatory architectures of class I and II genes. To quantify promoter–enhancer proximity, we computed the spatial enhancer count—the number of active enhancers located within 350 nm of a promoter in the folded 3D genome structures[91,92]. Although both classes show similar total counts from intra-chromosomal enhancers (Fig. 4h (left)), class I promoters are enriched for enhancers at close sequence distances (<100 kb), whereas class II promoters (Fig. 4g,h) exhibit a significantly greater enhancer count from ultra-long-range proximities (>1 Mb sequence distance) ($P < 10^{-5}$) (Fig. 4h (middle) and Supplementary Fig. 7a,b). This difference may help to explain the lower number of high-frequency enhancer–promoter loops observed for class II housekeeping genes using population data (see the section below), as ultra-long-range interactions are probably more variable between cells. Only class I promoters show substantial spatial enhancer counts from ICPs, probably due to their frequent protrusion beyond their chromosome territory towards nuclear speckles, sites of heightened ICP (Fig. 4h (right) and Extended Data Fig. 6o).

These distinctions are supported by Hi-C data: pile-up analyses on aligned promoters reveal that class I genes show enriched short-range contacts, whereas class II genes show reduced local interactions but stronger ultra-long-range contact frequencies (Fig. 4i). For example, the contact frequency map of class I gene *LMNA* (encoding lamin A/C) shows enriched short-range contacts, whereas the Hi-C contact patterns surrounding the class II gene *FBN2* (encoding fibrillin 2) displays reduced local interactions and enhanced ultra-long-range contacts (Extended Data Fig. 6n).

## Single-cell 3D genome analysis

We further investigated variability in single-cell 3D genome structure using different integrative modelling approaches. Previous comparisons between model-predicted single-molecule 3D structures and multiplex microscopy data showed that both loop-extrusion and polymer phase separation (strings and binders switch model[93–95]) recapitulate not only the average contact probabilities and patterns, but also the entire ensemble of microscopically observed single-molecule conformations in single cells[94,96,97]. Imputation of single-cell Hi-C contact maps using Higashi has also enhanced the single-cell 3D genome folding analysis with graph representation learning[98], which complements polymer modelling.

The consortium generated single-cell Hi-C data for WTC-11 pluripotent stem cells. These cells share many features with H1 cells. We used these data to analyse variability in genome folding. We first verified that both SBS polymer models and Higashi-imputed single-cell Hi-C (scHi-C) contact maps of the *DPPA* locus (chromosome 3: 108.3–110.3 Mb) in WTC-11 pluripotent stem cells at 10-kb resolution, result in ensembles of structures that are consistent with bulk Hi-C at the population level (Fig. 5a). These models improve correlations between merged scHi-C and bulk Hi-C data (Fig. 5a and Supplementary Note).

Next, we calculated single-cell TAD-like domain boundaries using single-cell insulation scores[98] and observed that both SBS polymer models and Higashi-imputed scHi-C data reveal the variability of TAD-like domain boundaries while remaining consistent with the insulation scores calculated from bulk Hi-C (Fig. 5a). To view the direct correspondence of Higashi-imputed scHi-C contact maps and SBS polymer models, we found nine mutual nearest neighbours by quantifying the pairwise similarities between scHi-C and polymer models (Fig. 5b). This analysis supports cell-to-cell variability of TAD-like structures using an integrative approach from different analysis methods, consistent with observations based on multiplexed imaging methods[99].

We investigated the variability of chromatin loops at the single-cell level, which has not been analysed extensively owing to the sparsity of scHi-C data and the limitation of spatial resolution in multiplexed imaging methods. We used polymer models[94,96,97], Higashi-imputed scHi-C contact maps[98] and the recent SnapHiC contact maps[100] derived from WTC-11 scHi-C datasets, to identify chromatin loops. This integrative analysis of chromatin loops, A/B compartments and TAD-like domains identified from single cells with different analysis approaches revealed that genomic loci within the same compartment or TAD-like domain are more likely to form stronger chromatin loops in the same cell (Fig. 5c). For example, in a representative chromatin loop near gene *RABGAP1L*,

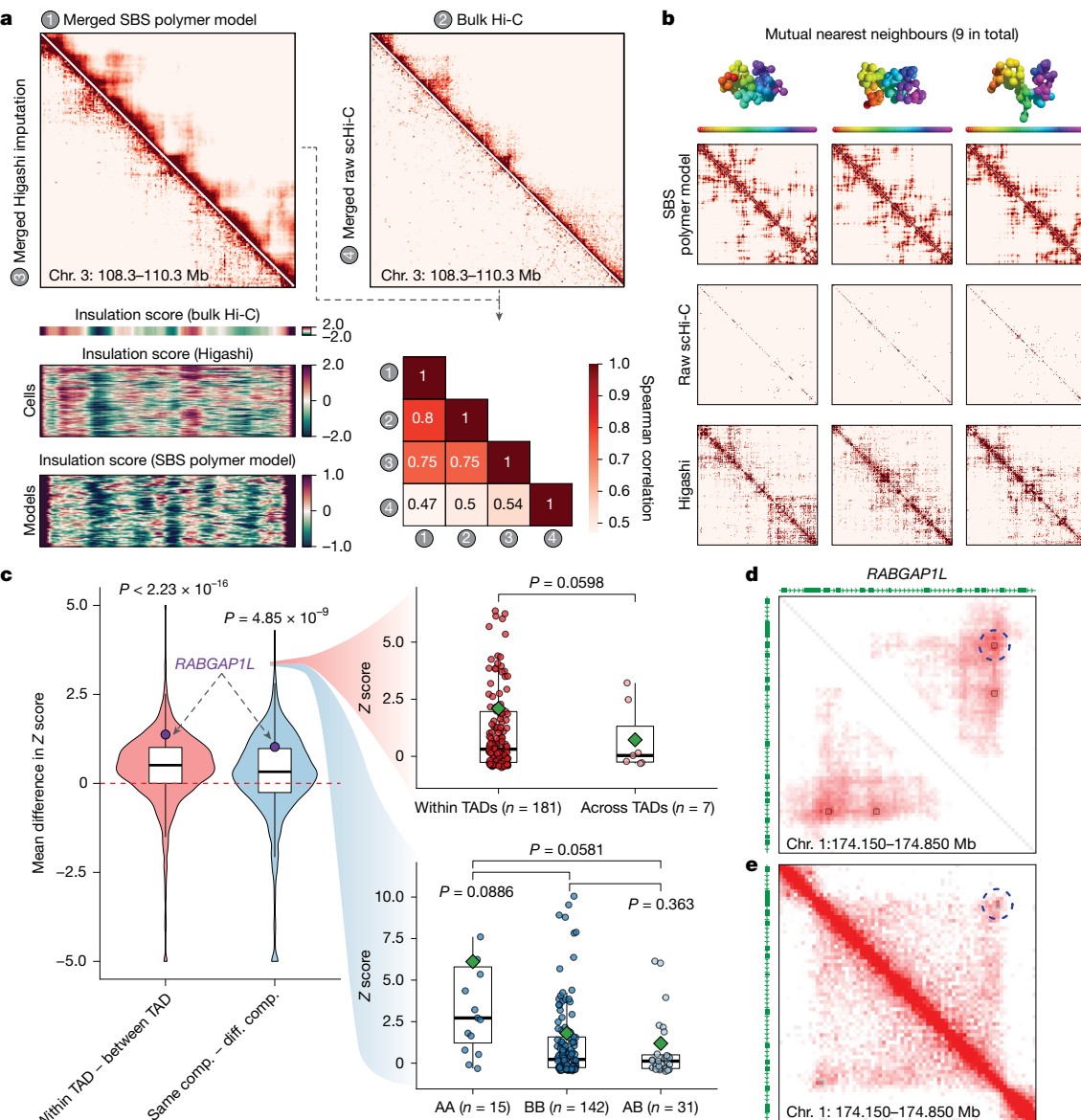

**Fig. 5 | Cell-to-cell variability in 3D genome features. a**, Merged scHi-C contact maps imputed by Higashi or predicted by the SBS model, as compared to bulk Hi-C and raw contact maps from scHi-C without imputation (top). Bottom left, insulation scores from bulk Hi-C, calculated after Higashi imputation, and SBS modelling. Bottom right, Spearman correlation coefficients between these contact maps. **b**, 3D models, raw scHi-C contact map, imputed maps from three similar cells between Higashi imputation and SBS model maps. The Higashi–SBS model contact map pairs have distance-stratified similarity scores of 0.69, 0.64 and 0.77 (left to right). **c**, The average normalized intensity of chromatin loops across 188 cells was calculated and compared by dividing loops on the basis of their position within TADs and A/B compartments (comp.). Left, the difference between loops in the same TAD ($n = 181$ cells) and loops spanning multiple TADs ($n = 7$ cells). Right, the difference between loops in the same

A/B compartment ($n = 157$ cells) and loops spanning different compartments ($n = 31$ cells). A chromatin loop near *RABGAP1L* is highlighted in the right plot. The original distribution of the normalized intensity of this loop in each cell is shown in the right plots. Loops are stratified into groups depending on whether they locate within one TAD ($n = 181$ cells) or span TADs ($n = 7$ cells), or the A/B compartment state of loop anchors in each single cell ($n = 15$ (AA), $n = 142$ (BB) and $n = 31$ (AB) cells). The box plots show the median (centre line), IQR (box) and the whiskers extend to $1.5 \times$ IQR. $P$ values were calculated using two-sample two-sided $t$-tests. **d**, Aggregated contact map from single-cell Hi-C data at *RABGAP1L* (for cells with $z$ score $> 1.96$). The circle with a dashed line indicates the 450-kb loop identified by SnapHi-C. **e**, Knight-Ruiz-normalized bulk Hi-C map from WTC-11 at *RABGAP1L*.

the normalized loop intensity is much higher for single cells where this loop is located within the same TAD-like domain or the two loop anchors located in the same compartment, specifically in A compartments (Fig. 5c (bottom right box plot) and 5e).

Together, our findings illustrate the cell-to-cell variation in chromatin folding in individual cells. Our analysis suggests that the formation of loops, TAD-like domains and compartments, although at different scales, are not merely correlated properties of chromatin folding observed in the bulk (averaged) contact maps. Instead, these

structures can be observed at the single-cell and single-molecule level. The variability of chromatin folding in individual cells, at the level of loops, TADs and compartments, probably contributes to the dynamic regulation of gene expression and other nuclear processes, providing further insight into the complex nature and dynamics of genome organization and function. Furthermore, cell-to-cell variability and dynamics can explain how, in a cell population, genes can be observed to interact with numerous distal regulatory elements, as examined below.

## Three-dimensional genome and genome function

### Enhancer interactions and transcription

Using the union set of chromatin loops defined above, we examined the relationship between the number of distal enhancers linked to promoters and the transcription levels of corresponding protein-coding genes. Among 19,693 protein-coding genes, 14,321 in H1 cells and 12,804 in HFFc6 cells interacted with at least one distal enhancer. The median distance between interacting enhancers and promoters was 173 kb, notably shorter than that of CTCF-mediated loops identified using the same datasets (Extended Data Fig. 7a). Importantly, genes with a greater number of interacting enhancers tended to exhibit higher transcription levels (Fig. 6a and Extended Data Fig. 7b), and this enhancer connectivity was closely associated with transcriptional differences between the two cell lines (Extended Data Fig. 7c).

### House-keeping genes and enhancers

Using RNA-sequencing (RNA-seq) data from 116 human tissues and cell lines (data sources are provided in the Methods and Supplementary Information), we observed a strong correlation between the number of interacting enhancers and the number of tissues in which a gene is expressed. Genes lacking enhancer interactions were generally tissue specific, whereas those with more than ten interacting enhancers in either H1 (Fig. 6b) or HFFc6 (Extended Data Fig. 7d) cells were notably enriched for genes expressed across all tissues (Methods). Among the 2,175 housekeeping genes annotated in the HRT Atlas v.1.0 database[101], more than 90% exhibited at least one distal enhancer interaction in both H1 and HFFc6 cells (Fig. 6c). Most of these enhancer–promoter loops were enriched in both Pol II ChIA-PET and H3K4me3 PLAC-seq contact maps, supporting their association with active transcription (Extended Data Fig. 7e). Additional analysis revealed that a substantial fraction of enhancer–promoter loops involving housekeeping genes also connect to distal promoters (Extended Data Fig. 7g), consistent with recent findings that housekeeping genes engage in extensive promoter-promoter interactions mediated by Ronin[102].

In the 3D modelling analysis described above, we categorized housekeeping genes into two classes: class I genes, which show strong nuclear speckle association; and class II genes, which do not (Fig. 4f,g). We found that class II housekeeping genes had significantly fewer interacting enhancers in both cell types (Extended Data Fig. 7f), consistent with our previous observation that speckle-associated regions tend to have more enhancers (Fig. 4h).

We also observed extensive differences in enhancer–promoter looping for housekeeping genes between cell types (Fig. 6d and Supplementary Fig. 8). Promoters of these genes frequently interacted with distinct enhancer regions in H1 and HFFc6 cells. To quantify this, we classified each enhancer–promoter pair as either cell-type-specific or shared. As shown in Fig. 6e, 80.3% of the pairs were specific to one cell type, suggesting that chromatin looping between housekeeping genes and distal enhancers is highly dynamic. APA analysis further confirmed that these cell-type-specific loops were enriched in the corresponding cell type but not in the other, across all chromatin interaction assays (Extended Data Fig. 7h,i), indicating that the observed loop dynamics reflect true biological differences rather than sampling artifacts.

Previous studies have suggested that housekeeping genes typically possess strong promoters that are less responsive to distal enhancers[103]. Moreover, a recent study by the ENCODE Consortium—based on CRISPR perturbation experiments and machine learning models—reported that housekeeping genes exhibit fewer functional enhancer interactions than cell-type-specific genes[8], which may appear to contradict our conclusions. To further consolidate our observations, we extended our analysis to include 32 additional cell lines and primary cells for which all necessary datasets were available from the ENCODE data portal. Given sample-to-sample variation in sequencing depth and

the number of detected enhancer–promoter loops, we grouped genes into 11 bins based on the percentile rank of their number of interacting enhancers in each sample. Consistent with our findings in H1 and HFFc6 cells, this broader analysis confirmed that genes with more enhancer interactions were more likely to be expressed across all tissues (Extended Data Fig. 7j). Moreover, sets of genes that exhibited extensive interactions with distal enhancers across a larger number of samples were more enriched for housekeeping genes (Extended Data Fig. 7k).

It is important to note that the enhancer–promoter loops identified in our analysis represent high-confidence physical interactions, not all of which may have direct regulatory functions. Redundancy among multiple enhancers can complicate the identification of functionally relevant pairs in perturbation experiments, where individual enhancers are tested one at a time. Further studies are needed to investigate how multiple enhancer–promoter loops are coordinated to regulate gene transcription, whether the abundance of such interactions is a cause or consequence of strong promoter activity, and whether they contribute to the robustness of housekeeping gene expression across diverse cell types.

### Enhancer–promoter loops near the lamina

LADs provide a generally repressive nuclear environment in which genes are typically silenced, although some can escape this repression, often exhibiting weaker interactions between their promoters and the nuclear lamina[104]. Consistent with a previous study[105], we observed that—albeit less frequently than in other nuclear environments—genes located within LADs can also form interactions with distal enhancers (Extended Data Fig. 8a,b). These genes were more likely to be expressed than other LAD-resident genes lacking enhancer interactions (Extended Data Fig. 8c). Conversely, among LAD-resident genes, those that were actively expressed (transcripts per kilobase per million (TPM) > 1) exhibited significantly more enhancer interactions than inactive ones (TPM < 1) (Extended Data Fig. 8d).

Inspection of lamin B1 DamID-seq signals surrounding several of these genes and their interacting enhancers (Fig. 6f and Extended Data Fig. 8e) revealed that both promoters and their associated enhancers were located in small regions enriched for active chromatin marks and depleted for lamin B1 signals. This pattern suggests that these genes may need to be locally looped out of the lamina-associated domain to establish functional enhancer–promoter communication (Fig. 6g and Extended Data Fig. 8f).

### Functional domains at different scales

We integrated A/B compartments, SPIN states, TADs, subTADs and loops to investigate links between specific structural categories of TADs and subTADs with replication timing and A/B compartments (Fig. 7a). SPIN states in H1 cells show distinct replication timing when aligned with high resolution 16-fraction Repli-seq[106] (Fig. 4a (middle)). When SPIN states were classified as co-registered with, nested within or encompassing A/B compartments (Fig. 7b), A/B compartments that encompass or co-register with SPIN states are generally larger (300 kb to 2 Mb) and span more than 80% of the human genome (Supplementary Fig. 9). We next examined whether SPIN states subdivide compartments into discrete genomic domains. Notably, the Near_Lm1/2 and Interior_Repr1/2 SPIN states are embedded within both A and B compartments, suggesting context-dependent regulatory roles. To explore this further, we analysed SPIN states in relation to replication timing and transcription using 16-fraction Repli-seq, RNA-seq and iMARGI data (Fig. 7d and Extended Data Fig. 9). Speckle and Interior_Act1 SPIN states exhibited enrichment for highly transcribed genes and earlier replication timing, suggesting that they represent local functional units within larger A compartments (Fig. 7c and Extended Data Fig. 9a). Similarly, lamina SPIN states within B compartments showed largely depleted transcription and even later replication timing compared to

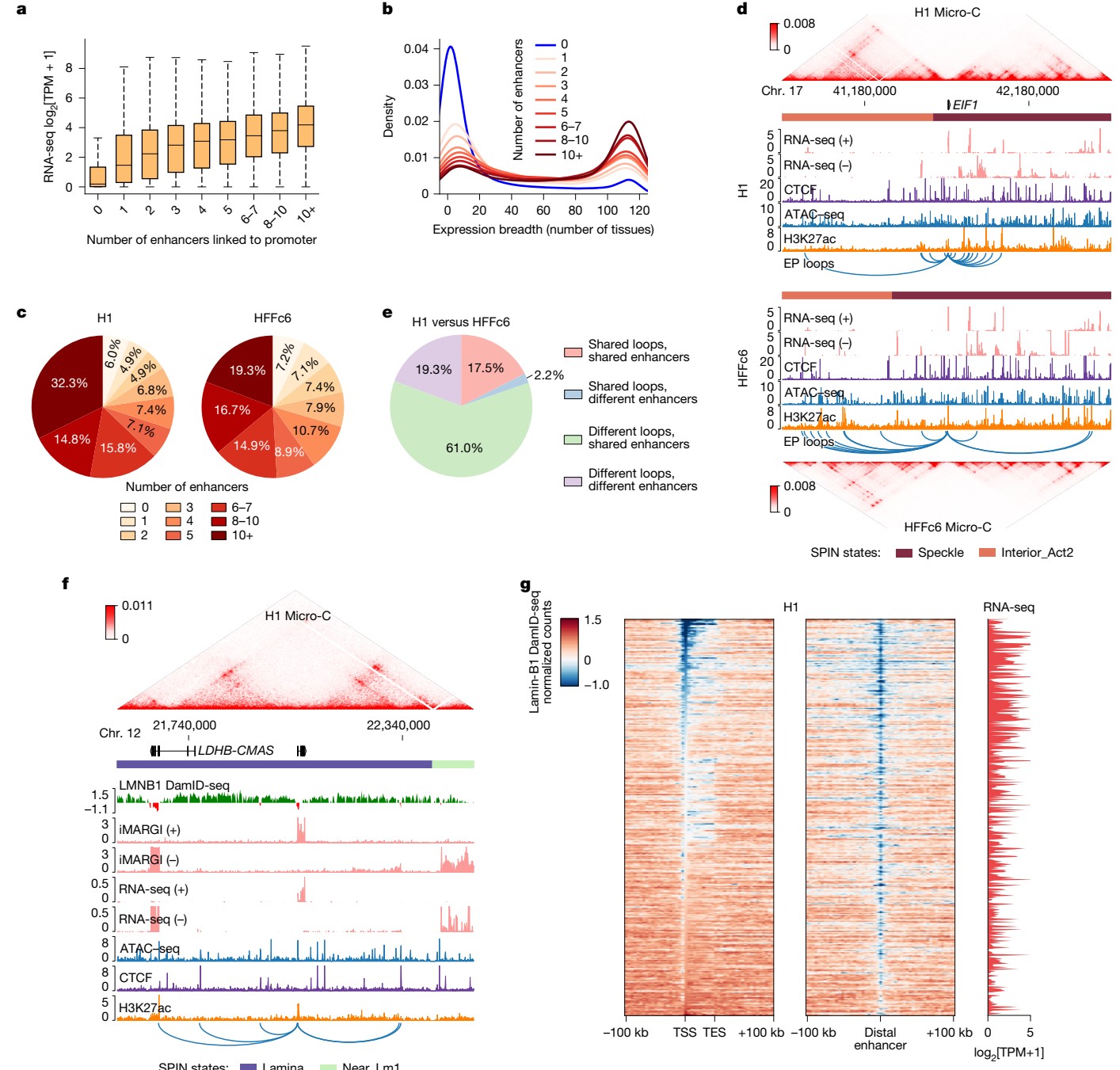

**Fig. 6 | Associations between enhancer–promoter loops and gene regulation.**
**a**, Gene transcription levels versus the number of interacting enhancers in H1 cells. In each box plot, the centre line indicates the median, the box limits represent the upper and lower quartiles, and the whiskers extend to 1.5 × IQR. The number of genes for each group (from left to right) is 5,328, 1,696, 1,540, 1,506, 1,342, 1,195, 1,981, 2,004 and 3,056, respectively. **b**, Expression breadth (that is, the number of tissues in which a gene is expressed) for genes with different numbers of interacting enhancers in H1 cells. **c**, The percentage of housekeeping genes with different numbers of interacting enhancers (HRT Atlas v1.0). **d**, Genome browser view of the region surrounding the housekeeping gene *EIF1*. The blue arcs represent chromatin loops linking the *EIF1* promoter with distal enhancers. **e**, Dynamics of chromatin loops linking housekeeping gene promoters and distal enhancers between H1 cells and HFFc6. **f**, Genome browser view of the *CMAS* locus in H1 cells. **g**, Lamin-B1 DamID-seq signals surrounding lamina-associated genes and their interacting enhancers in H1 cells. TES, transcription end site.

the B compartment expectation (Fig. 7c and Extended Data Fig. 9b). Together, these results reinforce that SPIN states represent local neighbourhoods of transcription and replication timing, and that their compartment context is critical to predict their functional impact.

We next examined functional patterns across TADs and loops accounting for their larger compartment and SPIN state environment. We identified TADs and nested subTADs in H1 cells using 3DNetMod[107] and stratified the domains by looping structural features (dot and dotless TADs and subTADs) as previously reported[108] (Supplementary Fig. 10a). We found that more than half of dot and dotless TADs and subTADs are nested within a single SPIN state (Supplementary Fig. 10b,c). To evaluate the interplay of domains and SPINs, we focused on TADs and subTADs that are nested within or co-registered with SPINs embedded within A or B compartments. We observed that nearly all dot and dotless TADs and subTADs resemble the replication timing of the larger SPIN state or A/B compartment and do not exhibit clear local

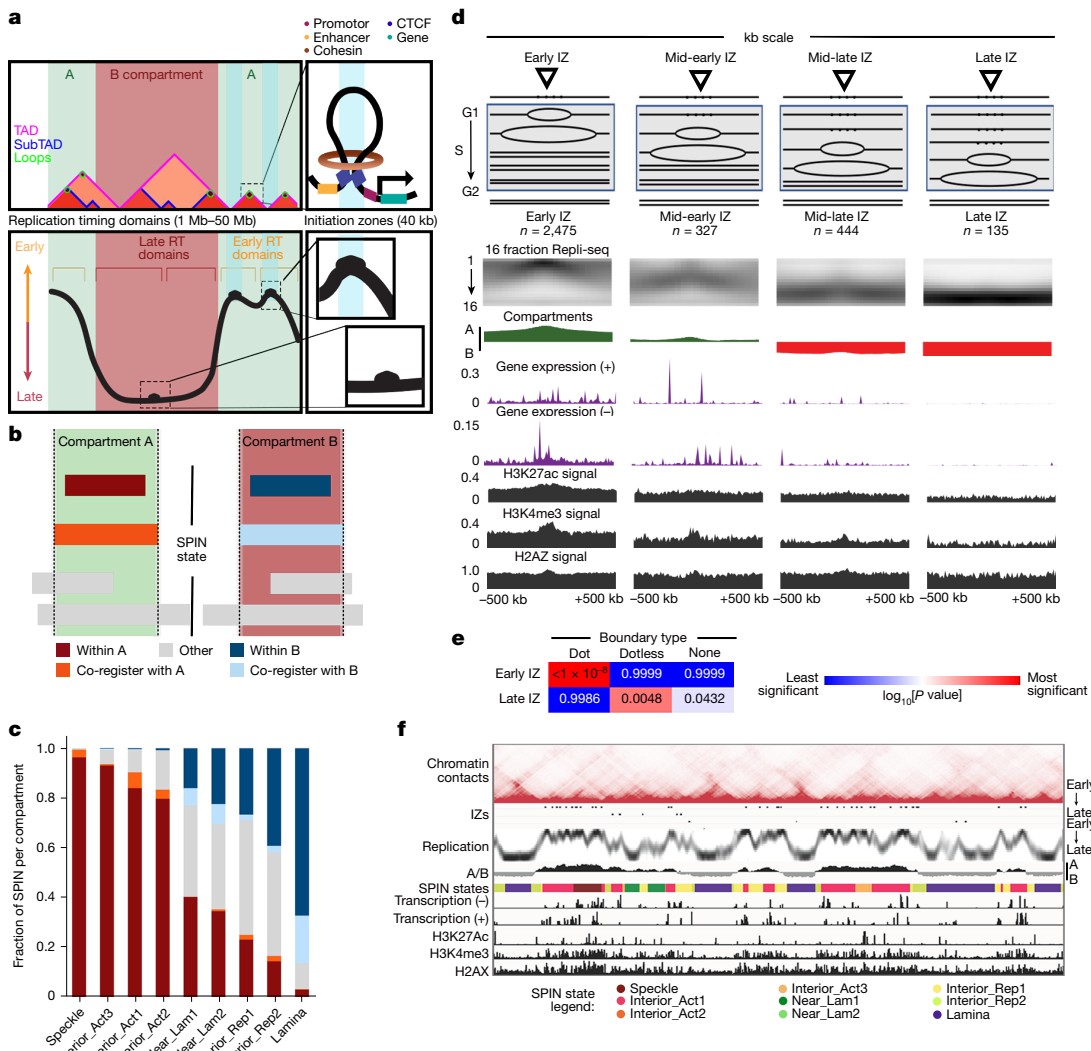

**Fig. 7 | A/B compartments and SPIN states represent subnuclear regions of distinct replication timing and gene expression. a**, Schematic of human genome folding into A/B compartments, SPIN states, TADs, subTADs and loops integrated with early/late replication timing and IZs. **b**, SPIN states were classified as either fully embedded within A/B compartments (within), co-registering A/B compartments (co-register) or partially overlapping (other). **c**, The fraction of each SPIN state co-registered or nested within A/B compartments in H1 cells. **d**, The average chromatin landscape at IZs in H1 cells. IZs have been grouped depending on their replication timing (RT). The tracks represent the

high-resolution replication timing, chromatin compartments, expression and histone marks. **e**, We computed right-tailed, one-tailed empirical *P* values using a resampling test with size and A/B compartment-matched null IZs for the intersection of early and late S phase IZs with dot boundaries, dotless boundaries and no boundaries. **f**, An example of chromatin profiles around IZs (portion of chromosome 2 from 20 Mb to 58 Mb). The tracks represent the chromatin contacts, four groups of IZs depending on their replication timing, the high-resolution replication timing, chromatin compartments, the SPIN states, expression (minus and plus strands), H3K27Ac, H3K4me3 and H2AX.

replication timing neighbourhoods (Supplementary Fig. 10d–g). Dot and dotless TADs and subTADs within more active SPIN states (speckles, Interior_Act1/2/3) show enrichment of local co-regulated gene expression domains and peaks of active genes localized at both boundaries (Extended Data Fig. 9d–g). Dot domains have stronger enrichment of gene expression at boundaries than dotless domains, and this pattern is more apparent when the dot domains are in A compartments. By contrast, dot and dotless TADs and subTADs within more repressive SPIN states largely do not show enrichment of expressed genes at boundaries. These data together indicate that the megabase-scale folding patterns of SPIN states and compartments more closely resemble replication timing domains compared to TADs and subTADs. TAD/subTAD boundaries, but not SPIN/compartment boundaries, are enriched for actively transcribed genes.

A distinct essential feature of genome function is the replication initiation zone (IZ; Fig. 7d–f), approximately 50-kb regions within which

replication initiates at one or more of many potential sites. Initiation within each IZ occurs in 5–20% of cell cycles[109], with the probability of an IZ firing early in S phase regulated in part by *cis*-acting elements termed early replication control elements[110]. Notably, dot TAD boundaries are enriched for early-firing IZs, whereas dotless TAD boundaries are enriched for late-firing IZs (Fig. 7e), revealing a biologically significant structure–function relationship correlated with cohesin-mediated chromatin extrusion[108]. However, when IZs in H1 cells, HCT116 cells and F121-9 mouse ES cells are stratified by their timing of firing and aligned to the insulation score of replication timing-matched sequences, this relationship to TAD boundaries is masked (Supplementary Fig. 11a–c) despite maintaining the canonical correlations of replication timing to gene expression and active histone marks (Supplementary Fig. 11d–l), consistent with IZs being enriched at a specific subcategory of TAD boundaries, namely, dot domains of which the boundaries co-localize with active transcription[108].

In summary, the relationship between higher-order chromatin structure and function is dependent on the length scale of the folding feature. Mb-scale folding patterns of SPIN states and compartments best correlate with replication timing domains. By contrast, TADs and subTADs appear to reflect replication timing of the larger compartment and SPIN state in which they reside, and do not show clear local replication timing neighbourhoods but rather enrichment of gene expression at boundaries. Folding features of TADs show clear functional diversity, exemplified by strong enrichment of early IZs at a specific subcategory of TAD boundaries.

## Discussion

We present results from an integrated project that was a focus of the first phase of the international 4D Nucleome Project. In the current ongoing second phase of the 4D Nucleome Project, a focus is on integrating genomic datasets with imaging data, development and application of a range of multi-omic single-cell datasets, and the analysis of 4D nucleome changes during development and in disease[29,30].

Here we provide an exceptionally detailed view of the human 4D nucleome, enabled by the integration of data obtained with a range of genomic methods. We show how each of these methods quantitatively contributes unique and common aspects of genome folding. We describe connections between chromosome folding and looping, nuclear positioning, proximity to nuclear bodies, cell-to-cell variation in organization and genomic functions such as transcription and replication.

The work provides tangible results. First, the extensive integration of a range of genomic datasets reporting on the spatial organization of the human genome in two cell types allowed us to benchmark these methods and to show which methods are best for specific inquiries. On the basis of these findings, we present a user guide in the form of a table and decision tree to provide advice on which methods to use for specific research questions (Supplementary Fig. 12). We find that all methods have their own strengths and weaknesses. Compartmentalization is most effectively detected using SPRITE and Hi-C, whereas looping interactions are best detected using Micro-C (especially structural loops), and enrichment-based assays such as PLAC-seq and ChIA-PET (gene expression related loops). The longer capture radius of SPRITE and GAM enable detection of colocalization of loci around larger subnuclear bodies. GAM and single-cell Hi-C can be applied when rare or mixed cell types are studied. Moreover, comparison with publicly available promoter capture Hi-C data in H1 cells revealed partial overlap with loops detected by Pol II ChIA-PET and H3K4me3 PLAC-seq, and identified additional interactions involving less-active regions, highlighting the complementary value of targeted capture-based assays (Supplementary Fig. 13).

Second, integration of these different and distinct genomic datasets has allowed us to compile an extensive catalogue of looping interactions between specific *cis*-elements, including CTCF–CTCF interactions, and interactions among promoters and enhancers for two widely used cell types. No single method was able to detect the full set of loops. Besides providing a resource that can be mined for future studies, this collection suggests mechanistic connections between the 4D nucleome and genome regulation. For example, we find that cohesin is enriched at a large proportion of anchors of all types of loops, consistent with cohesin's known role in loop formation through extrusion, although other mechanisms can also have roles. We find a strong correlation between expression of genes and the number of loops with distal putative enhancers these genes engage in. Housekeeping genes are particularly prone to interacting physically with distal enhancers, but with different sets of enhancers in different cell types. It is possible that this promiscuity in long-range interactions allows these genes to be expressed in many different cell types. However, we cannot rule out these are non-functional interactions that reflect the active transcriptional state of these genes.

Third, integration of the panel of genomic datasets enabled the generation of a detailed annotation track of spatial information, SPIN states, along the genome. This linear representation of 4D nucleome information will greatly facilitate integration of spatial genome data with other genomic datasets obtained in the larger community. In one example, our analysis of SPIN states, compartments and TADs, and DNA replication timing shows that one needs to take the heterogeneity of TADs into account when assigning a biological function to their structure. Indeed, we expect novel functions to be assigned to specific subsets of TADs.

Finally, we generated ensembles of spatial models through integrating and combining these datasets. Detailed analysis of these models starts to place genome functions such as transcription and replication in the 3D context, for example, in relation to nuclear bodies such as the nuclear lamina and nuclear speckles. The models also reflect the extensive cell-to-cell heterogeneity that defines the 4D nucleome, as also detected with single-cell assays. These models can provide a powerful resource for future studies, for example, to benchmark single-cell assays and imaging-based assays currently ongoing in the 4D Nucleome Project and elsewhere. One example is that the models highlight the existence of two types of housekeeping gene that occupy two quite distinct subnuclear neighbourhoods.

Moving forward, the rapid advancements in single-cell analysis of chromatin architecture, using either sequencing-based or imaging-based approaches, will undoubtedly further enhance our understanding of the role of 3D genome conformation in development, ageing, and disease pathogenesis. Indeed, in phase II of the 4D Nucleome Project, single-cell chromatin architecture assays have been broadly applied to a variety of biological processes, from cardiomyocyte differentiation, neuronal lineage specification, to heart failure, metabolic disorders and ageing. Compared with bulk assays such as Hi-C, Micro-C and other methods described here—which allow for detailed dissection of structural features, particularly in homogeneous cell populations—single-cell assays are particularly suitable for characterization of chromatin architecture in complex tissues and heterogeneous cell populations, owing to their ability to capture structural variability across cell types in the samples. However, sequencing-based assays such as scHi-C[111,112], snm3C-seq/Methyl-HiC[113,114], scSPRITE[115], HiRES[116], GAGE-seq[117] and Droplet Hi-C[118] are limited in the sparsity of chromatin contacts detected in each cell, necessitating the aggregation of many cells of the same cell type (pseudo-bulk) and development of advanced analytical tools for downstream in-depth analysis. Imaging-based approaches using FISH, such as chromatin tracing[82,99,119], ORCA[120], Oligo-DNA paint[121] and seqFISH[122,123], enable visualization of chromatin conformation in the 3D space at suboptical diffraction resolution, providing a powerful tool for analysing the chromatin architecture at single-molecular resolution across diverse cellular and tissue contexts. However, current FISH based strategies are still hindered by the number of loci being assayed in an experiment, resulting in either reduced genome coverage or genomic resolution.

One exciting new direction of the project is the development of approaches to use 4D nucleome data to predict cell-type-specific chromosome conformation from sequence[96,124–127]. One important application of these models is the identification of *cis*-elements, and thereby potentially new mechanisms, that drive chromosome folding. This can be done using explainable artificial intelligence techniques, such as importance scores[128] and in silico mutagenesis[129,130]. These approaches can predict effects of synthetic manipulations of the reference genome, as well as observed, and possibly disease-related, genetic variants on chromosome folding and chromatin looping between elements and thus start to relate alterations in chromosome folding to disease. To pilot these abilities, we trained separate models on 4D nucleome H1 cell and HFFc6 cell Micro-C data and used them to predict the effects of mutating motifs of cell-type-specific TFs at TAD boundaries unique

to each cell type. We found that the H1 cell model is more sensitive to mutation of motifs for the embryonic stem cell factors POU2F1–SOX2 (refs. 131,132), whereas the HFFc6 model is more sensitive to the fibroblast factors FOSL1–JUND[133] (Extended Data Fig. 10).

These findings suggest that deep learning models and explainable artificial intelligence can be used to screen DNA sequences at scale for the unbiased discovery of genome folding mechanisms and their associations with genome function. The potential of this approach is demonstrated by recent studies of cancer associated variants[124,134], de novo variants in individuals with autism[135] and an unbiased genome-wide screen of synthetic variants that revealed the importance of repetitive elements in genome-folding[129]. One caveat that future work needs to address is the fact that deep learning models tend to underestimate the true effects of genetic changes. While existing deep-learning models rely on one or a small set of data modalities (for example, Hi-C and Micro-C), in the future these models can be trained on richer models of the 4D nucleome, based on integration of multiple data types, building on recent publications focused on epigenetic marks[136–138] and polymer simulations[139], as well as emerging 4D nucleome datasets that integrate imaging and single-cell multi-omics data. It will be important to perform benchmarking studies to evaluate how well current and future predictive models can predict the effects of variants and decode sequence determinants of genome folding.

Taken together, the rich datasets on genome folding presented here, and their integration, reveal a detailed view of the living physical human genome as it is organized inside cells, and provide a foundation for future deep exploration of the structure and function of genomes, in humans and across the tree of life, in normal and disease states.

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

[1]Department of Systems Biology, University of Massachusetts Chan Medical School, Worcester, MA, USA. [2]Howard Hughes Medical Institute, Chevy Chase, MD, USA. [3]Ray and Stephanie Lane Computational Biology Department, School of Computer Science, Carnegie Mellon University, Pittsburgh, PA, USA. [4]Department of Microbiology, Immunology, and Molecular Genetics, Institute for Quantitative and Computational Biosciences, University of California Los Angeles, Los Angeles, CA, USA. [5]Department of Genetics, Department of Bioengineering, Epigenetics Institute, University of Pennsylvania, Philadelphia, PA, USA. [6]Gladstone Institutes, San Francisco, CA, USA. [7]Novo Nordisk Foundation Center for Protein Research, University of Copenhagen, Copenhagen, Denmark. [8]Department of Biochemistry and Molecular Biotechnology, University of Massachusetts Chan Medical School, Worcester, MA, USA. [9]Department of Biochemistry and Molecular Genetics, Feinberg School of Medicine Northwestern University, Chicago, IL, USA. [10]Basic Sciences Division, Fred Hutchinson Cancer Center, Seattle, WA, USA. [11]Department of Epigenetics, Van Andel Institute, Grand Rapids, MI, USA. [12]Epigenetic Regulation and Chromatin Architecture Group, Max-Delbrück-Center for Molecular Medicine in the Helmholtz Association (MDC), Berlin Institute for Medical Systems Biology (BIMSB), Berlin, Germany. [13]Department of Cellular and Molecular Medicine, University of California, San Diego School of Medicine, La Jolla, CA, USA. [14]State Key Laboratory of Genetics and Development of Complex Phenotypes, School of Life Sciences, Fudan University, Shanghai, China. [15]Department of Cell and Developmental Biology, University of Illinois at Urbana-Champaign, Urbana, IL, USA. [16]San Diego Biomedical Research Institute, San Diego, CA, USA. [17]Division of Gene Regulation, Netherlands Cancer Institute, Amsterdam, The Netherlands. [18]Oncode Institute, Utrecht, The Netherlands. [19]Life Sciences Institute, Zhejiang University, Hangzhou, P.R. China. [20]Department of Computational Medicine and Bioinformatics, University of Michigan, Ann Arbor, MI, USA. [21]Department of Genome Sciences, University of Washington, Seattle, WA, USA. [22]Division of Biology and Biological Engineering, California Institute of Technology, Pasadena, CA, USA. [23]Department of Chemical and Biological Engineering, Princeton University, Princeton, NJ, USA. [24]David Geffen School of Medicine at UCLA, Los Angeles, CA, USA. [25]Shu Chien-Gene Lay Department of Bioengineering, University of California San Diego, La Jolla, CA, USA. [26]Laboratory of Bioinformatics and Computational Genomics, Faculty of Mathematics and Information Science, Warsaw University of Technology, Warsaw, Poland. [27]Laboratory of Functional and Structural Genomics, Centre of New Technologies, University of Warsaw, Warsaw, Poland. [28]Centre for Advanced Materials and Technologies CEZAMAT, Warsaw University of Technology, Warsaw, Poland. [29]Department of Physics, University of Naples "Federico II", and INFN, Naples, Italy. [30]Program in Bioinformatics and Systems Biology, University of California San Diego, La Jolla, CA, USA. [31]Department of Quantitative Health Sciences, Lerner Research Institute, Cleveland Clinic Foundation, Cleveland, OH, USA. [32]Department of Biostatistics, Department of Genetics, University of North Carolina, Chapel Hill, NC, USA. [33]In silichrom Ltd, Newbury, UK. [34]Department of Biomedical Informatics, Harvard Medical School, Boston, MA, USA. [35]Department of Genetics, Center for Genome Sciences and Systems Biology, Washington University School of Medicine, St Louis, MO, USA. [36]Institute of Reproduction and Development, Shanghai Key Laboratory of Reproduction and Development, Obstetrics and Gynecology Hospital, Fudan University, Shanghai, China. [37]Robert H. Lurie Comprehensive Cancer Center of Northwestern University, Chicago, IL, USA. ✉e-mail: job.dekker@umassmed.edu; wangxiaotao@fudan.edu.cn; mario.nicodemi@na.infn.it; bren@nygenome.org; szhong@ucsd.edu; jennifer.cremins@wustl.edu; gilbert@sdbri.org; katherine.pollard@gladstone.ucsf.edu; falber@g.ucla.edu; jianma@cs.cmu.edu; william-noble@uw.edu; Yue@northwestern.edu

# Methods

## GAM methods and data processing

**Preparation of cryosections.** H1 cells were fixed and processed for cryosectioning as described previously[140]. In brief, H1 cells were grown to 70% confluency, the medium was removed and cells were fixed in 4% and 8% paraformaldehyde in 250 mM HEPES-NaOH (pH 7.6; 10 min and 2 h, respectively), gently scrapped and softly pelleted, before embedding (>2 h) in saturated 2.1 M sucrose in PBS and frozen in liquid nitrogen on copper sample holders. Frozen samples were stored in liquid nitrogen. Ultrathin cryosections were cut using a Leica ultracryomicrotome (UltraCut EM UC7, Leica Microsystems) at approximately 220 nm thickness, captured on sucrose-PBS drops and transferred to 4 μm PEN steel frame slide for laser microdissection (Leica Microsystems, 11600289). Sucrose embedding medium was removed by washing with 0.2-μm-filtered molecular biology grade PBS (3 × 5 min each) and filtered ultrapure water (5 min). For laser microdissection, cryosections on PEN membranes were washed, permeabilized and incubated (2 h, room temperature) in blocking solution (1% BSA (w/v), 5% FBS (w/v, Gibco 10270), 0.05% Triton X-100 (v/v) in PBS). After incubation (overnight, 4 °C) with primary anti-pan-histone (1:50) antibody (Merck, MAB3422) in blocking solution, the cryosections were washed three to five times for 30 min in 0.025% Triton X-100 in PBS (v/v) and immunolabelled (1 h, room temperature) with secondary antibodies in blocking solution, followed by three (15 min) washes in PBS.

**Isolation of NPs.** Nuclear staining was visualized using a Leica laser microdissection microscope (Leica Microsystems, LMD7000) using a ×63 dry objective. Individual nuclear profiles (NPs) were laser microdissected from the PEN membrane, and collected into PCR adhesive caps (AdhesiveStrip 8C opaque; Carl Zeiss Microscopy 415190-9161-000). GAM data were collected in multiplexGAM mode, where three NPs are collected into each adhesive cap. The presence of NPs in each lid was confirmed with a ×5 objective using a 420–480 nm emission filter. Control lids not containing NPs (water controls) were included for each dataset collection to keep track of contamination and noise amplification of whole genome amplification and library reactions. Collected NPs were kept at −20 °C until whole-genome amplification.

**WGA.** Whole genome amplification (WGA) was performed as described previously[141] with minor modifications. In brief, DNA was extracted from NPs at 60 °C in the lysis buffer (20 mM Tris-HCl pH 8.0, 1.4 mM EDTA, 560 mM guanidinium-HCl, 3.5% Tween-20, 0.35% Triton X-100) containing 0.75 U ml$^{-1}$ Qiagen protease (Qiagen, 19155). After 24 h of DNA extraction, the protease was heat-inactivated at 75 °C for 30 min and the extracted DNA was amplified by two rounds of PCR. First, quasi-linear amplification was performed with random hexamer GAT-7N primers with an adaptor sequence. The lysis buffer containing the extracted genomic DNA was mixed with 2× DeepVent mix buffer (2× Thermo polymerase buffer (10×), 400 μm dNTPs, 4 mM MgSO$_4$ in ultrapure DNase-free water), 0.5 μM GAT-7N primers (5′-GTGAGTGATGGTTGAGGTAGTGTGGAGNNN NNNN) and 2 U μl$^{-1}$ DeepVent (exo-) DNA polymerase (New England Biolabs, M0259L) and incubated for 11 cycles in the BioRad thermocycler. The second exponential PCR amplification was performed in presence of 1x DeepVent mix, 10 mM dNTPs, 0.4 μM GAM-COM primers (5′-GTGAGTGATGGTTGAGGTAGTGTGGAG) and 2 U μl$^{-1}$ DeepVent (exo-) DNA polymerase in the programmable thermal cycler for 26 cycles. WGA was performed in 96-well plates using Microlab STARLine liquid handling workstation (Hamilton).

**Preparation of GAM libraries for high-throughput sequencing.** After WGA, the samples were purified using the SPRI magnetic beads (1.7× ratio of beads per sample volume). The DNA concentration of each purified sample was measured using the Quant-iT PicoGreen dsDNA assay kit (Invitrogen, P7589). Sequencing libraries were then made using the in-house tagmentation-based protocol. After library preparation, the DNA concentration for each sample was measured using the Quant-iT PicoGreen dsDNA assay, and equal amounts of DNA from each sample was pooled together. The final pool of libraries was analysed using DNA High Sensitivity on-chip electrophoresis on an Agilent 2100 Bioanalyzer and sequenced on Illumina NextSeq 500 machine.

**GAM data sequence alignment.** Sequenced reads from each GAM library were mapped to the human genome assembly GRCh38 (December 2013, hg38) with bowtie2 (v.2.3.4.3) using the default settings. All non-uniquely mapped reads, reads with mapping quality <20 and PCR duplicates were excluded from further analyses.

**GAM data window calling and sample quality control.** Positive genomic windows present within ultrathin nuclear slices were identified for each GAM library as previously described[141]. In brief, the genome was split into equal-sized windows, and the number of nucleotides sequenced in each bin was calculated for each GAM sample with bedtools. Next, we determined the percentage of orphan windows (that is, positive windows that were flanked by two adjacent negative windows) for every percentile of the nucleotide coverage distribution. The number of nucleotides that corresponds to the percentile with the lowest percentage of orphan windows in each sample was used as an optimal coverage threshold for window identification in each sample. Windows were called positive if the number of nucleotides sequenced in each bin was greater than the determined optimal threshold.

The sample quality was assessed by the percentage of orphan windows in each sample, total genomic coverage in percent of positive windows, the number of uniquely mapped reads to the mouse genome and the correlations from cross-well contamination for every sample. Each sample was considered to be of good quality if it had ≤40% orphan windows, ≤60% of total genome coverage, >50,000 uniquely mapped reads and a cross-well contamination score determined per collection plate of <0.4 (Jaccard index).

**GAM data curation.** To exclude genomic windows which were under- or oversampled in the GAM collection, we computed a GAM-specific parameter, the window detection efficiency (WDF)[31] as previously described[142]. To detect genomic bins with outlying detection frequency, a smoothing algorithm was applied to the WDF values per chromosome in stretches of eleven equally sized genomic windows. Next, normalized delta (ND) was calculated for each window, according to: ND = (raw_Signal − smoothed_Signal)/smoothed Signal. If the ND was larger than a fold change of 5, the window was removed from the final dataset. Next, the four adjacent windows (2 upstream and 2 downstream) were also removed, to ensure good quality of sampling in the final GAM data used for further analyses.

Genomic bins with an average mappability score below 0.2 were also removed. Genome mappability for the hg38 human genome assembly was computed using GEM-Tools suite[143] setting read length to 75 nucleotides. The mean mappability score was computed for each genomic bin with bigWigAverageOverBed utility from Encode.

**GAM data normalization.** GAM contact matrices for all pairs of windows genome-wide were generated as previously described to produce pairwise co-segregation maps and pointwise mutual information (NPMI) maps that consider window detectability[141]. For visualization of the contact matrices, scale bars are adjusted in each genomic region displayed to a range between 0 and the 99th percentile of NPMI values in the region.

**GAM insulation scores.** The insulation scores were calculated from NPMI-normalized pairwise chromatin contact matrices, as previously described[25] with minor modifications adjusted for GAM input data by keeping both positive and negative values in the matrix[141].

**Identification of compartments A and B in GAM data.** Compartments were calculated using co-segregation matrices, as previously described[142]. In brief, each chromosome was represented as a matrix of observed interactions $O(i,j)$ between locus $i$ and locus $j$. The expected interactions $E(i,j)$ matrix was calculated, where each pair of genomic windows is the mean number of contacts with the same distance between $i$ and $j$. A matrix of observed over expected values $O/E(i,j)$ was produced by dividing $O$ by $E$. A correlation matrix $C(i,j)$ was calculated between column $i$ and column $j$ of the $O/E$ matrix. PCA was performed for the first three components on matrix $C$ before extracting the component with the highest correlation to GC content. Loci with eigenvector values with the same sign were designated as A compartments, whereas those with the opposite sign were identified as B compartments. For chromosomes 3 and 22, we manually picked PC1 and PC2, respectively, as the PC that correlated most with GC content did not display a typical AB compartmentalization pattern and good correlation with transcription levels.

## Data processing for Hi-C, Micro-C, ChIA-PET, PLAC-seq and SPRITE

Cooler files for Hi-C, Micro-C, ChIA-PET, PLAC-seq and SPRITE were downloaded from the DCIC Data Portal (links to all data are provided in the Supplementary Information). Contact matrices were normalized using the iterative correction procedure from a previous study[144]. Interaction heat maps were created using Python. The colour map is 'YlOrRd' and the colour scales are created taking the 10th and 90th percentile of the interaction frequencies of individual datasets. No additional processing was applied to GAM data.

## HiCRep correlations

HiCRep is used to calculate distance-corrected correlations of the multiple methods[145]. Correlation is calculated in two steps. First, interaction maps are stratified by genomic distances and the correlation coefficients are calculated for each distance separately. Second, the reproducibility is determined by a novel stratum-adjusted correlation coefficient statistic by aggregating stratum-specific correlation coefficients using a weighted average. Chromosome-specific correlation was performed for pairwise protocols and the correlations across those chromosomes were then averaged. Averaged pairwise correlations of chromosomes 1–22 and X were used to calculate the HiCRep correlations between Hi-C, Micro-C, ChIA-PET, PLAC-seq and SPRITE datasets; averaged correlations of chromosomes 1–22 were used to calculate HiCRep correlations between GAM and all other datasets. We used 50-kb binned interaction matrices to calculate HiCRep correlations.

## Compartment analysis

Compartments were assessed for Hi-C, Micro-C, ChIA-PET, PLAC-seq and SPRITE using eigenvector decomposition on observed-over-expected contact maps at 100-kb resolution using the cooltools package derived scripts[146]. An eigenvector that has the strongest correlation with gene density is selected, then A and B compartments were assigned based on the gene density profiles such that A compartment has high gene density and B compartment has low gene density profile. A and B compartment assignments of GAM were provided by the data producers.

Spearman correlation was used to correlate the eigenvectors of different experiments performed with various protocols and cell states. Saddle plots were generated as follows[49]: the interaction matrix of an experiment was sorted based on the eigenvector values from lowest to highest (B to A). Sorted maps were then normalized for their expected interaction frequencies; the top left corner of the interaction matrix represents the strongest B–B interactions, the bottom right represents the strongest A–A interactions, the top right and bottom left are B–A and A–B, respectively. To quantify saddle plots, we took the strongest 20% of BB and strongest 20% of AA interactions and normalized them by the sum of AB and BA (top(AA)/(AB + BA) and top(BB)/(BA + AB)). Saddle quantifications were used to create the scatter plots. The parameters that were used for the saddle plot are as follows: --strength, --vmin 0.5, --vmax 2, --regions hg38_chromsizes.bed, --qrange 0.02 0.98, --contact-type cis.

## Preferential interactions

Bigwig or bedgraph files for lamin B1 DamID-seq, TSA–seq and Repli-seq were downloaded from the DCIC Data Portal (a full table of links is provided in the Supplementary Information).

Heat maps that integrate 3D methods with genome activity plots were generated as follows: first, the data were binned into 50-kb bins for aforementioned assays and sorted from the highest to the lowest value. Additional filters were applied for early/late replication ratio. For early/late replication timing data, bins with no values and the bins with value of 0 were removed, and the outlier bins that have values >98th quantile were also removed and the minimum value for the first bin was kept as 0.

Second, the interaction matrices (Hi-C, Micro-C, ChIA-PET, PLAC-seq, SPRITE and GAM) are sorted based on the 1D tracks generated from the aforementioned assays from the highest to the lowest.

Next, sorted matrices were then normalized for their expected interaction frequencies; the upper left corner of the interaction matrix represents the strongest signal for non-preferential interactions, lower right represents strongest preferential interactions. To quantify these plots we took the strongest 20% of the preferential interactions. Saddle plot parameters are listed below for this quantifications: --strength, --vmin 0.5, --vmax 2, --regions hg38_chromsizes.bed, --qrange 0.02 0.98, --range min(Sorted 1D data) max(Sorted 1D data) --contact-type cis. For GAM, --strength, --vmin 0.1, --vmax 0.4, --regions hg38_chromsizes.bed, --qrange 0.02 0.98, --range min(Sorted 1D data) max(Sorted 1D data) --contact-type cis.

## Insulation score

For Hi-C, Micro-C, ChIA-PET, PLAC-seq and SPRITE, we calculated diamond insulation scores using cooltools (https://github.com/open2c/cooltools/blob/master/cooltools/cli/insulation.py) as implemented previously[25]. We defined the insulation and boundary strengths of each 25-kb bin by detecting the local minima of 25-kb binned data with a 100-kb window size. We used cooltools's diamond-insulation function with the following parameters: --ignore-diags 2. Insulation scores of GAM were provided by data producers. We separated weak and strong $\log_2$-transformed insulation scores using the mean insulation score of all protocols (that is, weak insulation scores < mean < strong insulation scores). We piled up strong insulation scores to compare the average insulation score strengths of methods.

## Identification of chromatin loops in different platforms

We used different strategies for detecting chromatin loops in different platforms.

For Hi-C and Micro-C, we combined results from HiCCUPS[23] and Peakachu[63]. To identify chromatin loops using HiCCUPS, we ran cooltools dots (v.0.5.1)[146] at 5-kb and 10-kb resolutions with the default parameters. Peakachu is a machine-learning based framework that learns contact patterns of pre-defined chromatin loops from a genome-wide contact map and applies trained models to predict loops on other maps generated by the same/similar experimental protocol. Here, we first trained Peakachu models on GM12878 Hi-C data at 2-kb, 5-kb and 10-kb resolutions, using a high-confidence loop set detected by at least two platforms among Hi-C, CTCF ChIA-PET, Pol2 ChIA-PET, CTCF HiChIP, H3K27ac HiChIP, SMC1A HiChIP, H3K4me3 PLAC-seq and TrAC-loop. These models were then used to predict chromatin loops on Hi-C and Micro-C maps of H1 and HFFc6 cell lines at corresponding resolutions. The probability cut-offs were manually adjusted to balance sensitivity and specificity based on visual inspection.

For ChIA-PET, we combined loop predictions from ChiaSig[147] and Peakachu. For each ChIA-PET dataset, we conducted multiple runs

of ChiaSig with varying parameter settings, specifically adjusting the -M, -C and -c parameters while keeping other parameters constant (-m 8000 -S 4 -s 6 -A 0.01 -a 0.1 -n 1000). The -M value was selected from 1000000, 2000000 and 4000000, while both the -C and -c values were set to either 2 or 3. Only chromatin loops consistently identified across all parameter settings were retained, while others were discarded. As ChiaSig heavily relies on 1D peak annotation for loop detection, chromatin interactions outside peak regions are not identified as loops. To capture loops with similar contact patterns to those detected by ChiaSig but outside peak regions, we again used Peakachu to learn the patterns. For each ChIA-PET dataset, we trained 23 Peakachu models using interactions detected by ChiaSig, with each model trained on data from different combinations of 22 chromosomes. During prediction, loops on each chromosome were predicted using the model trained on the other 22 chromosomes. The probability cut-offs were determined to ensure that Peakachu-predicted loops covered 90% of ChiaSig-detected interactions. Training and prediction were conducted separately at 2-kb and 5-kb resolutions, and the final loop predictions for each ChIA-PET dataset were obtained by combining ChiaSig-predicted interactions and Peakachu predictions.

For PLAC-seq, we combined outputs from MAPS[148] and a pipeline similar to that used for ChIA-PET. For MAPS, loops were identified at 10-kb resolution using the default parameters. For the ChiaSig-Peakachu pipeline, loop prediction was conducted at 5-kb resolution, using the same training, prediction and filtering strategies as described above for ChIA-PET.

To calculate the union of loops identified from different platforms, methods and resolutions, two chromatin loops $(i, j)$ and $(i', j')$ were considered overlapping if and only if $|i − i'| < \min(0.2|i − j|, 15\,\text{kb})$ and $|j − j'| < \min(0.2|i − j|, 15\,\text{kb})$. If two loops were overlapping, only the one predicted at a higher resolution (that is, with more precise coordinates) was retained.

## Calculation of overlap between capture Hi-C interactions and loops from other assays
For each capture Hi-C interaction $(i, j)$, if there exists a loop $(i', j')$ identified by another assay (one of Hi-C, Micro-C, CTCF ChIA-PET, Pol2 ChIA-PET or H3K4me3 PLAC-seq) such that the Euclidean distance between $(i, j)$ and $(i', j')$ is less than $\min(0.3|i − j|, 80\,\text{kb})$, we define $(i, j)$ as overlapping with a loop from the corresponding assay.

## Consensus chromatin-state annotations for H1 and HFFc6 cells
We computed epigenomic annotations using ChromHMM (v.1.23) on 14 observed and 2 imputed ChIP–seq datasets for 8 marks (H3K36me3, H3K4me1, H3K27ac, H3K9ac, H3K3me3, H3K4me2, H3K27me3 and CTCF) in both H1 and HFFc6 cells. All the ChIP–seq datasets were obtained from the WashU Epigenome Browser (https://epigenome.wustl.edu/epimap/data/) in bigwig format, and the coordinates were transformed from hg19 to hg38 using CrossMap (v.0.5.2, http://crossmap.sourceforge.net/).

To prepare the data for ChromHMM, we divided the whole genome into 200-bp bins and calculated the average signals within each bin. For the observed data, values were binarized with a $-\log_{10}[P]$ value cut-off of 2. For the imputed data (H3K9ac and H3K4me2 in HFFc6), we downloaded both the imputed and observed data in H1 cells for the same marks. Then, for each mark, we set the binarization cut-off for the imputed data to match the quantile in the observed data corresponding to the $-\log_{10}[P] > 2$ cut-off, enabling comparison with the observed data.

Finally, we ran the ChromHMM LearnModel command on the binarized data to segment both the H1 and HFFc6 genomes into 12 states. The name of each state was manually annotated based on prior knowledge about each mark. The 12_Heterochrom state was excluded from further analysis, as it did not contain signals of any marks (Supplementary Fig. 2a).

## Enrichment analysis of chromatin states for chromatin loops and loop anchors
To characterize the chromatin states of loop anchors detected by specific combinations of chromatin interaction methods, we calculated fold-enrichment scores by comparing the overlap with each ChromHMM state between the observed loop anchors and 100 randomly generated control sets. Specifically, for each chromatin state, we iterated through the loop anchor list and counted the number of anchors overlapping at least one region with that state. We then randomly shuffled the anchors in the genome to generate 1,000 control sets and repeated the same procedure for each control. For each control, we kept the size distribution and the number of random regions on each chromosome the same as the observed loop anchors, and the intervals of each region did not overlap with any gaps in the hg38 reference genome. Finally, the fold-enrichment score was calculated by dividing the number of anchors with a specific chromatin state by the average number of random loci with the same state.

We used a similar method to characterize chromatin states for a specific cluster of chromatin loops. In brief, for each pair of chromatin states, we iterated through the loop list and counted the number of loops with one anchor overlapping regions marked by one chromatin state and the other anchor overlapping regions marked by the other chromatin state. Again, we generated 1,000 random control sets for chromatin loops. Each random loop set maintained the same genomic distance distribution between loop anchors and the same number of random loops on each chromosome, ensuring that the interval between the two ends of each loop did not overlap any gaps in hg38. Finally, the fold-enrichment score was calculated by dividing the number of loops between a specific pair of chromatin states by the average number of random loops between the same states.

## Enrichment analysis of TFs for different loop clusters
To explore whether different loop clusters exhibit differential binding of various TFs at their anchors, we downloaded the ENCODE ChIP–seq peak files for 62 TFs in H1 cells. A fold-enrichment score was computed for each TF at loop anchors using a procedure analogous to the one described above. In brief, we first identified non-redundant loop anchors from each loop cluster in H1 cells. For each TF, we iterated through this anchor list and counted the number of anchors overlapping at least one ChIP–seq peak. Subsequently, we generated 1,000 random control sets by shuffling the loops and repeated the same procedure for each control set. The fold-enrichment score was then calculated by dividing the number of anchors containing ChIP–seq peaks by the average number of random loci containing ChIP–seq peaks for the same TF.

## APA
To evaluate the overall enrichment of chromatin loop signals in contact maps, we performed APA. For a given list of chromatin loops or interactions, we extracted contact frequencies from $21 \times 21$ submatrices centred at the 2D coordinates of each loop. Each submatrix was normalized by dividing each value by the average contact frequency at the corresponding genomic distance. To reduce the influence of outliers, submatrices with average signal intensities above the 99th percentile or below the 1st percentile were excluded. The average signal at each position was then computed across all retained submatrices and visualized as a heat map. The APA score shown on the plot is defined as the signal intensity at the centre pixel of the aggregated heat map.

## UMAP projection and clustering of chromatin loops
To construct the input feature matrix for projecting chromatin loops, we calculated the proportion of each ChromHMM state at the loop anchors in each cell line. This yielded a feature matrix $M_{ij}$ of size $141{,}365 \times 22$ for H1 cells and $146{,}140 \times 22$ for HFFc6 cells. Each row

represents a chromatin loop, with the first 11 columns corresponding to features from one anchor and the next 11 columns corresponding to features from the other anchor.

Next, we standardized (z-score normalized) the matrix $M_{ij}$. For each row in the normalized matrix, we swapped the order of the two anchors, if necessary, to ensure that the highest value appeared in the first 11 columns. The resulting matrix served as input for UMAP projection (https://github.com/lmcinnes/umap), using the parameters "n_neighbors=60, min_dist=0, n_components=2, metric=euclidean" for H1 cells, and "n_neighbors=65, min_dist=0, n_components=2, metric=euclidean" for HFFc6 cells.

For loop clustering, we first applied principal component analysis (PCA) to the swapped feature matrix, resulting in a transformed matrix of the same shape. We then constructed multiple $k$-nearest neighbour ($k$-NN) graphs using values of $k$ ranging from 50 to 2,000 in steps of 50. The Leiden algorithm was applied to each graph for community detection, with the resolution parameter set to 0.5. Consensus clusters were derived by integrating clustering results across all $k$-NN graphs.

## Calculation of the average contact strength for different loop clusters

To calculate the average contact strength for each loop cluster across different experimental platforms, we used distance-normalized (observed/expected) contact frequencies. Specifically, for Hi-C, Micro-C and DNA SPRITE datasets, we computed this value using interaction frequencies normalized by matrix balancing or iterative correction and eigenvector decomposition (ICE) at the 5-kb resolution. By contrast, for ChIA-PET and PLAC-seq datasets, we calculated the value using raw interaction frequencies at the same 5-kb resolution. For GAM, we used the NPMI-normalized co-segregation frequencies at the 25-kb resolution.

## Annotation of enhancer regions in different cell types

To define candidate enhancer regions in each cell type, we first downloaded the total set of human cis-regulatory elements (cCREs) from the ENCODE data portal website using the following link https://screen.encodeproject.org/. We then extracted all regions marked as ELS (enhancer-like signatures) from the downloaded file. Finally, enhancer regions in each cell type were defined as a subset of these regions that overlap with ATAC–seq or DNase-seq peaks in corresponding cells, based on data availability for those cells.

## Gene expression breadth analysis

To explore the gene expression profiles of a specific gene set across a diverse range of cell type or tissues, we collected RNA-seq datasets for 116 human cell types or tissues (from ENCODE; see the 'Datasets' table in Supplementary Note 1, section 'Methods for relating chromatin loops to gene expression'). The TPM values were used to measure gene transcription levels. To normalize the RNA-seq data, we first applied a logarithm transformation to the original TPM values using the formula $\log_2[\text{TPM} + 1]$ for each sample, and then quantile-normalized the transformed TPM values across all samples.

In each sample, genes with a normalized TPM value greater than 3 were considered expressed in the corresponding sample, and the gene expression breadth is defined as the number of samples in which a gene is expressed.

## Identifying SPIN states for large-scale genome compartmentalization

In this work, we used a modified SPIN method to perform joint modelling across multiple cell types. To ensure TSA–seq and DamID scores across different cell types are comparable, we first applied a data normalization method to transform data into a Gaussian or more-Gaussian-like distribution. To do that we identified genomic bins that are spatially stable by calculating the Pearson correlation of interchromosomal Hi-C interactions for each non-overlapping 25-kb genomic

bin. The bins were then ranked on the basis of the average Pearson correlation coefficient, and the top 25% were selected as spatially conserved regions. We then obtained TSA–seq or DamID scores for these bins in all cell types and standardized each data track by fitting a power-transformation function. We used the Yeo–Johnson transformation function with the default parameters from the Python scikit-learn package. Next, we modified the framework of SPIN by jointly modelling the probability across multiple cell types. The hidden Markov random field model is defined on an undirected graph $G^c = (V, E^c)$ for each cell type, where in our case $V$ represents non-overlapping 25-kb genomic bins and $E^c$ represents the cell-type-specific edges (that is, significant Hi-C interactions) in cell type $c$. For each node $i \in V$, $O_i^c \in \mathbb{R}^d$ is a vector with dimension $d$ indicating the observed TSA–seq and DamID signal of this bin in cell type $c$. Each node $i$ in $C$ also has a hidden state $H_i^c$ for each cell type, representing its underlying spatial environment relative to different nuclear landmarks that we want to estimate. In this work, we assume that the set of hidden states are shared across cell types. Edges $(i^c, j^c) \in E^c$ in the graph are cell-type specific and there are no edges that are connecting nodes from different cell types. Therefore, the hidden state $H_i^c$ is only dependent on cell-type-specific observation $O_i^c$ and the neighbours of node $i(N^c(i) = \{j | j \in V, (i, j) \in E^c\})$ in cell type $c$. The overall objective is to estimate the hidden states $H_i^c$ for all nodes in all cell types that maximize the following joint probability as shown below:

$$P(\vec{H}, \vec{O}) \propto \frac{1}{Z} \prod_{c \in 1 \ldots 4} \left( \prod_{i \in V} P_V(O_i^c | H_i^c) \prod_{(i^c, j^c) \in E^c} P_{E^c}(H_i^c, H_j^c) \right)$$

To estimate the number of SPIN states, we applied the same approaches as we used in the previous version of the SPIN method[72]. We used both the Elbow method based on $k$-means clustering and AIC/BIC scores to search for the optimal number of SPIN states. Both AIC and BIC scores decrease as the number of states increases. We found that the slope of the curve drops close to zero as the number of states exceeds 9. So, we chose 9.

**Data acquisition and processing.** We obtained TSA–seq, lamin B1 DamID-seq and Hi-C data for H1 and HFFc6 cells from the 4D Nucleome data portal (http://data.4dnucleome.org). The data generation and processing pipeline for TSA–seq data was described previously[32,59]. The data generation and processing pipeline for DamID data was described previously[104]. For the SPIN states inference, we used Hi-C data generated by the formaldehyde–disuccinimidyl glutarate Hi-C protocol (1% formaldehyde followed by incubation with 3 mM disuccinimidyl glutarate) using restriction enzyme DpnII[42]. We binned TSA–seq, lamin B1 DamID-seq and Hi-C mapped reads at 25-kb resolution. We then identified significant interactions from the normalized Hi-C data in each cell type previously described[72].

## Processing other epigenomic data

We downloaded or processed other epigenomic data and compared SPIN states with these datasets. For ChIP–seq datasets, we downloaded the processed $P$-value tracks from the ENCODE website for H1 cells and Avocado[73] imputed $P$-value tracks from the 4D Nucleome data portal. Multi-fraction Repli-seq data were collected from a previous study[106]. For CUT&RUN data, we downloaded raw sequencing reads from the 4D Nucleome data portal and processed them using a similar procedure according to the standard ENCODE ChIP–seq pipeline. First, we mapped reads to the hg38 reference genome using Bowtie2 (v.2.2.9) with the default parameters. We then used MACS3 to generate $P$-value tracks as well as peaks for CUT&RUN data. The enrichment score of epigenomic data on SPIN states is determined by the $\log_2$-transformed ratio between the average observed signals on each SPIN states over genome-wide expectation.

## SPIN states enriched caRNA sequence features

Processing of iMARGI data was performed with iMARGI-Docker[77]. To quantify enrichment of repetitive element (RE)-containing caRNAs with specific chromatin states, we computed an enrichment score ($\log_2$-transformed observed/expected interaction frequencies) for each RE caRNA class and SPIN state. Observed frequencies were derived from the number of iMARGI read pairs with RNA ends mapped to RE class of interest and DNA ends mapped to SPIN states. Expected frequencies are computed as the total number of iMARGI read pairs multiplied by the product of the marginal probabilities of RE class abundance (the proportion of all caRNAs mapping to each RE class, irrespective of DNA mapping locations) and SPIN state abundance (the proportion of DNA reads mapping to each SPIN state). Only interchromosomal iMARGI pairs were analysed to mitigate potential biases from nascent RNA transcripts interacting with proximal genomic regions.

## Measuring nuclear RNA with iMARGI

iMARGI RNA end coverage is derived from RNA, DNA interactions represented in bedpe files generated by iMARGI docker[77]. The RNA end abundance bigwig file is generated by calculating the pileup reads coverage (R, coverage function) on the genome using RNA ends only in a strand-specific manner.

## Methods for integrated genome structure modelling

Data preprocessing was performed as described previously[81], with the exception of the parameter $f_{\text{maxation}} = 16$ for Hi-C preprocessing.

## Genome representation

The genome is represented at 200-kbp resolution as described previously[81,83], resulting in $n = 29{,}838$ chromatin regions, modelled as hard spheres of radius $r_0 = 118$ nm. The HFFc6 nucleus is modelled as an ellipsoid of semiaxes $(a,b,c) = (7{,}840, 6{,}480, 2{,}450)$ nm, whereas the H1 cell envelope is represented by a sphere of radius $R = 5{,}000$ nm.

We define the population of single-cell genome structures as a set of $S$ diploid genome structures $X = \{X_1, \ldots, X_s\}$; A genome structure $X_s$ is a set of three-dimensional vectors $X_s = \{\vec{x}_{is} : \vec{x}_{is} \in R^3, i = 1,2, \ldots, N\}$ representing the centre coordinates of each chromatin domain within the structure $s$:, $N$ being the total number of chromatin domains in the genome and $\vec{x}_{is} = (x_{is}, y_{is}, z_{is}) \in R^3$ indicates the coordinates of a 200,000-bp genomic region $i$ in structure $s$. We use the notation $I = (i, i')$ to indicate the genomic region, where $i$ and $i'$ represent the two alleles of genomic region $I$.

## Data-driven simulation of genome structures

Genome structure populations were generated with IGM following procedures described previously[81]. The goal is to determine a population of 1,000 diploid 3D genome structures $X$ statistically consistent with all input data from different available genomics experiments. Given a collection of input data $D$ from different data sources, $D = \{D_k | k = 1, \ldots, 3\}$ (here, ensemble Hi-C, lamina DamID and SPRITE; see data in Supplementary Note 1, section 4 'Methods for integrated genome structure modeling'), we aim to estimate the structure population $X$ such that the likelihood $P(\{D_k\}|X)$ is maximized. To represent missing information at the single-cell and homologous chromosome level, we introduce data indicator tensors $D^* = \{D_k^* | k = 1, \ldots, K = 3\}$, which augment missing information about allelic copies in single cells. Thus, the latent variables are a detailed expansion of the diploid and single-structure representation.

To determine a population of 3D genome structures consistent with all experimental data, we formulate a hard expectation–maximization problem, where we jointly optimize all genome structure coordinates and all latent variables. Given $\{D_k\}$, we aim to estimate the structure population $X$ and latent indicator variables $\{D_k^*\}$ such that the likelihood $P(X)$ $P(X)$ is maximized. We therefore aim to find the optimal structures and the optimal latent variables which satisfy: $\hat{X}, \hat{\mathfrak{D}}^* = \underset{X, \mathcal{D}^*}{\arg\max}\, P(\mathfrak{D}, \mathfrak{D}^* | X) = \underset{X, \mathcal{D}^*}{\arg\max} \prod_k P(\mathcal{D}_k | \mathcal{D}_k^*, X) \cdot \prod_k P(\mathcal{D}_k^* | X)$. This is a high dimensional, hard expectation–maximization problem and it is solved iteratively by implementing a series of optimization strategies for scalable and effective model estimation. Any iteration first optimizes the latent variables, by using the input data $\{D_k\}$ and the coordinates of all genomic regions $X^{(t)}$ from the previous iteration step, that is, $\mathfrak{D}^{*(t+1)} = \arg\max_{\mathfrak{D}} P(\mathfrak{D}|\mathfrak{D}^*, X^t) P(\mathfrak{D}^* | X^t)$. Then, coordinates of the genomic regions are optimized, based on the data deconvolution $D^{*(t)}$, that is, $X^{t+1} = \arg\max_X P(\mathfrak{D}|\mathfrak{D}^{*(t)}, X) P(\mathfrak{D}^{*(t)}|X)$, and additional constraints such as the volume confinement effect by the nuclear envelope, chromosomal chain connectivity and excluded volume. The process is iterated until convergence is reached. Overall, the data deconvolution process ensures that the structure population expresses the single-cell variability of genome organization, while also aggregately reproducing the ensemble behaviour (for example, the ensemble contact probabilities). Details on the probabilistic formulation underlying the optimization process and how that is designed and implemented for the different data sources can be found in a previous report[81] and its accompanying supporting information.

## Structural features

The population of 1,000 single-cell 3D genome structures was used to calculate a host of structural features $f$ that characterize the folding of each genomic region. All features and their cell-to-cell variability are calculated as described previously[81,83], unless otherwise noted.

## Variability

If applicable, cell-to-cell variability $\delta f_I$ of structural feature $f$ for a chromatin region $I$ (from chromosome $c$) is defined as: $\delta f_I = \frac{\sigma[f]_I}{\sigma[f]_c}$, $\sigma[f]_I$ indicating the s.d. of the feature value across the population and $\sigma[f]_c$ being the mean s.d. of the feature values of all regions within the same chromosome. $\delta f_I > 0 (<0))$ indicates high (low) variability of that feature at locus $I$.

RAD, the normalized average radial position, and $\delta RAD$, its cell-to-cell variability, is calculated as the normalized radial distance of a locus $I$ to the nuclear centre averaged over all structures in the population:

$$\text{RAD}_I = \frac{1}{S} \sum_s \frac{1}{2} \sum_{i \in I} r_{is}$$

$r_{is}^2 = \left(\frac{x_{is}}{a}\right)^2 + \left(\frac{y_{is}}{b}\right)^2 + \left(\frac{z_{is}}{c}\right)^2$ being the squared radial distance of locus $i$ in structure $s$, and $(a,b,c)$ being the nuclear semi-axes. The cell-to-cell variability of the radial position is defined $\delta RAD_I$.

## Local folding properties

These features encode local properties of the chromatin fibre and chromatin-chromatin interactions.

- Local chromatin fibre compaction ($R_g$) indicates the chromatin local compactness. If $R_g[i, s]$ indicates the radius of gyration of a 1 Mb chromatin segment centred at the locus of interest in structure $s$, then

$$Rg[I] = \frac{1}{S} \sum_S \frac{1}{2} \sum_{i \in I} R_g[i, s]$$

The compaction variability is denoted with $\delta Rg$.

- ICP indicates the average fraction of *trans* interactions out of all contacts formed by a genomic region $I$,

$$\text{ICP}[I] = \frac{1}{S} \sum_S \frac{1}{2} \sum_{i \in I} \frac{n_{i,\text{trans}}^s}{n_{i,\text{trans}}^s + n_{i,\text{cis}}^s}$$

$n_{i,\text{trans}}^s (n_{i,\text{cis}}^s)$ being the number of *trans* (*cis*) contacts formed by region $i$ in structure $s$.

- ILF indicates the fraction of structures in which a locus $I$ (either copy) occupies an inner position,

$$\text{ILF}[I] = \frac{1}{S} \sum_S \theta\left(\text{RAD}_i \le \frac{1}{2} \text{ or } \text{RAD}_{i'} \le \frac{1}{2}\right)$$

with $\theta$ being the Heaviside function

- Median trans AB ratio (transAB): for each chromatin region $i$ we define its trans A $n_{is,A}^t$ and B $n_{is,B}^t$ neighbourhoods as $n_{is,A}^t(n_{is,B}^t) = \#\{j : \text{chr}[i] \ne \text{chr}[j] \wedge |x_{is} - x_{js}| \le 500 \text{ nm} \wedge j \in A(B)\}$; $j \in A/B$ indicates locus $j$ is assigned to compartment A/B. The median trans AB ratio for that region across the population is computed by pooling the values from all homologues and structures,

$$\text{transAB}_I = \text{median}\left[\left\{\frac{n_{is,A}^t}{n_{is,B}^t}\right\}_{i \in I, s}\right]$$

The values are then rescaled so that $0 \le \text{transAB}_I \le 1$.

## Prediction of nuclear body locations using Markov clustering

A chromatin interaction network (CIN) is calculated for nuclear body-associated chromatin regions as described previously[83]. Speckle-associated chromatin regions are defined as the top 10% 200-kb regions with highest SON-TSA–seq signal; nucleolus-associated chromatin regions are 200-kb regions that overlap with nucleolus-organizing regions identified previously[149]. Spatial partitions of nuclear bodies are further calculated by the Markov clustering algorithm (MCL). Specifically, MCL clustering is performed for each nuclear body's CIN with mcl tool in the MCL-edge software[150]. Speckle locations are defined as the geometric centre of speckle partitions identified by MCL in speckle CINs. nucleolus locations are identified following the same protocol in nucleolus CINs. Only spatial partitions with size larger than three chromatin regions are considered for downstream analysis.

## Structure features defining the location of genomic regions with respect to nuclear bodies and compartments

SpD, NuD and LaD define the population-averaged distance of a genomic region to the nearest nuclear speckle, nucleolus or the nuclear envelope, respectively, while SAF, NAF and LAF quantify the frequency with which a genomic region is in association with a speckle, nucleus or the nuclear envelope in the population of cells. Approximate locations of nuclear speckles and nucleoli are predicted in each single-cell structure according to a procedure described previously[83]. Specifically, we identified locations of nuclear bodies in single cells by the geometric centres of highly connected subgraphs determined from a CIN, where each node represents a chromatin region with a high probability to be associated with a specific nuclear body and edges if their distances is smaller than an interaction cut-off. Details of the procedure are described previously[83].

- The average distance to the lamina (LaD) and its cell-to-cell variability ($\delta$LaD) is the (normalized) radial distance of a locus $I$ to the nuclear lamina averaged over the cell population:

$$\text{LaD}_I = \frac{1}{S} \sum_s \frac{1}{2} \sum_{i \in I} (1 - r_{is})$$

The cell-to-cell variability is defined by $\delta\text{LaD}_I$

- Average distances to closest speckle (SpD) and nucleolus (NuD) and their cell-to-cell variabilities ($\delta$SpD, $\delta$NuD)

$$\text{SpD}_I = \frac{1}{S} \sum_s \frac{1}{2} \sum_{i \in I} d_{is}^{\text{Sp}}, \text{NuD}_I = \frac{1}{S} \sum_s \frac{1}{2} \sum_{i \in I} d_{is}^{\text{Nu}}$$

where $d_{is}^{\text{Sp}}(d_{is}^{\text{Nu}})$ is the distance of genomic region $i$ to the predicted closest speckle (or nucleolus) in structure $s$. The related variability across the population is $\delta\text{SpD}_I$ ($\delta\text{NuD}_I$).

- For association frequencies with nuclear bodies (SAF, LAF, NAF), the SAF is defined as:

$$\text{SAF}_I = \frac{1}{S} \sum_s \frac{1}{2} \sum_{i \in I} \theta(d_{is}^{\text{Sp}} \le d_{\text{Sp}})$$

where $d_{\text{Sp}} = 500$ nm and $\theta$ is the Heaviside distribution. Analogous formulas are valid for association frequencies of genomic regions with the lamina (LAF) and nucleoli (NAF):

$$\text{LAF}_I = \frac{1}{S} \sum_s \frac{1}{2} \sum_{i \in I} \theta(d_{is}^{\text{Lam}} \le d_{\text{Lam}}), \text{NAF}_I = \frac{1}{S} \sum_s \frac{1}{2} \sum_{i \in I} \theta(d_{is}^{\text{Nu}} \le d_{\text{Nu}}),$$

with $d_{\text{Lam}} = 0.2$ rad and $d_{\text{Nu}} = 1,000$ nm.

## Calculation of enrichment scores for expression/genes/SPIN groups for structure features

All structure feature values are min–max normalized to scale the feature value to a 0–1 range. For structure features RAD, SpD and NuD, the normalized value is subtracted by 1 to signify lower value for closer proximity to the respective nuclear bodies.

To calculate the enrichment fold of a structure feature in a selected group (either selected based on gene expression, genes, SPIN and so on), we calculated the $\log_2$-transformed ratio of the feature mean within this group over the average mean value calculated from 100 random permutations of chromatin regions in the genome.

## Dimension reduction of structural features with $t$-SNE

Structure features RAD, LAF, TransAB, RG, SpD, ICP are normalized by $Z$-score and combined in feature vector for each genomic region. The $t$-SNE algorithm in the Python scikit-learn library is then applied to the structure feature vector of all chromatin regions with the following parameters: verbosity level of 1, perplexity of 80, maximum of 300 iterations for the optimization and random state seed of 123 to ensure reproducibility. The first two components are selected for 2D visualization.

## WTC-11 scHi-C data analysis

We used SnapHiC[100] to identify chromatin loops at 10-kb resolution with genomic distances of 100 kb to 1 Mb. In total, 5,390 chromatin loops were detected, and the loop strength in each single cell was quantified using normalized contact probabilities ($z$-scores). We then applied Higashi[98] to infer A/B compartments and TAD-like domain boundaries at 50-kb resolution. Finally, we performed integrative analyses of chromatin loops, compartments and TAD-like domains across the 188 WTC-11 single cells.

## Categorization of genes by expression levels

Gene-level expected counts were estimated using the RSEM package from alignment BAM files and normalized using the NOIseq R package into reads per kb per million mapped reads (RPKM) values. Among genes with non-zero RPKM values, the 25th percentile of RPKM values (expression Q1) and 75th percentile (expression Q3) of RPKM values were used as thresholds. Genes with RPKM > 0 and <Q1 were classified as lowly expressed, while those with RPKM > Q3 were classified as highly expressed.

## Categorization of class I and II genes

Class I and II genes were defined based on speckle association frequencies (SAFs). All 200-kb regions in the genome were divided into SAF quartiles. Class I genes are highly expressed genes in regions of the upper SAF quartile (that is, top 25%, above SAF Q3), and class II genes are highly expressed genes in regions of the bottom SAF quartile (that is, bottom 25%, SAF Q1).

## Average Hi-C contact decay profiles around TSS for class I and II genes

For class I and class I genes, each gene's annotated TSS was assigned to its corresponding 10 kb genomic bin in the Hi-C contact matrix

(cool file). For each TSS, we extracted contact counts between this bin and all 10-kb bins within ±2 Mb up/downstream to generate a per-gene contact count profile. Profiles are then aligned at the TSS and averaged across all genes in within the same class to produce mean contact frequency decay profiles for class I and class II genes.

### Spatial enhancer count per TSS
The spatial enhancer count for a given TSS is defined as the number of active enhancer regions within all 200 kb genomic loci (modelled as hard spheres with a 118-nm radius) whose geometric centres lie within a 350-nm distance of the TSS-containing region. Counts are stratified into all intrachromosomal, ultra-long-range intrachromosomal (>1 Mb) and interchromosomal regions.

### Calculated enhancer features
Calculated enhancer features in Supplementary Fig. 7 were as follows. E: the number of active enhancers within a 200-kb window where the gene TSS is located. E/G: the number of active enhancers within a 200-kb window normalized by the total number of transcription starts sites of genes. $E^N\_Inter/G$: the number of active enhancers from other chromosomes that are located within a spatial distance of 350 nm, normalized by the number of transcription starts sites of genes within the local 200-kb window. $E^N\_Intra/G$: the number of active enhancers within the same chromosome that are located within a spatial distance of 350 nm, normalized by the number of transcription starts sites of genes within the local 200-kb window. $E^N\_Intra(<2$ Mb$)/G$: the number of active enhancers within a 2-Mb sequence distance on the same chromosome that are located within a spatial distance of 350 nm, normalized by the number of transcription starts sites of genes within the local 200-kb window.

### Methods for single-cell 3D genome analysis
**Cell culture.** The modified WTC-11 (GM25236) hiPS cell line with GFP tagged AAVS1 locus (clone 6 and clone 28) was cultured following the 4D Nucleome-approved standard operating procedure (https://data.4dnucleome.org/protocols/d5889062-ec16-4246-9606-8d51f6b02dfa/) with two minor differences: (1) For penicillin–streptomycin, 15140-122 (Gibco) was used with a final concentration of 1% (v/v); (2) the density of seeding cells into six-well plate was 50,000–100,000.

**Generating scHi-C data.** The scHi-C libraries were prepared using methods previously described with slight modifications[151] In brief, 1–3 million cross-linked WTC-11 cells were first lysed and permeabilized by 0.5% SDS. The cells were then incubated overnight at 37 °C with 300 U MboI followed by proximity ligation with T4 ligase at room temperature with slow rotation for 4 h. Then the nuclei were stained with Hoechst and the single 2N nuclei were sorted by FACS into wells of 96-well plate. After overnight reverse crosslinking at 65 °C, the 3C-ligated DNA in each cell was amplified using the GenomiPhi v2 DNA amplification kit (GE Healthcare) for 4.5 h. After purification with AMPure XP magnetic beads and quantification, 10 ng WGA product was used to construct a library with Tn5. The detailed experimental procedures can be found in the 4D Nucleome portal (https://data.4dnucleome.org/protocols/3286b08d-d1d6-4853-a201-7dd08400d357/).

**The SBS polymer model of the studied *DPPA* locus.** To investigate at the single-molecule level the 3D folding of the *DPPA* locus (chromosome 3: 108.3 Mb–110.3 Mb) in WTC-11 pluripotent stem cells, we used the strings and binders (SBS) polymer model[93,95]. In the SBS, a chromatin region is represented as a self-avoiding polymer chain of beads, along which different types of binding sites are located for diffusing cognate molecular binders that can bridge them. The specific attractions between the binders and the polymer binding sites drive the folding of the system through thermodynamic mechanisms of polymer phase separation[94]. The model binding domains are determined by using the

PRISMR algorithm, which infers the optimal, that is, minimal, sets of different types of polymer binding sites by taking as input only bulk Hi-C contact data[139]. In our studied 2 Mb wide *DPPA* locus, PRISMR returned ten different binding domains. To derive a statistical ensemble of in silico *DPPA* single-molecule 3D conformations, we performed massive molecular dynamics (MD) simulations. In the MD implementation of the model, the system of polymer beads and binders is subject to a stochastic Langevin dynamics based on classical interaction potentials of polymer physics with standard parameters[94,152]. We ran the SBS simulations up to $10^8$ MD time iteration steps, when stationarity is fully reached. To ensure statistical robustness, we collected up to $10^3$ independent model conformations in the steady-state. We used the free available LAMMPS software (v.30july2016) to run MD simulations highly optimized for parallel computing[153].

### Methods for analysis of relation among A/B compartments, SPIN states, TADs/subTADs, loops and replication timing
**A/B compartment detection.** We called compartments in Hi-Cv2.5 data generated from H1 cells[42] (https://data.4dnucleome.org/files-processed/4DNFIOUDCJRH/) through eigenvector decomposition on each 25-kb chromosomal balanced matrix. We then normalized each matrix by an expected distance dependence mean counts value with removal of rows or columns with less than 2% non-zero counts coverage. We transformed to a z-score each off-diagonal count and a Pearson correlation matrix was computed. Subsequently, we performed eigenvector decomposition on the z-scored Pearson correlation matrix using LA.eig() (linalg package in numpy), selecting the eigenvector with the largest eigenvalue. We identified inflection points demarcating boundaries of compartments by genomic coordinates with a transition in eigenvector sign. We assigned compartments to either an A or B identity by collecting intervals of same eigenvector sign orientation (positive or negative) and counting total number of unique genes per direction then reassigning those with greater gene number intersection as A and the lesser as B.

**TAD and subTAD detection.** We called TADs and subTADs as previously reported[107,108,154,155] in 10-kb binned Hi-Cv2.5 data generated from H1 cells (https://data.4dnucleome.org/files-processed/4DNFI82R42AD/) using 3DNetMod (https://bitbucket.org/creminslab/cremins_lab_tadsubtad_calling_pipeline_11_6_2021), as previously described (https://data.4dnucleome.org/files-processed/4DNFIR94OF6S/).

**Dot detection.** We used dots indicative of loops called in Hi-C data from H1 cells (https://data.4dnucleome.org/files-processed/4DNFIEEF14ST/) as recently described using our published methods (https://bitbucket.org/creminslab/cremins_lab_loop_calling_pipeline_11_6_2021/src/initial/)[108,155]. Using geometric donut-shaped, lower-left, vertical and horizontal filters (parameters: $p = 2$ bins, $w = 10$ bins), we compute an expected interaction frequency for every given bin–bin pair. We computed P values for each bin–bin pair using a Poisson distribution and corrected for multiple testing using the Benjamini–Hochberg procedure. The final clusters were identified using dynamic FDR thresholding.

### SPIN-centric intersection with A/B compartments
We stratified H1 cell SPIN states into compartment A or B if they either co-register with a Jaccard index of greater than 0.70 or are embedded within a compartment. All other SPINs partially overlapping compartments were assigned into an 'other' category.

### TAD/subTAD-centric intersection with SPIN states
We used H1 cell dot and dotless TADs and subTADs previously described (https://data.4dnucleome.org/files-processed/4DNFIW5EIIO2/ and https://data.4dnucleome.org/files-processed/4DNFIU7GTTMW/)[108]. We classified dot TADs and subTADs as those in which loops intersect the

midpoint (that is, apex of the TAD/subTAD triangle) ±20% the size of the domain. All of the remaining TADs and subTADs were assigned as dotless. We then intersected dot and dotless TADs and subTADs with SPIN states. All those with a Jaccard index of greater than 0.70 were stratified as co-registering or residing within a SPIN state. We then further stratified dot and dotless TADs co-registered/embedded within SPINs into those that were further co-registered/embedded within A or B compartments with Jaccard index of greater than 0.70. All TADs and subTADs not co-registered/embedded in SPINs nor co-registered/embedded in compartments were assigned into an 'other' category.

## RNA-seq and iMARGI

We used RNA-seq (https://www.encodeproject.org/experiments/ENCSR537BCG/) and iMARGI (https://data.4dnucleome.org/files-processed/4DNFI2LXIREI/ and https://data.4dnucleome.org/files-processed/4DNFIOYVWEYZ/) data generated in H1 cells. iMARGI's RNA-end reads represent nuclear transcripts[78]. The RNA-end and DNA-end reads were processed with iMARGI-Docker[77] and mapped to the human hg38 reference genome and processed as described. See data availability through the 4D Nucleome portal below.

## High-resolution 16-fraction Repli-seq

We used 16-fraction Repli-seq data from H1 cells (https://data.4dnucleome.org/experiment-sets/4DNESXRBILXJ/) processed as described into normalized and scaled data arrays (https://data.4dnucleome.org/files-processed/4DNFI3N8GHKR/)[108]. The Repli-seq experiment was performed using an antibody against BrdU (anti-BrdU BD 555627, 1:100). We identified IZs (https://data.4dnucleome.org/files-processed/4DNFIRF7WZ3H/) as described previously[108]. Raw counts in each fraction ($S_{i,j}$) were normalized by sequencing depth by virtue of read per million (RPM) such that $S_{norm,j,50kb\_bin} = S_{j,50kb\_bin}/S_{i,j} \times 10^6$. Repli-seq arrays were subsequently constructed from RPM bedgraphs to form 16 rows with each row representing an S phase fraction and each column representing a 50-kb bin. The array was smoothed by applying a Gaussian filter and scaled such that each column sums to 100.

## Data visualizations

We visualized h1ESCv2.5 Hi-C (https://data.4dnucleome.org/files-processed/4DNFI82R42AD/) counts at SPIN state calls or dot and dotless TADs and subTADs by adding 60% of the size of the domain or SPIN to the edges of the maps and stretching to a defined length $L$. Each domain or SPIN was used once in the visualization and the counts in every pixel were normalized to the mean distance dependence expected value and then averaged across all 2D matrices. We similarly visualized high-resolution 16-fraction Repli-seq, total RNA-seq, iMARGI and compartment eigenvectors resized to the same intervals as TADs, subTADs or SPIN states. The signal for high-resolution 16-fraction Repli-seq is the average pileup; for total RNA-seq and iMARGI, the signal is the median pileup of averaged plus and minus strands. The compartment eigenvector is the mean pileup signal. We resized to defined length $L$ with the resize() method in OpenCV image package (https://pypi.org/project/opencv-python/).

## IZ resampling test

We computed the proportion of early and late IZs intersecting with dot TAD/subTAD boundaries (https://data.4dnucleome.org/files-processed/4DNFIWNJ5RR7/), dotless boundaries (https://data.4dnucleome.org/files-processed/4DNFIT6QE9YU/) and no boundaries. To create a null set of IZs, we computationally sampled the genome for random intervals matched by number, size and A/B compartment distribution of real early or late IZs. We created a null distribution by sampling $1.5 \times 10^8$ times and computing a one-tailed empirical $P$ value as the area under the null distribution to the right of the real IZs. We used only null and real IZ sets in autosomal regions with sufficient counts for the statistical test by filtering unmappable telomeric/centromeric regions.

## Chromatin dataset processing

CUT&Run datasets were processed by trimming adaptors using cutadapt, locally mapping the reads using bowtie2, filtering for quality, removing duplicates and ENCODE blacklisted regions (ENCFF419RSJ) using samtools, and computing the coverage using deeptools. The average chromatin landscape at IZs was computed using HOMER on a 1 Mb region centred on each IZ centre and plotted using R. The chromatin profile was plotted using the WashU browser and IGV on chromosome 2: 20404583–58108703.

## IZ integration with chromatin and transcription

HCT116, H1 and mES cell IZs were grouped by replication timing by splitting the corresponding Repli-seq data into quartiles. Matching random regions were generated by shuffling the IZs regions within their respective replication timing quartile using bedtools. Insulation score at IZs and random regions were computed by extracting the minimum insulation score in each IZ or random region using bedtools. Accessibility at IZs and random region was computed by extracting the total ATAC–seq signal at each IZ or random region using bedtools. H1 cell RNA-seq data were processed by mapping reads on hg38 using Hisat2, filtering for quality using samtools and computing the coverage using bedtools. Expression at IZs and random regions was computed by extracting the total RNA-seq coverage (adding plus and minus strands for the H1 cell RNA-seq) at each IZ or random region using bedtools. The H1 cell CUT&Run chromatin marks dataset was processed by trimming adaptors using cutadapt, locally mapping the reads using bowtie2, filtering for quality, removing duplicates and ENCODE blacklisted regions using samtools and computing the coverage using deeptools. The average chromatin marks at IZs and a random region were computed using HOMER on a 1 Mb region centred on each IZ centre. Figures were plotted using R.

## Methods for predicting Hi-C maps from sequence (Akita)

Two convolutional neural network models with the Akita architecture[124] were trained, one on H1 cells and one on HFFc6 Micro-C data. Micro-C data were downloaded from the 4D Nucleome data portal[156] and processed as previously described[124]. The coordinates of the deletion at the *TAL1* locus with experimentally verified changes on genome folding was obtained from a previous study[157]. The deletion was centred and systematically extended on both sides to get the input sequence for Akita ($2^{20}$ bp).

FIMO[158] was used to scan the human genome (hg38) to identify the potential binding sites ($P < 1 \times 10^{-5}$) of cell-type-specific TFs POU2F1–SOX2 (MA1962.1) and FOSL1–JUND (MA1142.1) using their annotations in the JASPAR database[159]. TAD boundaries in the Micro-C datasets at 5-kb resolution were downloaded from the 4D Nucleome data portal. The ones that were not shared between the cell types (no boundary in the other cell type within 20 kb) were identified as cell-type-specific TAD boundaries. The binding sites overlapping cell-type-specific TAD boundaries of each cell type were extracted using bedtools[160]. In silico mutagenesis was performed on the resulting binding sites by replacing the motifs with random sequences to evaluate their effects on predicted genome folding. Motif logos were visualized using the contribution scores of the motifs, which were computed by DeepExplainer (DeepSHAP implementation of DeepLIFT)[128,161].

## Reporting summary

Further information on research design is available in the Nature Portfolio Reporting Summary linked to this article.

## Data availability

This study analysed data publicly available through the 4D Nucleome data portal (https://data.4dnucleome.org; https://data.4dnucleome.org/joint-analysis) and the ENCODE data portal (https://screen.

encodeproject.org/; https://www.encodeproject.org) (Supplementary Note 1). We refer to the Supplementary Information for a full list of all data accession numbers and public links. This study produced three new datasets that are publicly available: loop calls in H1-hESCs and HFFc6 cells (https://doi.org/10.5281/zenodo.17451616 (ref. 162); https://data.4dnucleome.org/files-processed/4DNFIX6VZKOA/; https://data.4dnucleome.org/files-processed/4DNFI32J1C6W/), SPIN state annotations for H1-hESCs and HFFc6 cells (https://data.4dnucleome.org/joint-analysis#spin-states; https://data.4dnucleome.org/files-processed/4DNFITDBR2LR/; https://data.4dnucleome.org/files-processed/4DNFI22FB8HD/) and 3D models and 3D structure feature profiles (https://zenodo.org/records/17459402 (ref. 163); https://data.4dnucleome.org/files-processed/4DNFIJ36KR6X/; https://data.4dnucleome.org/ files-processed/4DNFILC1YWO6/; https://data.4dnucleome.org/ files-processed/4DNFIHF82NAI/; https://data.4dnucleome.org/ files-processed/4DNFITOP9BQE/).

## Code availability

All code applied in this paper is publicly available through the following links: methods benchmarking (https://doi.org/10.5281/zenodo.17475348)[164]; GAM analysis (https://doi.org/10.5281/zenodo.17477229)[165]; loop calling and transcription-loop analysis (https://doi.org/10.5281/zenodo.17456776)[166]; SPIN analysis (https://doi.org/10.5281/zenodo.17469228)[167]; iMargi analysis (https://doi.org/10.5281/zenodo.17502239)[168]; 3D genome structure modelling: Integrative Genome Modelling (IGMv2.0) platform[169] (https://doi.org/10.5281/zenodo.17478448; www.github.com/alberlab/igm); single-cell analysis (https://doi.org/10.5281/zenodo.17477302)[170]; TADs, SPIN, compartment and replication analysis (https://doi.org/10.5281/zenodo.17467184)[171]; replication timing analysis (https://doi.org/10.5281/zenodo.17485344)[172]; and predicting Hi-C data from sequence (https://doi.org/10.5281/zenodo.17469981)[173].

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

**Acknowledgements** This work has been collectively funded by grants from the National Institutes of Health Common Fund (The 4D Nucleome Project): U01HL157989, U54DK107965, UM1HG011593, U54DK107981, U01CA200059, U54DK107977, UM1HG011585, U01DA052769, U01CA200147, U01CA200060, U01DA040612, U01DK127420, U01HL130007, U54DK107967, U54DK107979, UM1HG011531, U54DK107980, HG011536, U01HL129998, U01DA052715 and U01DK127405.

**Author contributions** This work is the result of a collaboration among all authors as their contribution to the 4D Nucleome Consortium. Project coordinators: J.D., F.Y. and W.S.N. Topic leaders: J.D., F.Y., W.S.N., X. Wang, J.M., F.A., M.N., B.R., S.Z., J.E.P.-C., D.M.G. and K.S.P. Manuscript writing and editing: J.D., F.Y., W.S.N., J.M., B.R., T.W., Y. Li, A.S.B., K.S.P., F.A., M.N., S.K., Y. Zhang, Ruochi Zhang, X. Wen, S.Z. and X. Wang. Data production (execution of data production experiments, experiment coordination and supervision): J.D., L.Y., J.H.G., N.K., O.J.R., J.X., D.H.J., S.H., A.K., A.W., W.W.-N., R.K., B.R., M.Y., P.K., L.Z., A.S.B., D.M.G., T. Sasaki, T.v.S., L. Brueckner, D.P.-H., B.v.S., S.Z., P.W., H.C., Y.R., Ran Zhang, W.S.N., S.A.Q., M.G. M.K. and A.P. Methods benchmarking: J.D., F.Y., B.A.O., X. Wang, J.H.G., S.V.V., D.P., M.N., A.K., I.I.A., D.S., C.J.T., T. Szczepińska and A.P. Annotation of spatial chromatin compartmentalization through integrative modelling: Y. Zhang, Ran Zhang, M. Chiliński, K.S., A.Y., M. Conte, A.E., A.A., Ruochi Zhang, Yuchuan Wang, A.S.B., W.S.N., X. Wen, Q.W., W.Z., S.C., Y.Y., S.Z., J.L., D.P., M.N., F.A. and J.M. 3D modelling: Ye Wang, A.Y., L. Boninsegna, A.M.C., S.B., Y. Zhan, M. Chiliński, K.S., M. Conte, A.E., A.A., Y. Zhang, D.P., M.N., J.M. and F.A. Single-cell 3D genome analysis: Ruochi Zhang, Ran Zhang, M. Conte, A.E., A.A., M.Y., L.L., M.H., B.R., M.N., Y. Zhang and J.M. 3D genome and genome function X. Wang, Ye Wang, D.J.E., R.J.B., M.K.M, A.L.C., J.E.P.-C., C.M., P.Z., D.M.G., S.Z., X. Wen, Y. Liu, Y. Zhang and J.M. Predicting chromosome folding from sequence: S.K. and K.S.P. Data coordination and visualization: P.J.P., B.H.A., A.J.S., R.N., C.B., W.R., S.E., A.J.V., N.G., T.W., D.L., Y. Zhang and J.M.

**Competing interests** J.D. is a member of the scientific advisory board of Arima Genomics and Omega Therapeutic. S.Z. is a founder and shareholder of Genemo and Neurospan. B.R. has equity in Arima Genomics and Epigenome Technologies. C.M. is the director and founder of In silichrom.

**Additional information**
**Correspondence and requests for materials** should be addressed to Job Dekker, Xiaotao Wang, Mario Nicodemi, Bing Ren, Sheng Zhong, Jennifer E. Phillips-Cremins, David M. Gilbert, Katherine S. Pollard, Frank Alber, Jian Ma, William S. Noble or Feng Yue.

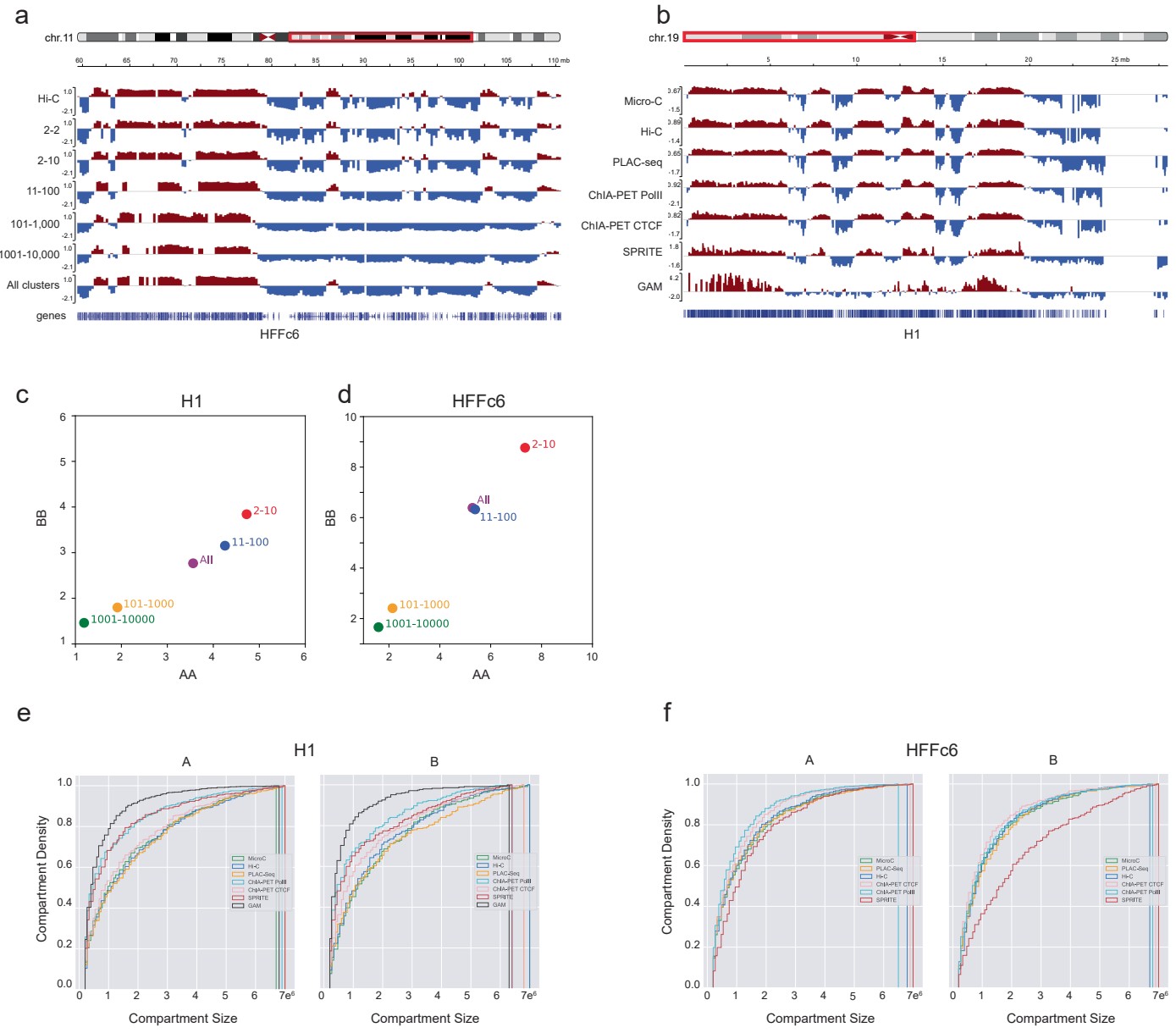

**Extended Data Fig. 1 | Chromatin interaction assays quantitatively differ in detection of small compartment domains. a**. Eigenvector 1 obtained from SPRITE data derived from a range of cluster sizes along a typical genomic loci, showing that small compartments are not detected when data from larger SPRITE clusters is used. **b**. Examples of Eigenvector 1 profiles obtained from data generated with the different genomic assays indicated. **c**. Compartmentalization strength calculated with interaction data obtained with different SPRITE clusters, for H1 and HFFc6. **e,f**. Cumulative distributions of compartment sizes, as detected with Hi-C, Micro-C, ChIA PET, PLAC Seq, and SPRITE (cluster size 2–100) for H1 and HFFc6 cells.

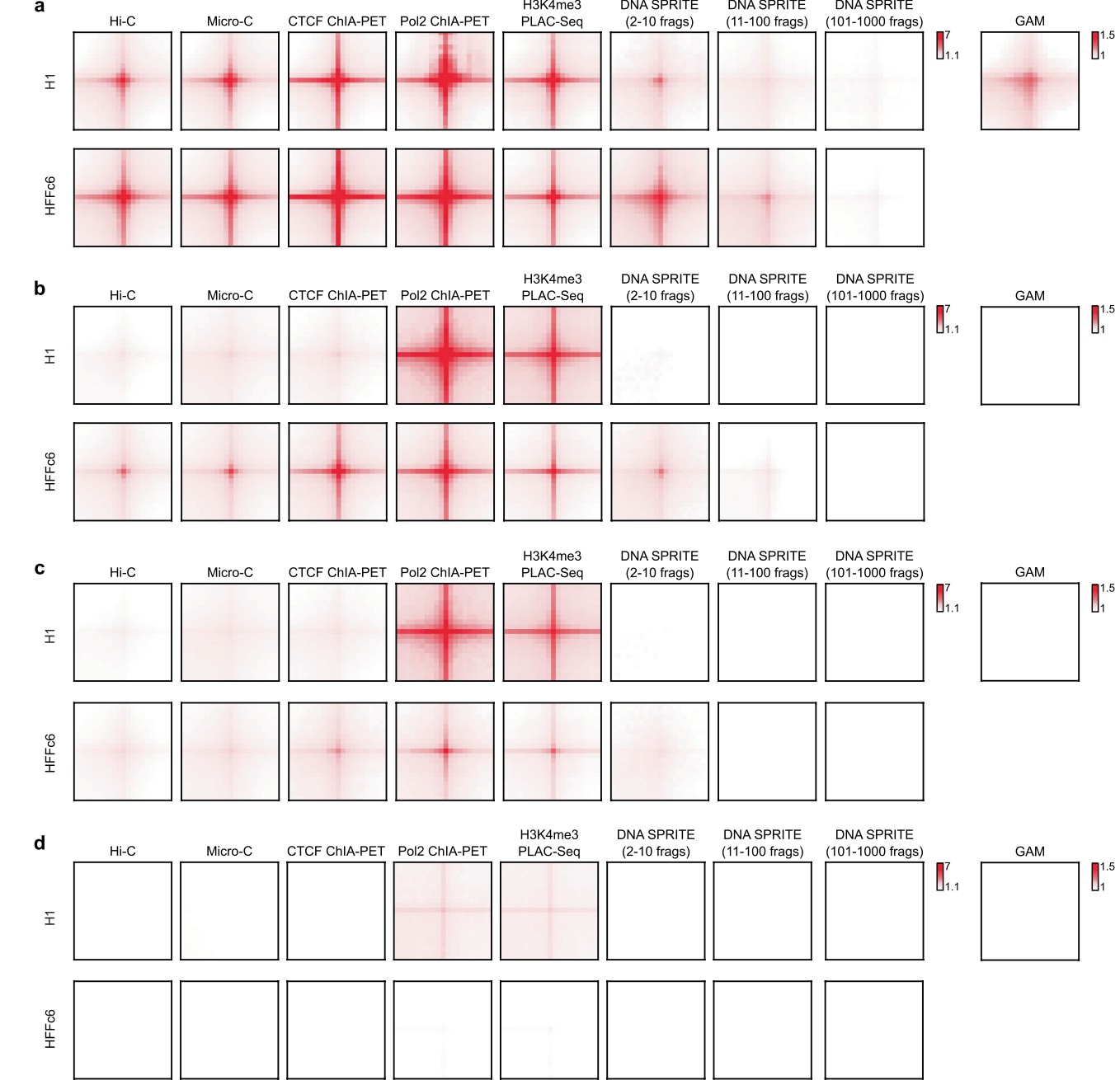

**Extended Data Fig. 2 | Aggregate peak analysis of different chromatin loop sets across platforms.** Contact maps at 25-kb resolution were used. **a**. High-confidence loops detected by at least two platforms in each cell line. **b**. Pol2 ChIA-PET loops in each cell line. **c**. Loops uniquely detected by Pol2 ChIA-PET. **d**. Random control loops for Pol2 ChIA-PET-unique loops. For each Pol2 ChIA-PET-unique loop, control loops are defined by considering interactions between each anchor and loci positioned at the same genomic distance on the opposite side.

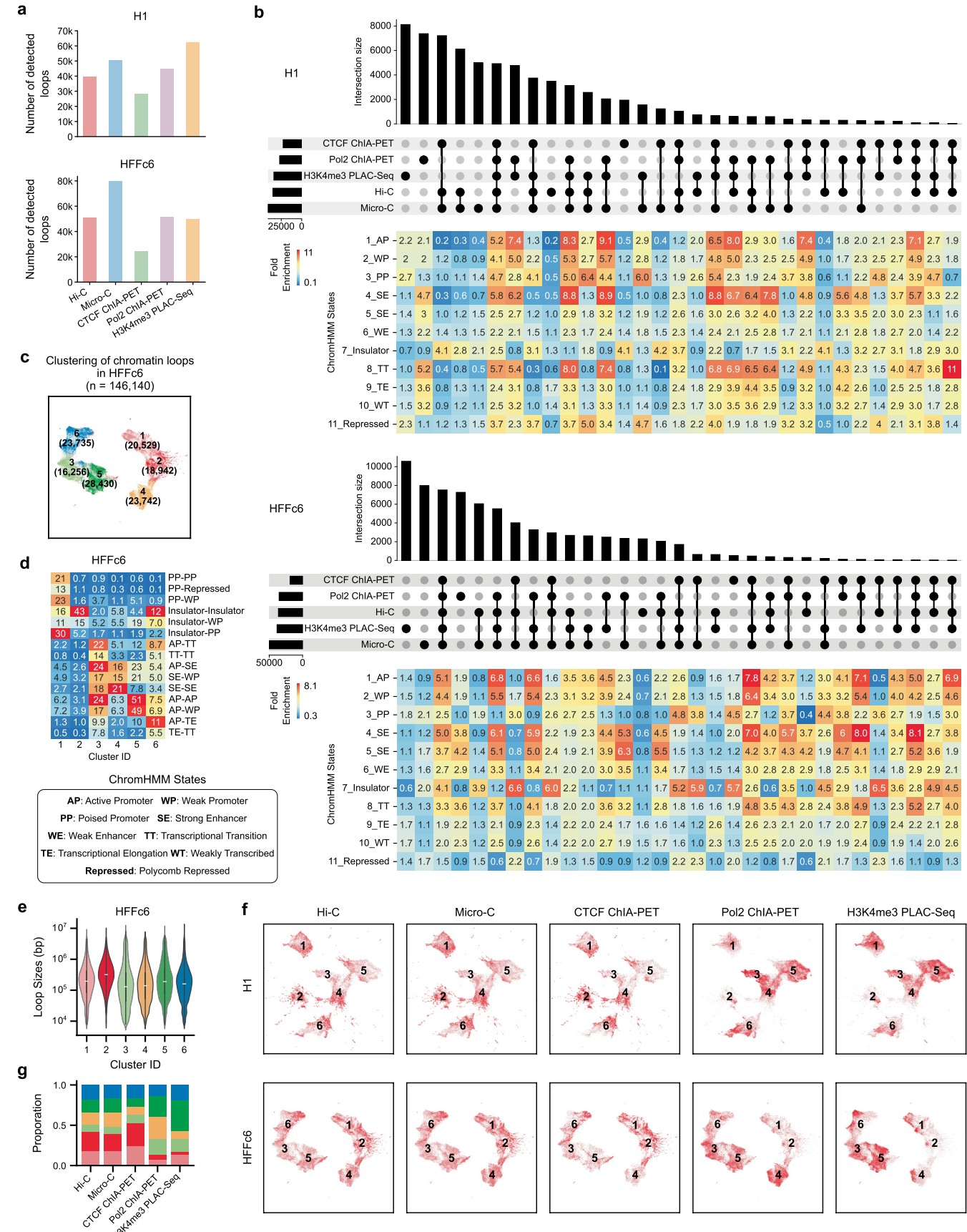

**Extended Data Fig. 3** | See next page for caption.

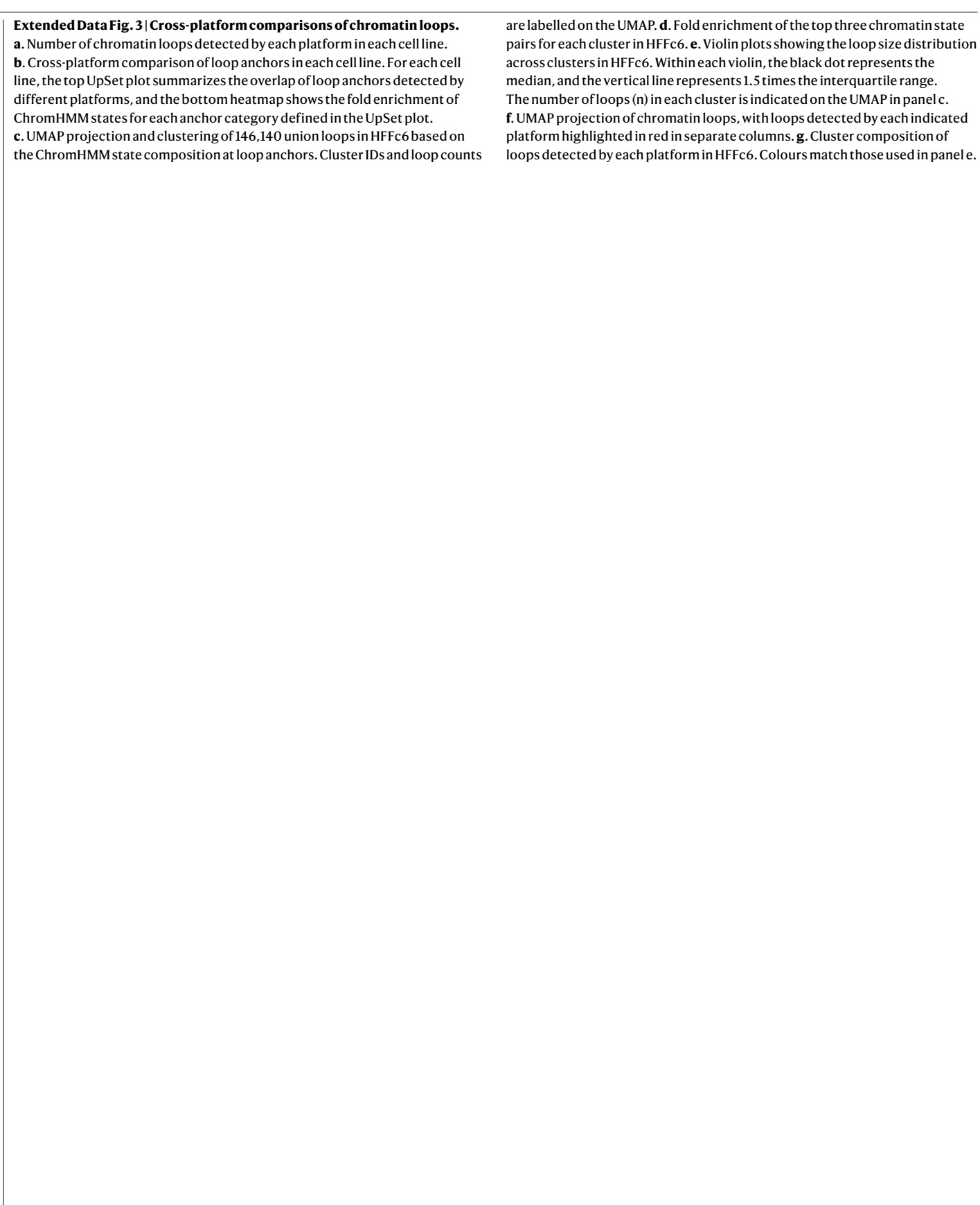

**Extended Data Fig. 3 | Cross-platform comparisons of chromatin loops.**
**a**. Number of chromatin loops detected by each platform in each cell line.
**b**. Cross-platform comparison of loop anchors in each cell line. For each cell line, the top UpSet plot summarizes the overlap of loop anchors detected by different platforms, and the bottom heatmap shows the fold enrichment of ChromHMM states for each anchor category defined in the UpSet plot.
**c**. UMAP projection and clustering of 146,140 union loops in HFFc6 based on the ChromHMM state composition at loop anchors. Cluster IDs and loop counts are labelled on the UMAP. **d**. Fold enrichment of the top three chromatin state pairs for each cluster in HFFc6. **e**. Violin plots showing the loop size distribution across clusters in HFFc6. Within each violin, the black dot represents the median, and the vertical line represents 1.5 times the interquartile range. The number of loops (n) in each cluster is indicated on the UMAP in panel c.
**f**. UMAP projection of chromatin loops, with loops detected by each indicated platform highlighted in red in separate columns. **g**. Cluster composition of loops detected by each platform in HFFc6. Colours match those used in panel e.

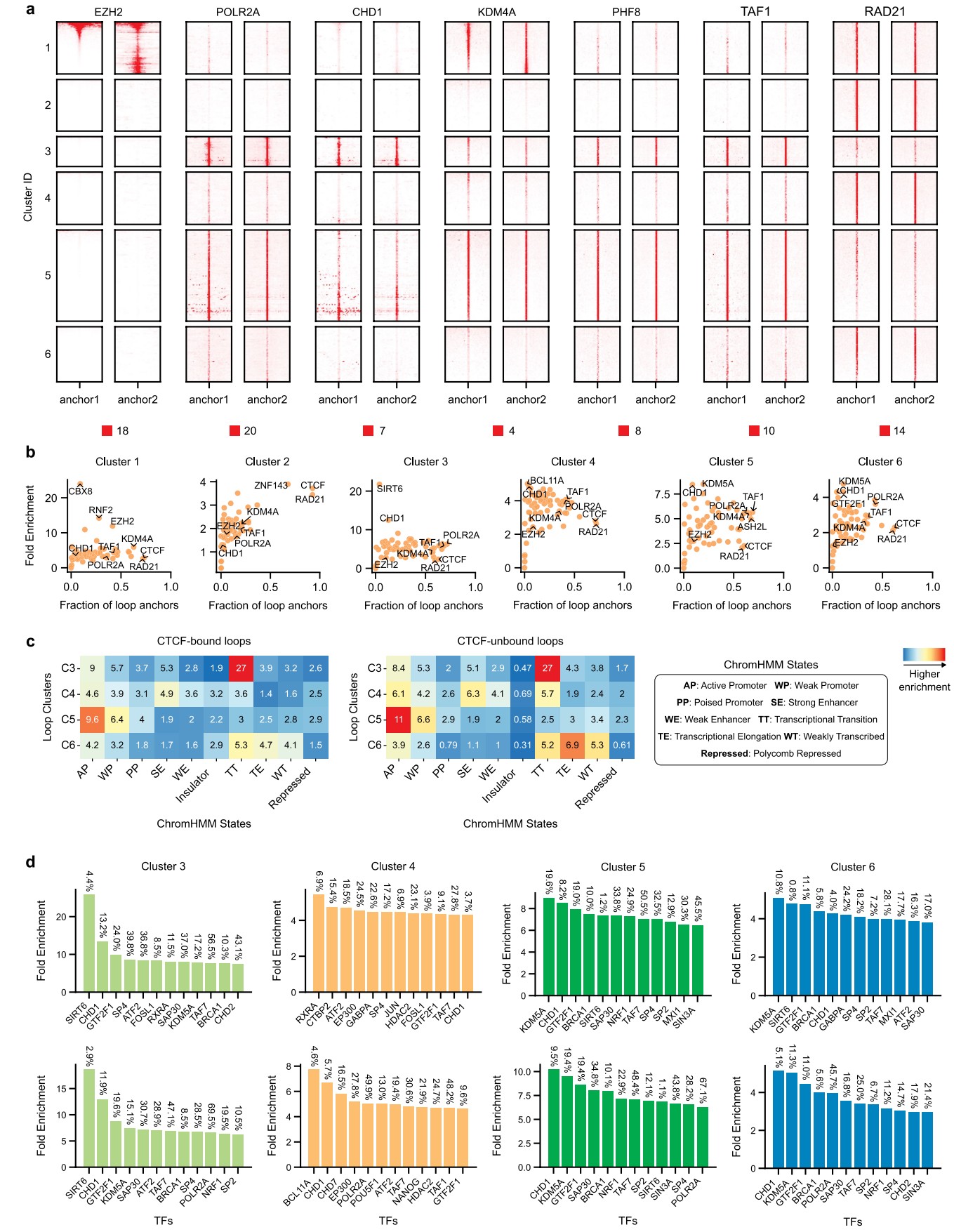

**Extended Data Fig. 4** | See next page for caption.

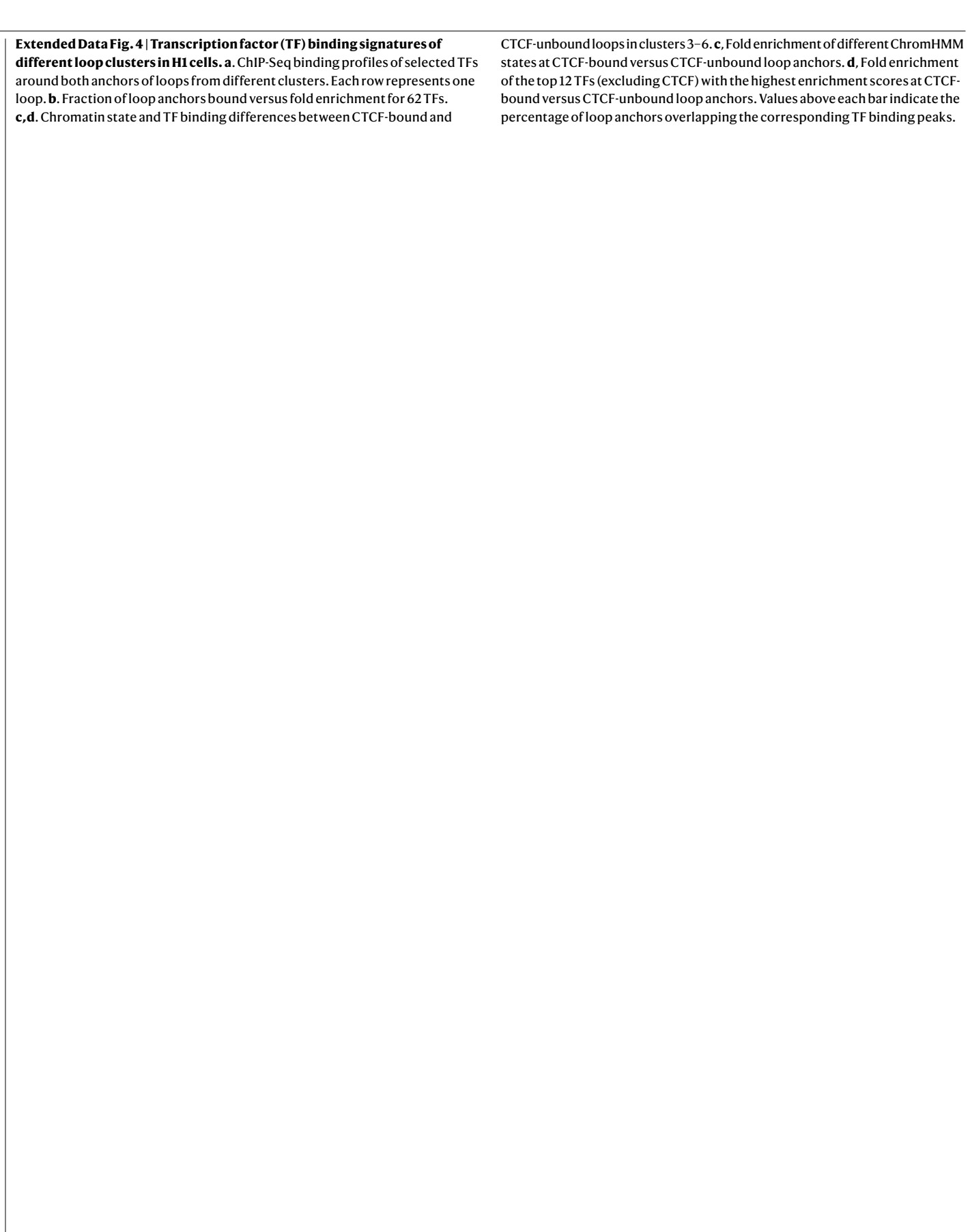

**Extended Data Fig. 4 | Transcription factor (TF) binding signatures of different loop clusters in H1 cells. a**. ChIP-Seq binding profiles of selected TFs around both anchors of loops from different clusters. Each row represents one loop. **b**. Fraction of loop anchors bound versus fold enrichment for 62 TFs. **c,d**. Chromatin state and TF binding differences between CTCF-bound and CTCF-unbound loops in clusters 3–6. **c**, Fold enrichment of different ChromHMM states at CTCF-bound versus CTCF-unbound loop anchors. **d**, Fold enrichment of the top 12 TFs (excluding CTCF) with the highest enrichment scores at CTCF-bound versus CTCF-unbound loop anchors. Values above each bar indicate the percentage of loop anchors overlapping the corresponding TF binding peaks.

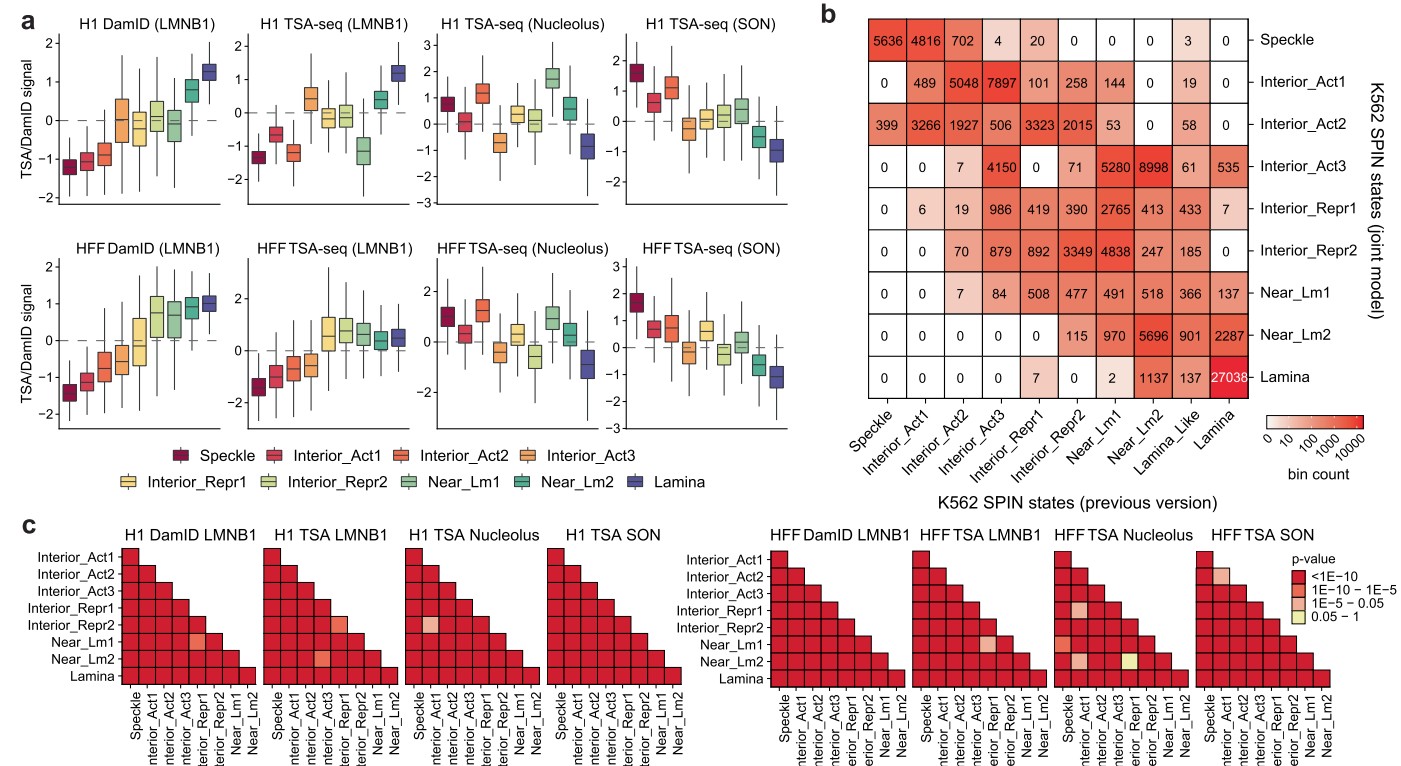

**Extended Data Fig. 5 | a.** Box plots show the distribution of normalized TSA-seq and DamID scores on distinct SPIN states in H1 and HFFc6 cell lines. The number of 25 kb bins in H1 (from left to right) is 11,073, 15,446, 10,002, 18,327, 8,771, 5,129, 6,705, 81,96, and 29,226, respectively. The number of 25 kb bins in HFFc6 (from left to right) is 9,693, 14,111, 9,032, 15,749, 9,018, 18,411, 8,249, 7,900, and 20,712, respectively. Box plot shows median (centre line), interquartile range (IQR; box), and whiskers extending 1.5×IQR. **b.** A confusion matrix shows the comparison of SPIN states in K562 between the previous version and the new result based on the joint modelling across four cell lines. Note that the new result is based on a new nucleolus TSA-seq data. The numbers in the heatmap indicate the number of 25 kb bins. **c.** The differences of the distributions of normalized TSA-seq and DamID between any two pairs of SPIN states are tested by the two-sided Wilcoxon rank sum test. Colours of heatmap indicate the p-value under the null hypothesis that two distributions are derived from the same population. P-value is classified as p ≤ 1e-10, 1e-10 <p ≤ 1e-5, 1e-5 <p ≤ 0.05, or p > 0.05.

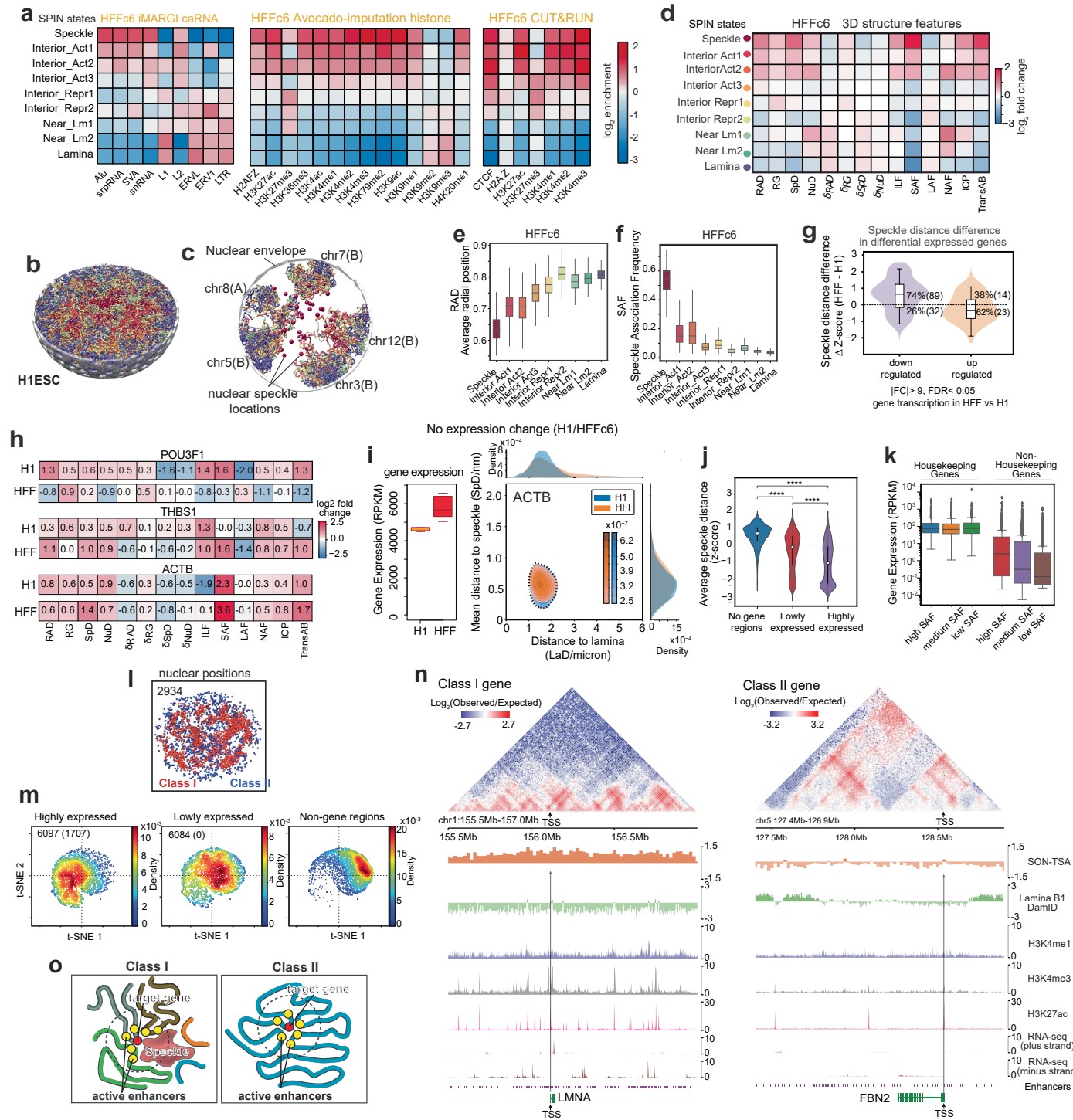

**Extended Data Fig. 6** | See next page for caption.

**Extended Data Fig. 6 | 3D Structure-function relationship and its cell-to-cell variability. a**. SPIN states define spatial genome compartments. Heatmaps show enrichment of caRNAs and histone marks (columns) across SPIN states (rows) in HFFc6. Fold-change is calculated as the ratio of observed signals over genome-wide expectation. **b**. Single-cell genome structure of H1 with regions coloured by SPIN state. **c**. Slice of the structure in (a) showing a subset of chromosomes and predicted speckle locations (red spheres). **d**. Log2-enrichment of structural features across SPIN states from model populations over genome-wide expectation. Structure features as defined in Fig. 4f. **e**. Box plots of average radial positions of chromatin by SPIN state in HFFc6. Box plots show median (centre line), interquartile range (IQR; box), and whiskers extending 1.5×IQR (n = 7 H1, 5 HFFc6). **f**. Box plots of speckle association frequencies (SAF) by SPIN state in HFFc6. Box plots as in (e). **g**. Violin plots of speckle distance z-score differences (HFFc6 - H1) for genes with significant changes in expression ($|\log_2$ fold-change$| > 9$, FDR < 0.05). The percentages and numbers in brackets indicate the fraction and number of genes with significantly increased (top) or decreased (bottom) speckle-distance z-scores in HFFc6 relative to H1 (FDR 0.05). Box plots within the violin show the median (centre line), interquartile range (box), and whiskers extending to 1.5X the interquartile range (n = 121, 37 for down-regulated and up-regulated genes). **h**. $\log_2$-fold enrichment of 14 structural features for *POU3F1*, *THBS1* and *ACTB* genes in H1 and HFFc6. **i**. (left) *ACTB* gene expression in H1 and HFFc6. (Right) Joint nearest speckle and lamina distance distribution for *ACTB* in 1,000 single-cell models.

Box plots show the median (centre line), interquartile range (box), and whiskers extending to 1.5X the interquartile range (n = 7 H1, 5 HFF). **j**. Speckle distance (SpD) distributions of genes across gene expression categories in HFFc6. Within each violin, the white dot represents the median, and the vertical black line represents interquartile range. Pairwise group differences were assessed using two-sided independent-samples t-tests (Student's t-test; no multiple-testing correction applied). Significance is indicated as ns (p > 0.05), * ($p \leq 0.05$), ** ($p \leq 0.01$), *** ($p \leq 0.001$), and **** ($p \leq 0.0001$). **k**. Gene expression distributions for housekeeping vs. non-housekeeping genes stratified by SAF. Box plots show the median (centre line), interquartile range (box), and whiskers extending to the minimum and maximum data points within 1.5X the interquartile range; individual points represent outliers. **l**. Representative HFFc6 genome structure with Class I (red) and Class II (blue) highly expressed genes. A representative single 3D genome structure of HFFc6 showing the locations of all highly expressed class I genes (red) and class II genes (blue). **m**. t-SNE of TSS-containing regions based on 3D structure features for 25% highest and lowest expressed genes and gene-less regions in HFFc6. Counts of total (and housekeeping) genes are indicated. **n**. Local genome architecture near TSS of Class I gene *LMNA* and Class II gene *FBN2*. Tracks (top to bottom): Hi-C (10 kb bins, KR norm. O/E), SON-TSA-seq, Lamina B1 DamID, H3K4me1, H3K4me3, H3K27ac, plus/minus strand RNA-seq. **o**. Schematic illustration explaining reduced long-range contact contributions for Class I vs. increased long-range interactions for Class II genes.

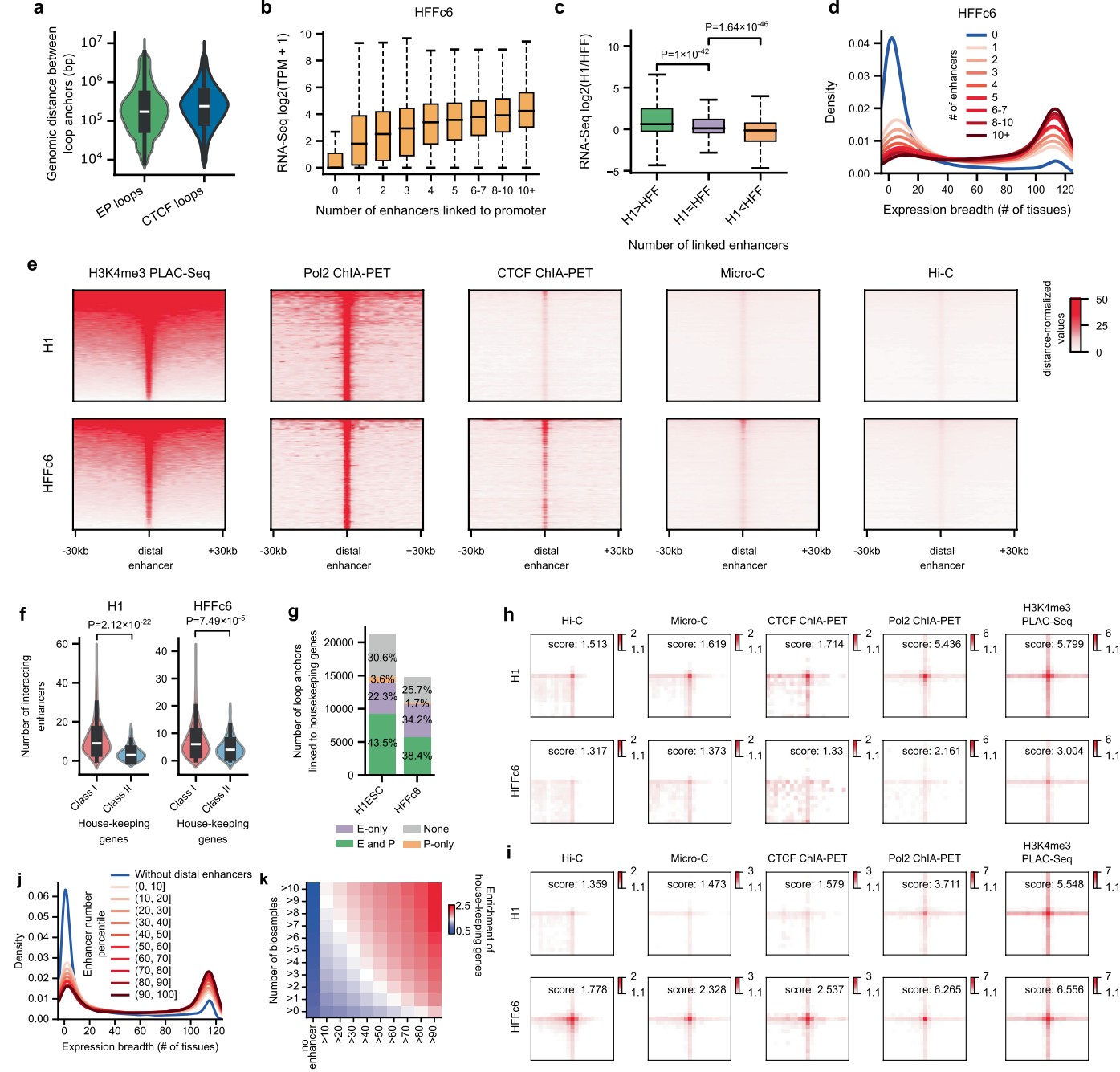

**Extended Data Fig. 7 | Housekeeping genes are engaged in extensive enhancer-promoter loops.** All boxplots in this figure follow the same format: the centre line indicates the median, the box limits represent the upper and lower quartiles, and the whiskers extend to 1.5 times the interquartile range. P-values were calculated using a two-sided Mann-Whitney U test. **a.** Size distributions of enhancer-promoter (EP) loops (n = 79,708) and CTCF-mediated loops (n = 99,447). Data from H1 and HFFc6 were combined. **b.** Gene transcription levels versus the number of interacting enhancers in HFFc6. The number of genes for each group (from left to right) is 6,846, 1,626, 1,505, 1,354, 1,276, 1,164, 1,833, 1,894, and 2,150, respectively. **c.** Fold change in transcription levels for genes with more enhancers in H1 (n = 7,109), the same number of enhancers in both cell types (n = 3,147), or more enhancers in HFFc6 (n = 4,535). **d.** Expression breadth of genes with different numbers of interacting enhancers in HFFc6. **e.** Distance-normalized contact signals between housekeeping gene promoters and ±30 kb regions surrounding their interacting enhancers. **f.** Comparison of enhancer counts between two housekeeping gene classes defined in the 3D modelling section. The number of enhancers in class I and class II genes is 1,262 and 86 in H1, and 1,259 and 79 in HFFc6, respectively. **g.** Number of loop anchors linked to housekeeping genes, categorized by whether they contain enhancers, promoters, both, or neither. **h,i.** Aggregate peak analysis of cell-type-specific loops connecting distal enhancers to housekeeping gene promoters across platforms at 5-kb resolution. Panels h and i show the results for loops uniquely detected in H1 and HFFc6, respectively. **j.** Expression breadth across 32 samples for genes grouped by enhancer count percentiles in each sample. **k.** Enrichment of housekeeping genes across gene sets defined by both the number of interacting enhancers and the number of supporting samples. Each bin represents a specific combination of these two factors. For example, the top-right bin corresponds to genes with more than the 90th percentile of interacting enhancers in over 10 samples.

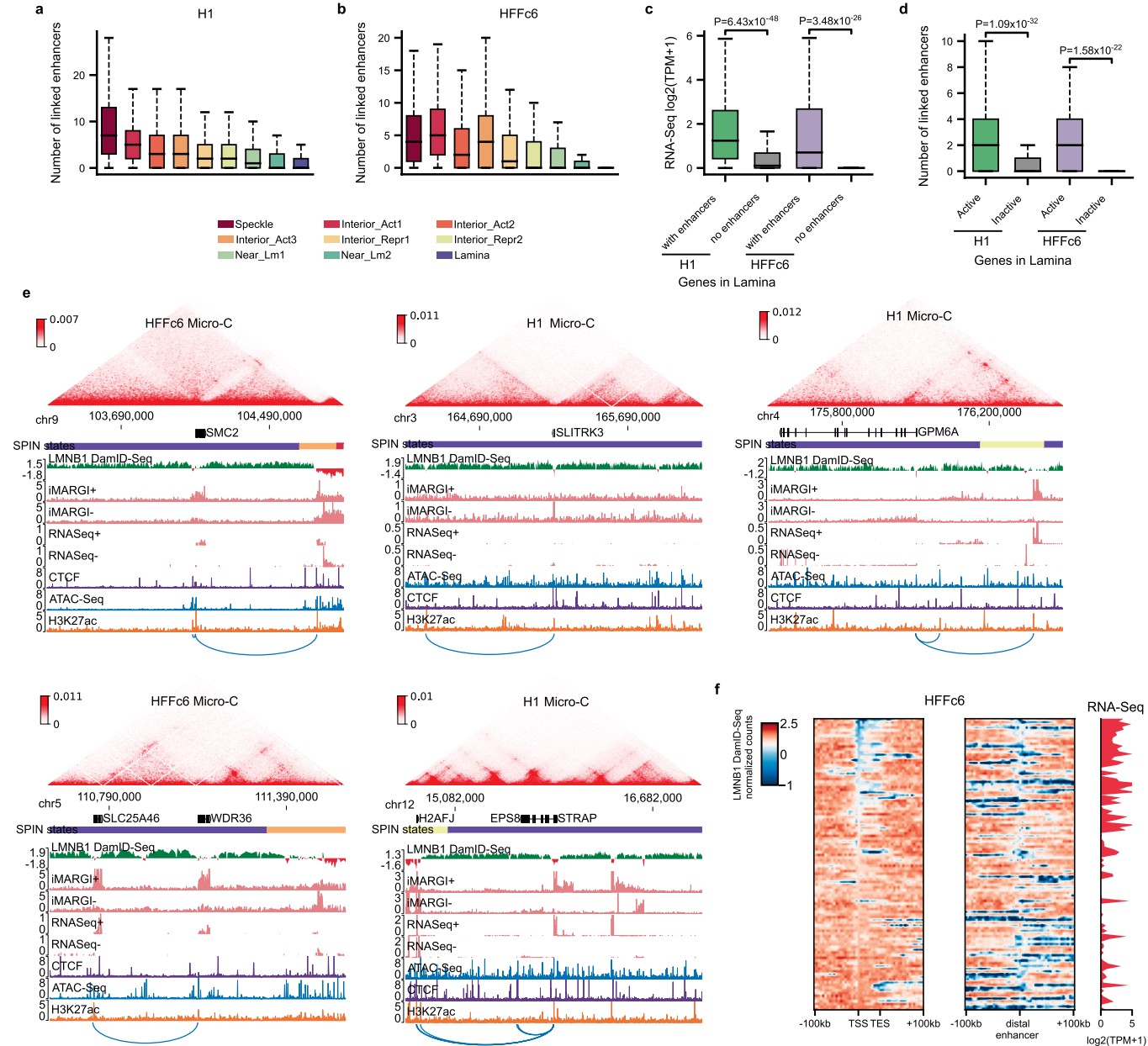

**Extended Data Fig. 8 | Enhancer-promoter loops within the nuclear lamina and their relationship to gene regulation.** All boxplots in this figure follow the same format: the centre line indicates the median, the box limits represent the upper and lower quartiles, and the whiskers extend to 1.5 times the interquartile range. P values were calculated using a two-sided Mann-Whitney U test. **a,b**. Distributions of the number of interacting enhancers for genes in different SPIN states in H1 (a) and HFFc6 (b). The number of genes in each group (from left to right) is 5,092, 2,626, 2,966, 1,769, 1,743, 2,450, 975, 928, and 965 for H1, and 5,250, 1,770, 4,521, 1,792, 1,540, 3,145, 303, 714, and 479 for HFFc6, respectively. **c**. Comparison of transcription levels between genes with or without interacting enhancers in the Lamina SPIN state. The number of genes

with and without interacting enhancers is 421 and 544 in H1, and 108 and 371 in HFFc6, respectively. **d**. Comparison of the number of linked enhancers between active and inactive genes in the Lamina SPIN state. The number of active and inactive genes is 337 and 628 in H1, and 74 and 405 in HFFc6, respectively. **e**. Example genome browser views showing that expressed genes and their interacting enhancers are often synergistically looped out of the nuclear lamina to facilitate gene regulation. Blue arcs represent chromatin loops linking the gene at the centre of each region to distal enhancers. **f**. Lamin-B1 DamID-seq signals surrounding lamina-associated genes and their interacting enhancers in HFFc6. TSS, transcription start site; TES, transcription end site; TPM, transcripts per kilobase million.

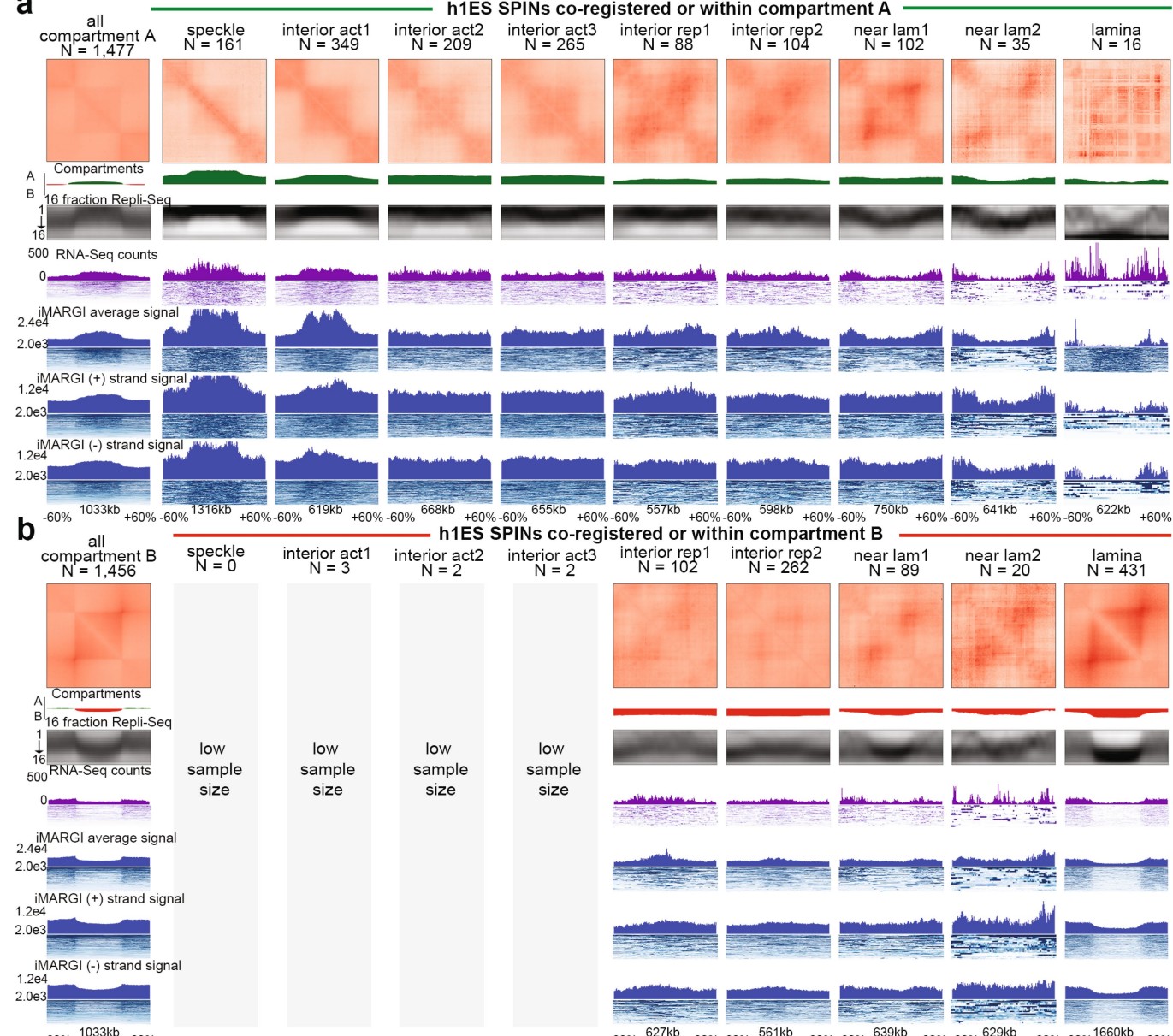

**Extended Data Fig. 9 | Compartment and SPIN integration with replication timing, RNA-seq, and nascent transcripts from iMargi. a, b**. Averaged Hi-C, replication timing (16 fraction Repli-seq), nascent transcription (iMargi), and mRNA levels (RNA-seq) for h1ESCs at all A/B compartments (column 1) and SPIN states either co-registered or co-localized within A/B compartments (columns 2–10). All genomics data is plotted as the average signal across all genomic intervals representing SPINs in a particular column. SPIN genomic intervals of (SPIN genomic interval +/− flanks of 60% of the size of the genomic interval) are stretched laterally to scale by size before average signal is computed. **a**. All A compartments or select SPINS co-registered or within compartment A and **b**. All B compartments or select SPINS co-registered or within compartment B. Tracks show pileups in H1 for Hi-C Aggregate-Peak-Analysis (APA), A/B compartment, 16 fraction Repli-seq, median RNA-seq signal, condensed RNA-seq reads, median averaged iMARGI (+) and iMARGI (−) signal, condensed iMARGI (+) and iMARGI (−) reads, median iMARGI (+) signal, condensed iMARGI (+) reads, median iMARGI (−) signal, and condensed iMARGI (−) reads.

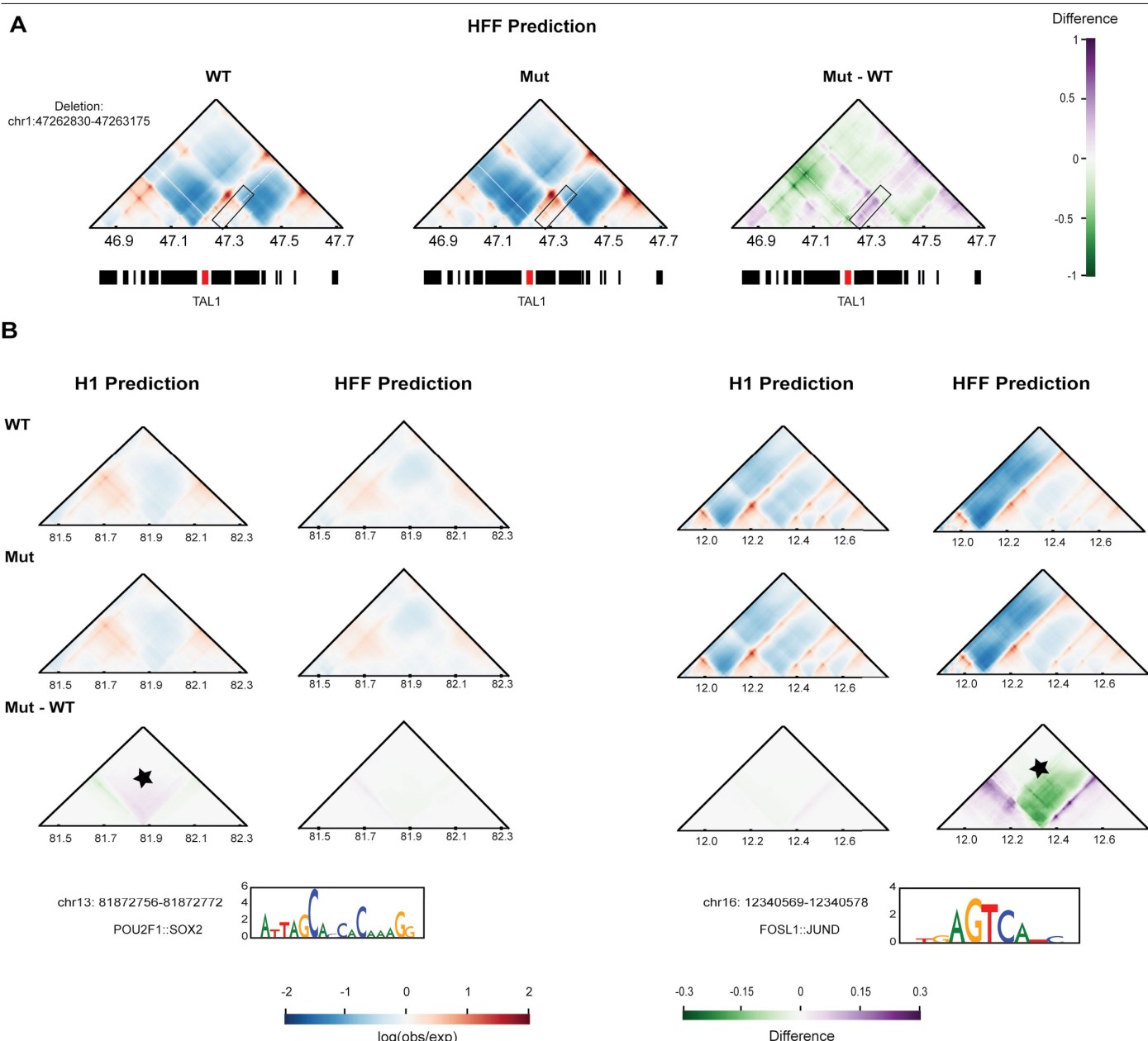

**Extended Data Fig. 10 | Predicting the effect of genomic variants on 3D genome folding with deep learning. a**. Example of a 345 bp deletion (chr1: 47262830-47263175) at the *TAL1* locus. We trained a model on HFFc6 Micro-C using the Akita architecture[124] and predicted contact maps for the reference human genome sequence (WT; ~1 Mb region) and the sequence carrying the deletion (Mut). Red: higher than expected interaction frequencies given genomic distance (log(observed/expected)); Blue: lower than expected. The effect of the deletion (Mut - WT) is plotted below. Purple: increased chromatin interactions; Green: decreased. Genes in the locus are plotted below the contact maps with *TAL1* highlighted in red. The deleted region has a CTCF binding site and is located in a TAD boundary. Mirroring the experimental deletion in HEK293T cells (Hnisz et al.[157]), the HFFc6 model predicts increased

contact frequency between *TAL1* and adjacent regions (black rectangle), as did an H1 model (data not shown). **b**. In silico mutation of transcription factor motifs (replacing motifs with random sequences) affects deep learning predictions of nearby chromatin interactions. An example POU2F1::SOX2 motif (left, chr13: 81872756-81872772) and FOSL1::JUND motif (right, chr16: 12340569-12340578) were generated using models with the Akita architecture trained on H1 or HFFc6 Micro-C data, respectively. Motif logos generated via model importance scores using DeepExplainer[128] are shown below the maps. These resemble the canonical motifs, though not precisely. Colour scales are the same as in (**a**), and motif sites are centred on the contact maps. Star symbols indicate regions with altered chromatin interaction predictions.

Corresponding author(s):    Job Dekker; Xiaotao Wang; Mario Nicodemi; Bing Ren; Sheng Zhong; Jennifer E. Phillips-Cremins; David M. Gilbert; Katherine S. Pollard; Frank Alber; Jian Ma; William S. Noble; Feng Yue

# Reporting Summary

## Statistics

For all statistical analyses, confirm that the following items are present in the figure legend, table legend, main text, or Methods section.

| n/a | Confirmed | |
|---|---|---|
| ☐ | ☒ | The exact sample size (*n*) for each experimental group/condition, given as a discrete number and unit of measurement |
| ☐ | ☒ | A statement on whether measurements were taken from distinct samples or whether the same sample was measured repeatedly |
| ☐ | ☒ | The statistical test(s) used AND whether they are one- or two-sided *Only common tests should be described solely by name; describe more complex techniques in the Methods section.* |
| ☐ | ☒ | A description of all covariates tested |
| ☐ | ☒ | A description of any assumptions or corrections, such as tests of normality and adjustment for multiple comparisons |
| ☐ | ☒ | A full description of the statistical parameters including central tendency (e.g. means) or other basic estimates (e.g. regression coefficient) AND variation (e.g. standard deviation) or associated estimates of uncertainty (e.g. confidence intervals) |
| ☐ | ☒ | For null hypothesis testing, the test statistic (e.g. *F*, *t*, *r*) with confidence intervals, effect sizes, degrees of freedom and *P* value noted *Give P values as exact values whenever suitable.* |
| ☒ | ☐ | For Bayesian analysis, information on the choice of priors and Markov chain Monte Carlo settings |
| ☒ | ☐ | For hierarchical and complex designs, identification of the appropriate level for tests and full reporting of outcomes |
| ☐ | ☒ | Estimates of effect sizes (e.g. Cohen's *d*, Pearson's *r*), indicating how they were calculated |

*Our web collection on statistics for biologists contains articles on many of the points above.*

## Software and code

Policy information about availability of computer code

| Data collection | Data from the 4D Nucleome Data Portal were manually downloaded. Data from the ENCODE portal were retrieved using a customized Python script, with the metadata file (metadata.tsv downloaded from ENCODE) as input. We have deposited this script along with the metadata.tsv file on GitHub at the following link https://github.com/XiaoTaoWang/4DN-joint-analysis/tree/main/data-collection.

iMARGI data for H1 or HFF cell lines were downloaded from: https://data.4dnucleome.org/experiment-set-replicates/4DNESNOJ7HY7/#processed-files for H1 and https://data.4dnucleome.org/experiment-set-replicates/4DNES9Y1GHK4/#processed-files for HFF. .iMARGI RNA ends genome wide coverage were downloaded from: https://data.4dnucleome.org/experiment-set-replicates/4DNESNOJ7HY7/#supplementary-files. Genome-wide SPIN states annotation were downloaded from: https://data.4dnucleome.org/joint-analysis#spin-states.

This study analyzed data publicly available through the 4D Nucleome Data Portal (https://data.4dnucleome.org; https://data.4dnucleome.org/joint-analysis) and the ENCODE data portal (https://screen.encodeproject.org/; https://www.encodeproject.org) (See Supplementary note 1). We refer to the Supplementary Materials for a full list of all data accession numbers and public links. This study produced three new datasets that are publicly available: loop calls in H1-hESCs and HFFc6 cells (https://doi.org/10.5281/zenodo.17451616; https://data.4dnucleome.org/files-processed/4DNFIX6VZKOA/; https://data.4dnucleome.org/files-processed/4DNFI32J1C6W/), SPIN state annotations for H1-hESCs and HFFc6 cells (https://data.4dnucleome.org/joint-analysis#spin-states; https://data.4dnucleome.org/files-processed/4DNFITDBR2LR/; https://data.4dnucleome.org/files-processed/4DNFI22FB8HD/), and 3D models and 3D structure feature profiles (https://zenodo.org/records/17459402; https://data.4dnucleome.org/files-processed/4DNFIJ36KR6X/; https://data.4dnucleome.org/files-processed/4DNFILC1YWO6/; https://data.4dnucleome.org/files-processed/4DNFIHF82NAI/; https://data.4dnucleome.org/files-processed/4DNFITOP9BQE/). |
|---|---|

Data analysis

Code availability
All code applied in this paper is publicly available through these links:
Methods benchmarking:
https://doi.org/10.5281/zenodo.17475348
GAM analysis
https://doi.org/10.5281/zenodo.17477229
Loop calling, and transcription-loop analysis:
https://doi.org/10.5281/zenodo.17456776
SPIN analysis:
https://doi.org/10.5281/zenodo.17469228.
iMargi analysis:
https://doi.org/10.5281/zenodo.17502239
3D genome structure modeling: Integrative Genome Modeling (IGMv2.0) platform
https://doi.org/10.5281/zenodo.17478448
www.github.com/alberlab/igm
Single cell analysis:
https://doi.org/10.5281/zenodo.17477302
TADs, SPIN, Compartment, and replication analysis:
https://doi.org/10.5281/zenodo.17467184
Replication timing analysis
https://doi.org/10.5281/zenodo.17485344
Predicting Hi-C data from sequence:
https://doi.org/10.5281/zenodo.17469981

FIMO (version 5.5.2) (Grant et al., 2011) and the JASPAR database (2023) (Castro-Mondragon et al., 2022) were used to identify transcription factor binding sites in the hg38 human genome assembly. In -house script was used to identify binding sites overlapping topologically associating domain (TAD) boundaries. The Akita model (https://github.com/calico/basenji/tree/master/manuscripts/akita) (Fudenberg et al. 2020) was used to quantify the effects of sequence changes on genome folding.  DeepExplainer (DeepSHAP implementation of DeepLIFT) (version 0.40.0) (Avsec et al., 2021; Shrikumar et al., 2019) was used to visualize the resulting contribution scores.

3D modeling display items: The analyses and most of the figure panels were performed using custom Python scripts (matplotlibv3.5, scikit-learnv1.0, scipyv1.5 and networkxv2.8) together with the publicly available alabtools platform (https://github.com/alberlab/alabtools). Spatial partitions were identified using the MCL algorithm (https://micans.org/mcl/). Images of 3D genome structures were generated using UCSF Chimera1.13.

Chromatin loops were identified using multiple tools depending on the assay type. For Hi-C and Micro-C, loops were called using cooltools (v0.5.1, dots subcommand; https://github.com/open2c/cooltools) and Peakachu (v2.3; https://github.com/tariks/peakachu). For ChIA-PET, loops were identified using ChiaSig (v1.19.44-r2; https://github.com/cheehongsg/ChiaSigScaled) and Peakachu (v2.3). For PLAC-Seq, loops were identified using MAPS (v1.1.0; https://github.com/ijuric/MAPS), ChiaSig (v1.19.44-r2), and Peakachu (v2.3).

Consensus chromatin states were calculated using ChromHMM (v1.23; https://compbio.mit.edu/ChromHMM/), with input ChIP-Seq tracks downloaded from the WashU Epigenome Browser and converted from hg19 to hg38 using CrossMap (v0.5.2; http://crossmap.sourceforge.net/). UMAP projections of chromatin loops were generated using the umap package (v0.5.5; https://github.com/lmcinnes/umap).

The software Higashi (https://github.com/ma-compbio/Higashi) is used for imputing the sparse scHi-C contact maps. The software SPIN (https://github.com/ma-compbio/SPIN) is used for SPIN states inference.
.
For SnapHiC: We used R version 4.2.2, Python version 3.6. SnapHiC version v0.2.3 code is available at: https://github.com/HuMingLab/SnapHiC.

Some Hi-C maps were drawn using ORCA (https://doi.org/10.1038/s41588-022-01065-4)

For manuscripts utilizing custom algorithms or software that are central to the research but not yet described in published literature, software must be made available to editors and reviewers. We strongly encourage code deposition in a community repository (e.g. GitHub). See the Nature Portfolio guidelines for submitting code & software for further information.

# Data

Policy information about availability of data

All manuscripts must include a data availability statement. This statement should provide the following information, where applicable:

- Accession codes, unique identifiers, or web links for publicly available datasets
- A description of any restrictions on data availability
- For clinical datasets or third party data, please ensure that the statement adheres to our policy

All data are available at 4DN data portal unless stated otherwise in the manuscript. The supplementary materials includes exhaustive lists of all dataset used in the manuscript, and provides all links. All data is publicly available.

This study analyzed data publicly available through the 4D Nucleome Data Portal (https://data.4dnucleome.org; https://data.4dnucleome.org/joint-analysis) and the ENCODE data portal (https://screen.encodeproject.org/; https://www.encodeproject.org). We refer to the Supplementary Materials for a full list of all data

# Research involving human participants, their data, or biological material

Policy information about studies with human participants or human data. See also policy information about sex, gender (identity/presentation), and sexual orientation and race, ethnicity and racism.

| | |
|---|---|
| Reporting on sex and gender | This is not relevant to our study. Our study is based on cell lines. |
| Reporting on race, ethnicity, or other socially relevant groupings | This is not relevant to our study. Our study is based on cell lines. |
| Population characteristics | This is not relevant to our study. Our study is based on cell lines. |
| Recruitment | This is not relevant to our study. Our study is based on cell lines. |
| Ethics oversight | This is not relevant to our study. Our study is based on cell lines. |

Note that full information on the approval of the study protocol must also be provided in the manuscript.

# Field-specific reporting

Please select the one below that is the best fit for your research. If you are not sure, read the appropriate sections before making your selection.

☒ Life sciences  ☐ Behavioural & social sciences  ☐ Ecological, evolutionary & environmental sciences

For a reference copy of the document with all sections, see nature.com/documents/nr-reporting-summary-flat.pdf

# Life sciences study design

All studies must disclose on these points even when the disclosure is negative.

| | |
|---|---|
| Sample size | There are 188 cells in the WTC-11 scHi-C dataset. No sample size calculation was performed. This is the number of cells that passed that passed our internal threshold for sufficient read coverage.<br><br>The number of  cells to be used for high-resolution repli--seq was determined so the final sequencing libraries do not have notable PCR duplicates.<br><br>The chromatin structure population contains 1,000 models. |
| Data exclusions | In general: No data or 3D models were excluded from analysis.<br>For single cell analysis: Ren lab filtered the dataset to get 188 cells so that each cell has sufficient read coverage |
| Replication | All 3D genome mapping assays analyzed in this study were performed with at least two biological replicates to ensure the reproducibility of the results. |

| | Each high-resolution repli-seq was done as only one replicate. However, in this multi-fraction assay, each fraction serves as control of other fractions. |
|---|---|
| Randomization | In general: This is not relevant to our study. Our study does not involve any allocations of samples/organisms/participants.<br><br>In the case of 3D models, this can be relevant: All our genome population calculations (via IGM) start out with fully randomized genome configurations. |
| Blinding | This not relevant to our study. Our study does not involve any group allocations. |

# Reporting for specific materials, systems and methods

We require information from authors about some types of materials, experimental systems and methods used in many studies. Here, indicate whether each material, system or method listed is relevant to your study. If you are not sure if a list item applies to your research, read the appropriate section before selecting a response.

### Materials & experimental systems

| n/a | Involved in the study |
|---|---|
| ☐ | ☒ Antibodies |
| ☐ | ☒ Eukaryotic cell lines |
| ☒ | ☐ Palaeontology and archaeology |
| ☒ | ☐ Animals and other organisms |
| ☒ | ☐ Clinical data |
| ☒ | ☐ Dual use research of concern |
| ☒ | ☐ Plants |

### Methods

| n/a | Involved in the study |
|---|---|
| ☒ | ☐ ChIP-seq |
| ☐ | ☒ Flow cytometry |
| ☒ | ☐ MRI-based neuroimaging |

## Antibodies

| Antibodies used | BD Pharmingen™ Purified Mouse Anti- BrdU, BD Biosciences, cat# 555627, clone 3D4, lot 7033666<br>Mouse monoclonal antibody Anti-Histone (Merck, Cat# MAB3422, clone H11-4, Chemicon® ) |
|---|---|
| Validation | Purified Mouse Anti- BrdU: this is a monoclonal antibody that has been used for DNA replication/cell proliferation assay for decades. In addition to the QC described in supplier's datasheet, we confirm the antibody specificity by quantifying the marker loci's DNA from immuno-precipitated nascent DNA from BrdU pulse-labeled cells.<br><br>Anti-histone antibody: validated by vendor via immunohistochemistry and western blotting. |

## Eukaryotic cell lines

Policy information about cell lines and Sex and Gender in Research

| Cell line source(s) | Stem cells (H1), hTert-immortalized human foreskin fibroblasts (HFF), and chronic myelogenous leukemia lymphoblasts (K562) were obtained from the 4D Nucleome (4DN) Cell Repository and cultured following the 4DN Consortium's approved culture protocol for each cell line (https://www.4dnucleome.org/cell-lines.html).<br><br>F121-9 from Rudolf Jaenisch lab at Whitehead Institute for Biomedical Research, female<br>HCT116 from ATCC, male<br><br>For scHi-C data generation<br>- The modified WTC-11 (GM25236) hiPS cell line with GFP tagged AAVS1 locus:  iPSC from Fibroblast, male, 30YR at sampling |
|---|---|
| Authentication | No specific authentication of cell lines was perfored. Note that we did perform RNAseq for all lines, and the data confirmed cell type (H1-hESC markers expressed in H1-hESC but not in HFFc6, and vice versa).<br><br>The cell lines in the 4DN Cell Repository were established by the 4DN Consortium in collaboration with WiCell and ATCC for providing quality-controlled cells from the identical batch to minimize cell source and culture condition variations. The cell culture protocols were developed by the 4DN Cell Line Working Group and approved by the 4DN Steering Committee.<br><br>For scHi-C data generation<br>- The modified WTC-11 hiPS cell line with GFP tagged AAVS1 locus (clone 6 and clone 28) was recommended by 4DN Consortium. |
| Mycoplasma contamination | Negative of mycoplasma from genomic sequencing data.<br><br>For scHi-C data generation<br>- The modified WTC-11 hiPS cell line with GFP tagged AAVS1 locus (clone 6 and clone 28) was tested negative for mycoplasma contamination. |

| Commonly misidentified lines<br>(See ICLAC register) | None of the cell lines used are on the list of commonly misidentified cell lines. |
|---|---|

## Plants

| Seed stocks | This not relevant to our study. |
|---|---|
| Novel plant genotypes | This not relevant to our study. |
| Authentication | This not relevant to our study. |

## Flow Cytometry

### Plots

Confirm that:

☐ The axis labels state the marker and fluorochrome used (e.g. CD4-FITC).

☐ The axis scales are clearly visible. Include numbers along axes only for bottom left plot of group (a 'group' is an analysis of identical markers).

☐ All plots are contour plots with outliers or pseudocolor plots.

☐ A numerical value for number of cells or percentage (with statistics) is provided.

### Methodology

| Sample preparation | Formaldehyde-crosslinked WTC-11 cells were lysed and permeabilized with 0.5% SDS. The lysates were then incubated overnight at 37 °C with 300 U of MboI, followed by proximity ligation using T4 DNA ligase at room temperature with gentle rotation for 4 hours. After staining with Hoechst, the nuclei were subjected to fluorescence-activated cell sorting (FACS). |
|---|---|
| Instrument | SH800 cell sorter (Sony Biotechnology) |
| Software | SH800 software (Sony Biotechnology) |
| Cell population abundance | Due to aggregation of nuclei caused by the Hi-C preprocessing steps, fewer than 10% of WTC-11 nuclei remained as single, isolated nuclei that could be sorted for downstream library preparation. |
| Gating strategy | Single nuclei were identified and gated based on FSC-A versus FSC-W parameters to distinguish singlets from aggregates. The Hoechst fluorescence channel was then used to differentiate 2N nuclei from those in replication or mitotic phases. |

☐ Tick this box to confirm that a figure exemplifying the gating strategy is provided in the Supplementary Information.

