## [Peer Review file · Nature]

An integrated view of the structure and function of the human 4D nucleome

Corresponding Author: Dr Job Dekker

Version 0:

Reviewer comments:

Referee #1

(Remarks to the Author)

This flagship manuscript, authored by a large and authoritative collaborative team, presents the outcomes of the NIH-funded 4D Nucleome Consortium. Focusing on two human cell lines, this manuscript presents data and analyses under the broad themes of (a) benchmarking of 3D genomic assays; (b) annotation of key features of 3D chromosomal architecture; (c) reconstruction of 3D structure from bulk and single-cell data; (d) linking 3D architecture and genome function and, finally, (e) predicting 3D genome structure from sequence.

Consortium papers of this scope and ambition are inherently challenging to write, and I fully appreciate the substantial talent and effort invested in both this manuscript and the 4DN project overall. That said, I felt that in its current form, the manuscript still has some way to go to fulfil the aspiration associated with flagship consortium papers of this calibre.

I don't imply that I have the exact solutions for how to make this story truly stand out, but I envisage that an ideal flagship consortium paper would:

- Incorporate a broad range of assays and datasets, pooling together data generated by consortium members and, where it is helpful, independent teams, reanalysing them consistently and releasing them as a user-friendly resource.
- Provide clear guidance on the best assays and post-hoc analysis pipelines to use for specific purposes.
- Highlight any novel methods and biological insights arising from specific analyses, methods and systems, ideally presenting them in more detail in companion papers presented and reviewed separately.
- Distill the key results of integrative analyses into conceptual advances in a way that highlights their implications, impact and potential to broader research audiences, opening up clear perspectives for new research and translation.
- Feature at least some attempts to validate the results using entirely independent methods and/or perturbation techniques.
- Acknowledge prior research, clearly distinguishing between the conceptually novel findings of the project and those that are more confirmatory—albeit potentially obtained on a larger scale and/or in a better-controlled setting.

While fully achieving this aspirational goal may not be feasible, I felt that there was room for this manuscript to be meaningfully improved in each of the areas mentioned above. Doing so would ensure that the paper does not get bogged down in technical details or reads as an undigested 'data dump'; presents a broadly useful, reusable and compelling resource; builds on, rather than 'rediscovers', prior research in this active field; and finally, delivers an exciting, cohesive and actionable message, rather than fragmenting into mini-papers of varying quality.

A particular challenge here is that the presented data are not substantially more extensive than those from numerous prior studies, some of which have been performed by the consortium members themselves. Similarly, the majority of the methodologies used are not novel. Nonetheless, I do not doubt that the wealth of findings and data resulting from this work has considerable potential. Therefore, investing some additional effort to substantially reframe and revise this manuscript to strengthen its core messages and weed out non-mission-critical technical details and secondary findings, while extending some analyses that currently appear quite rudimentary, would, in my view, be a great service to the legacy of this consortium.

My more specific comments are listed below, some of which, but not all are related to my general message above:

p10: I was surprised that Capture-(C/Hi-C/Micro-C) didn't get at least an honourable mention alongside ChIA-PET/HiChIP, and was not selected in the benchmarking analysis. Could at least some cross-method analysis use published Capture-C/Hi-C/Micro-C data?

p12: I was not sure what was meant by "seen as a composite"
also p12: If the enrichment for RNAPII is similar across Hi-C, I wonder why RNAP-specific effects on contacts were prominently detected by Micro-C (Zhang, Nat Genet 2023; Barshad, Nat Genet 2023) but not (Capture)Hi-C (e.g., El Khattabi, Cell 2019).

p13: Please clarify what is meant by "dynamic range".

p14: Do the authors have any ideas why compartments are stronger with Hi-C than with Micro-C and whether this could have something to do with Micro-C being biased for the A compartment? Also, what could drive differences in compartment size?

p15: It would be great to have some mechanistic explanation for these differences that could be used to potentially improve the protocols detecting weaker effects (if it is desirable), particularly Micro-C which is rapidly gaining popularity - at least speculatively, but potentially also with some experimental evidence. also perhaps some recommendations on what method to use and when?

p18-23. The loop anchor section is informative and useful but gets somewhat lost in detail. Can it be further digested to be more readable?

p18: Did I get it right that clustering was done based on the two UMAP dimensions rather than the underlying multidimensional space (e.g., PCA)? This is generally not recommended.

p19, Figure 3e: It is quite hard to see differences between plots except for Pol2 ChIA-PET, and if this is the only point you're making perhaps it doesn't merit a main figure.

p22: I'm not sure I fully got this point: "Loops in the fourth and sixth clusters are of particular interest given that they are less enriched for RNA polymerase II, but are highly enriched for cohesin binding, which suggests that cohesin might be important in mediating enhancer-promoter loops (note that these clusters are enriched with interactions between enhancers and promoters based on Figure 3d)."

p22: "Interestingly, although both Hi-C and Micro-C are designed to detect all-to-all chromatin interactions without bias, loops detected with these methods are less enriched with transcription-related loops. These assays appear to excel in detecting structural CTCF-anchored loops (Figure 3f-g)." I think it's important to make a slightly more nuanced point here, given that Micro-C was used to demonstrate transcription-dependent loops and Hi-C and Capture Hi-C cohesin/CTCF-independent E-P contacts.

p26-27: the SPIN framework result, as presented, is somewhat underwhelming. It would be good for the authors to explain in what way this annotation is more useful than histone mark-derived states (if they believe so) and/or what can we learn from the correlation between these two annotations. Do SPIN states have stronger associations with caRNAs than histone marks, for example? It would also be great to explain more clearly why SPIN is more useful/relevant for the 3D analyses that follow compared to other epi- and 3D genome annotations. One possibility is that this is simply a better annotation as it provides some 3D structural anchoring - but for this, it would be good to have a better idea of how confident the SPIN state assignments to various microscopic subnuclear structures are.

p28-35: Given the rise of imaging-based technologies, it would have been great to see some validation of the structural findings with direct measurements from high-resolution microscopy. In addition, from a reader's perspective, I wanted more clarity in the authors' message regarding bulk vs single cell-based modelling approaches. Which ones are "better" (given the sparsity challenges of scHi-C) and/or how do they complement each other?

p34: Minor comment: I think the last sentence on the page has some textual issues.

p36-43: The relationship between chromatin loops and the genome function is the part where I felt the most that this study is trying to reinvent the wheel. A large number of comparative and perturbation studies, including using high-resolution flavours of Hi-C, have looked at this question very extensively and comprehensively. It is very important to make it clear what exactly the robust new finding(s)/concept(s) are from the analyses presented here - including for the authors' sake, as this will help protect them from the usual criticisms that large, government-funded genomics consortia typically attract, and often undeservedly. Perhaps focusing on house-keeping genes was one attempt at providing such a novel finding, but (a) if it's novel, why would others not see it before and (b) given the sparse nature of most techniques employed here, how much confidence should we put in the findings of "dynamics" (as opposed to sampling error)?

p.44-49: Again, I have reservations regarding the conceptual novelty of most of the presented findings, including the relationship with replication timing that was covered by David Gilbert in his lab's earlier work.

p53: The sequence-based 3D modelling part is rudimentary, particularly in light of the ever-expanding array of tools and approaches for deep learning-based 3D structure prediction. Given that the authors dedicated much effort to benchmarking genomics assays, would it make sense for them to at least attempt 3D structure prediction using a variety of models, if not benchmark them formally?

Referee #2

(Remarks to the Author)

In the paper by Dekker and colleagues, titled “An Integrated View of the Structure and Function of the Human 4D Nucleome”, the authors provide a thorough description of the work conducted by numerous labs as part of the NIH’s 4D Nucleome Project. The account is highly detailed and will likely be well-received by the global 4D nucleome research community. The manuscript is well-structured, with clearly outlined sections. The figures are appropriate, and the supplementary material and figures are extensive. Altogether, this work represents a monumental effort in delineating the structure and function of the genome.

Overall, the findings presented in the manuscript are highly engaging for a researcher like myself, working in the field of 4D nucleomics. However, some sections are challenging to follow or overly repetitive for a non-specialized audience. I suggest that the authors simplify certain sections to make the content more accessible and appealing to readers who are less familiar with nuclear organization. Similarly, some figures could be simplified without losing their ability to convey the main message. Conversely, the discussion section feels somewhat limited and could benefit from expansion to better contextualize and interpret the findings presented.

Below, I outline additional suggestions for improvement beyond the two main points mentioned above:

- I recommend generating a narrative text for the final part of the introduction, where the authors outline the detailed analysis and integration of data.
- Figure 1 is beautifully crafted, but this is not the case for many of the subsequent figures. Some of these (particularly Figures 7 and 8) are overly cluttered, with excessive panels that obscure the message they aim to convey.
- The authors have provided a thorough introduction to the specificities and particularities of each experimental method used to interrogate the genome. This allows readers to understand how methods like SPRITE and GAM differ from other approaches. However, it would be beneficial to include a section that contextualizes these methods practicalities discussing when each is most appropriate and how their results should be interpreted. This addition could enrich the discussion section.
- The authors could speculate as to why there are large quantitative differences between the methods (page 15) in detecting AB compartments.
- As indicated by the authors (page 23), Hi-C and Micro-C are unbiased when detecting interactions. However, they are less enriched when detecting transcription-related loops. Can the authors speculate why is that?
- Figure 4 seems to be incomplete as for the text as well as the figure itself.
- Could the authors elaborate on the sentence indicating that in certain cell types, S[^]PIN states have a cell type-specific distribution of repressive histone marks such as H3K27me3 (page 26).
- Can the authors quantify the “frequently” in the sentence “Differences in gene expression between H1-hESC and HFFc6 cells frequently correlated with notable differences in the nuclear microenvironment of genes”. Page 28.
- Following the text in the section “Gene expression” when discussing highlight expressed genes is difficult (pages 29-30). I would revisit it to clarify the points.
- The entire text focuses on the works for 2 cell types for genome-wide analysis. However, in the section of integrative modelling of single-cell 3D genomes, the work is done in a different cell type and for a 2Mb region of the genome. Maybe an introductory text as to why would be very welcome.
- When analysing how SPIN states are embedded or not into different hierarchical organization of the genome (AB, TAD, etc...), could the authors speculate as to for some matchings do not agree (SPINs for silencing embedded in A and vice versa). Could it be related to the resolution of the data for the specific analysis? Other possible reasons explaining such contradictory results? (page 45).
- The section “predicting chromosome folding from sequence” seems not to be well integrated into the main text. Additionally, the results in Figure 9 as explained in the text are poor compared with other sections of the manuscript. How much it adds to the overall story as it is now written?

Other minor points.

- Add space between “comparing” and “the” (page 8)
- Figure 2. Panel b. I would add a label to the bar legend. There seems to be a misplacing in the labels for the x-axis.
- Figure 2. Panel c. Move legend outside the plot. Right now, the dots in the legend could be mistakenly interpreted as data.
- Figure 2 Panel d. I would add a label to the bar legend.
- Lower case “f” in “(For” page 15.

Referee #3

(Remarks to the Author)

This manuscript from the 4D nucleome consortium brings together a range of different methods to study the 3D genome on two different cell lines (H1-hESCs) and HFFc6 (a human foreskin cell line). The most exciting aspect of this enterprise is that different methods (Hi-C, Micro-C, ChIA-PET, HiChIP SPRITE and GAM) can be compared side-by-side, enabling the assessment of the merits of these methods. Comparison of various genome organisation assays is important since different papers make claims using different methods and it is not necessarily clear how suited various methods are for quantifying 3D genome features at different scales and which artifacts they present with. The authors extensively quantify how the methods differ in quantification of different features, however it remains somewhat vague which methods should practically

be used for which purpose.

This reviewer appreciates the amount of data that has been generated, unfortunately, the biological insight that is gleaned from this massive dataset is rather limited. In fact, many trivial observations (e.g. what a saddle plot is) are discussed at length, whereas in the second half of the paper many non-standard methods (e.g. iMARGI) are discussed without proper introduction. Currently, the manuscript feels very disjointed with the different sections not linked in any meaningful way. The manuscript could benefit from someone taking the lead and streamlining the text and making it into a more cohesive story.

Below specific comments are presented, to improve the readability of the manuscript:

- Figure 2a: What are the line plots underneath the heatmap. Insulation score is assumed, but please indicate.
- Figure 2b has misaligned labels on x axis.
- The authors state that “Visual inspection of these contact maps at large genomic distances show that ligation-based methods capture similar chromatin organization patterns, independently of whether contacts are mapped with Hi-C, Micro-C, GAM, SPRITE, or based on occupancy of specific proteins (CTCF, RNA Pol II, H3K4Me3)”. Based on the example plot in Supp. Fig. 1a GAM does not seem to capture similar patterns. Additionally, the correlation with GAM compartment scores vs others is really low.
- Please plot Supp. Figure 1a the same way as Figure 2a: i.e. compartment level, TAD/loop level. This allows for a proper comparison of all methods.
- Please make Supp. Figure 1a a main figure. It is the only figure where all methods are (and can be) compared directly. This seems like very useful data for the community. It would be unfortunate if this ends up in the Supplement.
- Extended Data Figure 2a: The color bars for the APAs differ greatly. Please set all the color bars to be consistent with the Hi-C/Micro-C (i.e. 1.1-4.5) with the exception of the ChIA-PET data.
- Please also generate APAs for the PolII loops identified by PolII ChIA-PET. It would be important to know whether these loops can be detected with the other methods or whether they can only be detected with ChIA-PET.
- Adding a schematic to show that chromHMM was used at loop anchors to create UMAP in Figure 3C and Extended Figure 3B would be helpful to the reader.
- In Figure 3, how do the authors choose 6 clusters in c, d, e, whereas in the text roughly 2 groups are mentioned, while the figure 3b shows 4 clusters?
- In Figure 3c, what is the point of the UMAP? If it is to quantify the separation between clusters, the authors should rather show more quantitative clustering metrics (e.g. proportions of k-nearest neighbours belonging to the same cluster; silhouette widths etc.) instead of a 2D embedding.
- In Figure 3e, it is not clear what the colour encodes – signal strength in each loop? If so, it is hard to get any sort of quantitative insight of loop strengths in each cluster from a 2D embedding, a better alternative might be to visualize the distribution of loop strengths for each method in each cluster.
- Is there a more quantitative way of presenting the data in Figure 3e? The highlighted UMAP projections are not very easy to interpret in this particular case.
- In Figure 3g, transcription-related loops are not convincing. Loop calling in Pol2 ChIA-PET data in the provided example locus seems overly zealous and could easily be false positives, which are enriched overall at active gene promoters only because that is where Pol2 is localized so that is where any non-specific signal is found. See above regarding comments on APAs for Micro-C.
- Is the locus shown in Figure 3g the best representation of transcription-related loops? For example, would it be clearer to select a region where there are loops detected by both ChIA-PET and PLAC-seq with clear overlap?
- The authors state that “Unbiased genome-wide assays such as Hi-C and Micro-C most efficiently capture CTCF/cohesin based loops, while assays targeting RNA Pol II or H3K4me3 capture more transcription-related loops”. In Ext. Data Figure 4b, it appears that clusters 3-6 have sizeable fractions of loops which are not CTCF-bound. Perhaps it would be of interest to explore the differences in epigenetic states and binding of transcription factors in CTCF-bound vs unbound loops in these categories.
- The authors state: “We further characterized loops using additional transcription factor binding data and found that loops from different clusters are enriched for distinct transcription factors (Extended Data Figure 4; Supplemental Methods). Notably, Polycomb-group (PcG) proteins, such as EZH2 and RNF2, are specifically enriched at loop anchors in the first cluster.” Polycomb group proteins are not transcription factors, please reword.
- The cell types used for SPIN modelling are not clear. Supplementary Figure 4B mentions K562 comparison of previous versus joint K562 states, while the text and methods only mention SPIN modelling for H1-hESC and HFFc6. Could the authors clarify this part?
- It is unclear how Figure 4 fits with the rest of the manuscript. The SPIN model and its relation with histone marks and genomic structures has already been published as referenced. caRNA data analysis in this context is novel but does not provide much biological insight. The flanking box plots are confusing as they do not refer to anything. This is perhaps better suited for extended/supplementary material.
- The reviewer appreciates the analyses of the relationship between 3D structure and gene expression presented in Figure 5 and extended Figure 5. It would be of great benefit to the reader if the authors could provide an example of class I and class II genes with corresponding Hi-C maps, genomic tracks of histone modifications, TSA-seq and Dam-ID to make it even clearer to the reader that these are inherently different in terms of their genome architectural properties.
- In Figure 5c, it is not stated what NAF abbreviation stands for.
- In Figure 5i and 5j, it is not clear what the colour scale encodes and what information does a 2D embedding provide.
- In Figure 5j, how is it possible that the bottom quartile of the genes by SAF contains a different number of genes than the top SAF quartile? Were the genes first divided into class 1 and class 2? It should be made clearer how these subsets of genes were selected.
- It is unclear what the biological rationale for normalizing the number of enhancers to the number of nearby genes (i.e. enhancer density) is, since enhancers are not exclusive to single genes. This could be clarified more.

- The analyses of gene expression in relation to nuclear positions of genes in Figure 5 and Ext. Data Figure 5 contain interesting observations. Perhaps it would make sense to implement interesting examples of genes in Ext. Data Figure 5g and 5h into the main figure and move Figure 5d and 5f-g to extended/supplementary material. This would put more focus onto biological insights one can gain from these data.
- Regarding scHi-C imputation with Higashi and the SBS polymer model, it is not visually apparent that the insulation scores in Figure 6a are concordant between the two models. Same for the contact strength matrices in 6b. Please quantify.
- The authors should provide Hi-C maps for the RABGAP1L loop mentioned in the boxplot of Figure 6c?
- Figure 6d is mentioned in the text but not in the figures.
- It is not clear how the p-values in Figure 6c are calculated and are potentially influenced by pseudoreplication if each cell is treated as an independent replicate (e.g. for scRNA data see <https://www.nature.com/articles/s41467-021-21038-1>, <https://www.nature.com/articles/s41467-022-35519-4>, <https://www.nature.com/articles/s41467-021-25960-2>).
- The hypothesis that promoters of active genes near LADs locally carry active marks and may need to loop out of this environment for activation is very interesting (Figure 7). Given the amount of chromatin interaction data generated by the authors, would it be possible to perform quantitative analysis of the interactions of these genes? For instance, the authors could compare these to a set of control genes from similar environment to show that active genes near LADs have more of such interactions.
- Figure 8 and the related text present interesting findings on the association of SPIN states to gene expression in relation to the genomic compartment context that these SPIN states are embedded in. Particularly interesting is the finding that certain SPIN states seem to “bookmark” particular sets of genes irrespectively of which compartment they are in. The authors hypothesise that these could be hESC domains that are poised to switch on during differentiation. Perhaps it would be useful to quantify whether indeed these domains are enriched for genes that need to be switched on during differentiation to establish proper transcriptional programmes. Additionally, since this part of the paper further explores relation of nuclear positioning to gene expression, it would perhaps better fit after Figure 5 to maintain the flow and readability of the paper.
- Figures 8f-h do not seem to be referenced anywhere in the text.
- What is 3DNetMod? It is not introduced in the text and does not seem to be referenced. Please explain.
- The data on predicting chromatin structure from DNA sequence is very exciting. Regarding the trained models, is it possible to extract the DNA motifs that these models have learned to understand which are important for establishing chromatin interactions? For example, would these models learn cell-type-specific TF motifs in different cell types (such as POU2F1::SOX2 and FOSL1::JUND in H1-hESC and HFFc6, shown in Figure 9B)?
- In Figure 9a, the Akita model seems to nicely capture the effect of the deletion in the TAL1 locus in the HFFc6 model. Does this locus show different 3D genome organisation and gene expression in hESC? If so, what does the prediction look like for the hESC model?
- In Figure 9b, the effect size appears to be tiny. If this is the best example of a prediction in hESC, this approach seems to be of limited usefulness. This should be discussed.
- The authors state: “We found that the H1-hESC model is more sensitive to mutation of motifs for the embryonic stem cell factors POU2F1::SOX2, while the HFFc6 model is more sensitive to the fibroblast factors FOSL1::JUND.”. This is not shown or quantified anywhere, only the two examples are shown in Figure 9.

Textual comments:

In the discussion, the authors state that “Looping interactions with distal enhancers is strongly correlated with gene expression”. This is too general/strong of a statement. The authors have only qualitatively shown correlation of expression with the number of loops formed between the gene and potential enhancers and the correlation with the interaction frequencies was not assessed.

“Loops in the fourth and sixth clusters are of particular interest given that they are less enriched for RNA polymerase II, but are highly enriched for cohesin binding, which suggests that cohesin might be important in mediating enhancer-promoter loops”

“Further, most loop anchors, of any type, are associated with cohesin, suggesting a general role of this loop-extrusion complex in chromatin looping.”

“For instance, we find that cohesin is enriched at a large proportion of anchors of all types of loops, suggesting that cohesin, and possibly loop extrusion, is involved in their formation.”

The use of the words “suggests” and “suggesting” suggests that the role of cohesin in loop formation and loop extrusion is still unresolved. However, many papers have shown the role of cohesin in loop formation (e.g. Rao et al. 2017; Rhodes et al. 2020; Davidson et al 2019 and many more).

“However, this is not directly demonstrated through perturbation experiments, and other mechanisms most likely will also play roles.”

Not in this manuscript perhaps, but RAD21-degrons have clearly shown the role of the cohesin complex in loop formation. See examples given above.

“We found that the H1-hESC model is more sensitive to mutation of motifs for the embryonic stem cell factors POU2F1::SOX2”

This is a somewhat puzzling comment. The PWM in the figure does not look like the POU2F1::SOX2 motif in JASPAR. The most commonly identified motif in pluripotent cells is POU5F1::SOX2. Of the two citations one is not referring to Oct-1/Sox2 motif and the other does, but it is a rather obscure observation. Some more explanation towards the relevance of this observation would be useful.

Version 1:

Reviewer comments:

Referee #1

(Remarks to the Author)

The authors have done a good job addressing my specific comments. Whether these revisions have sufficiently elevated the paper above the level of the initial submission remains an open question. To me personally, it still feels somewhat underwhelming for a consortium and a journal of this calibre. A key challenge may lie in presenting high-impact, novel insights that go significantly beyond those already reported in the previously published consortium papers, which are now helpfully listed in Supplemental Table 1.

What might work better for me in this context would be a stronger emphasis on synthesizing the results of those earlier papers, while incorporating new high-level analyses such as benchmarking, rather than striving to present this paper as a set of entirely novel findings. However, I defer to the editors on whether this approach aligns with their expectations.

Specific comments:

- 1) I could not find a reference to Supplemental Table 1 in the main text.
- 2) The penultimate sentence in the abstract is missing a verb and is likely truncated.
- 3) Supplemental Figure 12:

Panel (a): It's a bit odd that global methods such as Hi-C and Micro-C are shown as equally suitable for detecting cCRE interactions as enrichment-based methods. After all, if this were the case, there would be no need to develop enrichment-based methods such as HiChIP in the first place. Unless the authors have strong opinions on that topic, I would suggest giving two stars to Hi-C and Micro-C for global cCRE detection rather than three.

Panel (b): Capture-C and Capture Micro-C seem to be missing from the decision tree entirely. I'd also suggest adding some colour/framing to the diagrams so they are easier to use.

- 4) Supplemental Figure 13: The figures and text are low-resolution and appear blurred. I also note that they are based on Capture Hi-C data, which is not referred to in the Supplemental Figure 12 at all.

Referee #2

(Remarks to the Author)

The authors have addressed all my initial concerns. Thanks.

Referee #3

(Remarks to the Author)

The reviewers have responded to my comments. They have made a new figure (Supplemental Figure 12) to guide the choice of method for 3/4D genome analysis, unfortunately this figure was not included in the submission, so this could not be assessed.

Bullet list of major changes to the revised manuscript

- We have added a supplemental table of all 4D Nucleome consortium companion papers (32 papers associated with this Flagship manuscript). These papers describe individual methods and datasets. This table serves to show how the integrated analysis presented in the current manuscript is related to the overall efforts of the consortium.
- We have added Supplemental Figure 12 that provides guidelines and recommendations for when to use which assay dependent upon research question and biological material, as several reviewers requested. This was a great suggestion, and another important output of this analysis for the research community.
- Most figures have been thoroughly redesigned to make them more consistent and less cluttered
- The text has been shortened to less than 10,000 words, as agreed upon with the editor. Any changes in the text are in blue font color.
- As requested, we have added a new analysis of capture Hi-C data that was generated by 4DN consortium members and show that adding another assay adds even more chromatin loops, further emphasizing a main finding from our integrative analysis that combining multiple datatypes is important for achieving more comprehensive loop annotations.
- Former Figure 8 (current Figure 7) - has been extensively redesigned, as requested. We deleted half of Panel a and deleted panels d and e that summarized data also shown in Extended Data Figure 9. Text related to this figure was substantially reduced, redundancy with earlier sections was eliminated and sections were clarified to reflect reviewer's requests. The section was re-titled to eliminate the appearance of redundancy.
- The sequence-based machine learning section was moved to a future prospects section in the Discussion, with the figure moved to Extended Data, addressing comments about length and bolstering the discussion.
- Following recommendation for the reviewers we have performed new clustering analyses of chromatin loops. While the main results did not change, this clustering improved the robustness of the analysis.
- We now include representative examples of class I and II genes along with their corresponding Hi-C maps, and genomic tracks of histone modifications, SON-TSA-seq, Lamina Dam-ID signals and RNA-seq signal tracks.
- We now illustrate differences in long-range promoter interactions between class I and II genes through a pile-up analysis of Hi-C contact frequencies around class I and II promoters.
- We now provide the absolute enhancer count within a distance of 350 nm from gene promoters without normalizing enhancers by the number of nearby genes.
- Reorganized and consolidated former Figure 4, 5 into current Figure 4 and Extended Figure 5 (current Figure 4, Extended Figure 6, Supplementary Figure 6).
- We have revised the text when discussing structural differences of genes with significant differences in gene expression between cell types.

Point-to-point reply

Referee #1 (Remarks to the Author):

This flagship manuscript, authored by a large and authoritative collaborative team, presents the outcomes of the NIH-funded 4D Nucleome Consortium. Focusing on two human cell lines, this manuscript presents data and analyses under the broad themes of (a) benchmarking of 3D genomic assays; (b) annotation of key features of 3D chromosomal architecture; (c) reconstruction of 3D structure from bulk and single-cell data; (d) linking 3D architecture and genome function and, finally, (e) predicting 3D genome structure from sequence.

Consortium papers of this scope and ambition are inherently challenging to write, and I fully appreciate the substantial talent and effort invested in both this manuscript and the 4DN project overall. That said, I felt that in its current form, the manuscript still has some way to go to fulfil the aspiration associated with flagship consortium papers of this calibre.

We thank the reviewer for their thoughtful comments, and have made all efforts to address them below.

I don't imply that I have the exact solutions for how to make this story truly stand out, but I envisage that an ideal flagship consortium paper would:

- Incorporate a broad range of assays and datasets, pooling together data generated by consortium members and, where it is helpful, independent teams, reanalysing them consistently and releasing them as a user-friendly resource.

We completely agree with the reviewer. The focus and aim of the 4D Nucleome consortium, and this flagship paper, have indeed included all these criteria: we have incorporated a range of assays, pooled and shared the data, did cross-group analysis, and released all data publicly through a data portal (<https://data.4dnucleome.org>) that includes raw data, processed data, and data displayed in browser for direct visualization by all interested.

- Provide clear guidance on the best assays and post-hoc analysis pipelines to use for specific purposes.

The benchmarking section of the manuscript focused on exactly that. That section quantitates performance of a large set of assays, revealing critical quantitative differences. We agree that we have not stated explicitly any direct guidance on best assays. In the revised manuscript we added a table and decision tree with our recommendations for use of specific assays for addressing specific research questions (new Supplemental Figure 12).

- Highlight any novel methods and biological insights arising from specific analyses, methods

and systems, ideally presenting them in more detail in companion papers presented and reviewed separately.

This is an excellent suggestion. In the revised manuscript we have worked hard towards emphasizing new methods and biological insights.

We note that we failed to make clear that the consortium has indeed put together a large set of companion papers that have been reviewed and published separately. It has been the policy of the 4DN consortium that these papers were published on their own (and much faster) timeline than this Flagship manuscript so that work of trainees could be published without the further delay that is intrinsic to publishing complex consortium flagship papers.

In the revised manuscript we now include a table listing of all 32 companion papers. Most of these were published in Nature family journals, so that they could easily be linked when the Flagship would be finally accepted.

A major advance of the current manuscript is that it shows that 1) different methods differ quite distinctly in the 4D nucleome features they can quantitatively detect. This is critical for other scientists to be aware of. 2) It shows that integration of data obtained with different methods provide more comprehensive feature lists: while prior papers, including the companion papers, each use only 1, sometimes 2 methods, here we integrate many more leading us to present significantly more chromatin loops (over 140,000 per cell type, while single methods produce tens of thousands), more detailed chromatin domain types and annotations, and more refined 3D models of the genome. These comprehensive datasets and annotations for widely used cells, esp. H1-hESCs, can provide a rich resource for new types of analyses and of looping features and long-range gene regulation that can be leveraged as deep training sets for future computational modeling. Such methods include predicting similar levels of looping in other cell types using computational methods we highlight at the end of the manuscript.

- Distill the key results of integrative analyses into conceptual advances in a way that highlights their implications, impact and potential to broader research audiences, opening up clear perspectives for new research and translation.

We have made attempts to follow the reviewer's suggestions. Several sections of the manuscript have been streamlined. We hope this new version better emphasizes the key messages.

- Feature at least some attempts to validate the results using entirely independent methods and/or perturbation techniques.

The benchmarking section where we compare a set of genomic assays serves as a cross-method validation. Importantly, this section shows that all methods show significant overlap in what they detect, but there are rather large quantitative differences. The major challenge is that there are no gold standards or ground truths. Perhaps the reviewer would consider imaging

(e.g., FISH) as a useful orthogonal approach. In fact, the 4DN consortium has performed a very detailed comparison of FISH vs. Hi-C. This large scale comparison, published in *Cell* in 2019 (Finn et al. *Cell* 2019) showed extensive agreement and suggested the capture radius of Hi-C is around 200 nm. In the revised manuscript we cite this validation. We agree that it would be great to compare all genomic assays benchmarked against each other here vs. imaging data. We believe that is beyond the scope of what the consortium can do at this point. However, having all data public will allow the community to embark on such analysis.

We also refer the reviewer to our response to a similar comment about validation below.

- Acknowledge prior research, clearly distinguishing between the conceptually novel findings of the project and those that are more confirmatory—albeit potentially obtained on a larger scale and/or in a better-controlled setting.

We care deeply about giving appropriate and generous credit to prior work. We had made all efforts to ensure we cited prior research. As the reviewer makes clear, we obviously could have done better. In the revised manuscript we once more made efforts to cite all prior work and added many new citations.

While fully achieving this aspirational goal may not be feasible, I felt that there was room for this manuscript to be meaningfully improved in each of the areas mentioned above. Doing so would ensure that the paper does not get bogged down in technical details or reads as an undigested ‘data dump’; presents a broadly useful, reusable and compelling resource; builds on, rather than ‘rediscovers’, prior research in this active field; and finally, delivers an exciting, cohesive and actionable message, rather than fragmenting into mini-papers of varying quality.

A particular challenge here is that the presented data are not substantially more extensive than those from numerous prior studies, some of which have been performed by the consortium members themselves. Similarly, the majority of the methodologies used are not novel. Nonetheless, I do not doubt that the wealth of findings and data resulting from this work has considerable potential. Therefore, investing some additional effort to substantially reframe and revise this manuscript to strengthen its core messages and weed out non-mission-critical technical details and secondary findings, while extending some analyses that currently appear quite rudimentary, would, in my view, be a great service to the legacy of this consortium.

The reviewer captures the challenge of putting together such integrated paper very well. We have taken the reviewers’ comments, including the very helpful more specific suggestions below, very seriously and worked hard to make the manuscript more focused and concise, and highlighting better the new insights.

This manuscript does not present new methods because any such methods have been presented in earlier work from this group and in companion papers (See new Supplemental Table listing all 32 companion papers). Rather, as we now highlight in the revised manuscript, this manuscript summarizes the efforts of the consortium to integrate datasets obtained with a

comprehensive set of genomic assays. We show how combining analysis of different datasets adds important insights. For example, and this is a major contribution to the field, we find that the number of loops will be severely underestimated when only a single assay, e.g., Micro-C, is used. Combining loops called with a set of different assays leads to detection of more loops and more different types of loops (e.g., structural loops and transcription-related loops). This manuscript delivers, to our knowledge, the most comprehensive set of loop calls to date. This will provide a rich resource for the field, including as a training set for efforts to predict long-range interaction for other cell lines. In the revised manuscript we highlight this in more detail.

My more specific comments are listed below, some of which, but not all are related to my general message above:

p10: I was surprised that Capture-(C/Hi-C/Micro-C) didn't get at least an honourable mention alongside ChIA-PET/HiChIP, and was not selected in the benchmarking analysis. Could at least some cross-method analysis use published Capture-C/Hi-C/Micro-C data?

The reviewer is correct that Capture-C data was not included. This is an unfortunate consequence of the timing of the project. When this benchmarking section started (2015-2017), no Capture-Micro-C assay was available, and now the project is no longer generating new data. In addition, we feel that it will be extremely difficult to fully, and fairly, compare genome-wide assays with targeted analyses of "just" a few Mb here and there in the genome (including earlier variants such as Capture-C).

We do agree that a comparison to Capture-C/Hi-C/Micro-C data would add significantly to the manuscript. The Ren lab has more recently generated Promoter Capture Hi-C (PCHi-C) in H1-hESC cells (PMID: 31501517). We have now performed a new analysis to compare chromatin loops detected by PCHi-C and the set of 3D genomic methods profiled in this study (Supplemental Figure 13). Specifically, we classified PCHi-C interactions into two categories: promoter-promoter (PP) and promoter-other (PO) interactions. We observed that 90.2% of PP interactions and 72.7% of PO interactions overlapped with loops from other methods, with most of the overlap coming from H3K4me3 PLAC-Seq or Pol2 ChIA-PET. Notably, PCHi-C-specific interactions more frequently connected promoters with less active regions, compared to those that overlapped with loops from other platforms. This is consistent with the nature of PCHi-C as a sequence bait-based assay, in contrast to the active chromatin-enriching methods used in this study, such as H3K4me3 PLAC-Seq and Pol2 ChIA-PET. This result shows that PCHi-C is particularly suited for this subset of looping interactions.

Importantly, this new analysis shows that adding yet another assay leads to identification of additional looping interactions that were missed by the combined performance of all other methods already included. This is an important insight, showing that many more loops still remain to be discovered and this further emphasizes the need to use multiple chromatin interaction assays in parallel for comprehensive loop detection. We have added this result in the revised manuscript.

interaction number for each category. **b**, Comparison of genomic distances between interacting anchors (i.e., loop sizes) for interactions unique to PChi-C versus those also supported by other methods. **c**, Fold-enrichment of ChromHMM states at loop anchors for interactions unique to PChi-C or shared with other 3D genomic methods.

p12: I was not sure what was meant by “seen as a composite”

We meant to indicate that the “untargeted” approaches such as Micro-C capture the “sum” (or “composite”) of what is captured by “targeted” approaches such as ChIA-PET for Pol2. In the revised manuscript we have rewritten and shortened this section of the manuscript and we hope that we now make more clear how targeted approaches produce interaction maps that are subsets of the interaction maps generated by untargeted approaches.

also p12: If the enrichment for RNAPII is similar across Hi-C, I wonder why RNAP-specific effects on contacts were prominently detected by Micro-C (Zhang, Nat Genet 2023; Barshad, Nat Genet 2023) but not (Capture)Hi-C (e.g., El Khattabi, Cell 2019).

The reviewer raises an important issue that, in our view, directly illustrates that different methods capture the same features in quantitatively distinct ways. The extensive benchmarking we performed here showed this phenomenon more generally. We do not know “why” RNAP-specific effects can be detected so well in Zhang et al, but not in El Khattabi et al. It may relate to differences in capture radius for Micro-C (which uses double cross-linking) vs. Hi-C (using only formaldehyde).

We also address this specific issue below in response to a related comment by this reviewer.

p13: Please clarify what is meant by “dynamic range”.

We apologize for being not clear about this. In this context, dynamic range refers to the range between the lowest and highest interaction frequency detected in a given dataset.

p14: Do the authors have any ideas why compartments are stronger with Hi-C than with Micro-C and whether this could have something to do with Micro-C being biased for the A compartment? Also, what could drive differences in compartment size?

We do not have insights into this phenomenon. It may be related to the fact that contact radius for Hi-C/HindIII may be larger than for Micro-C, which would suggest compartmental interactions are less proximal than for instance cohesin-mediated looping interactions. We are aware of several modeling groups attempting to test such hypotheses.

p15: It would be great to have some mechanistic explanation for these differences that could be

used to potentially improve the protocols detecting weaker effects (if it is desirable), particularly Micro-C which is rapidly gaining popularity - at least speculatively, but potentially also with some experimental evidence. also perhaps some recommendations on what method to use and when?

Any mechanistic explanations would be highly speculative, here our goal is to give readers a guide to what aspects of chromatin architecture may be enhanced with each method.

This issue again shows the value of the comparison of data obtained with different methods for the same cell samples in a single manuscript: with further modeling this may reveal new insights into why these assays differ quantitatively in their ability to detect particular features of the 4D nucleome, and this in turn may provide new insights into the molecular nature of the structures themselves. We believe this is a major contribution of this manuscript.

In the revised manuscript we added a “user guide” in the form of a table and decision tree with our recommendations for use of specific assays for addressing specific research questions (Supplemental Figure 12).

p18-23. The loop anchor section is informative and useful but gets somewhat lost in detail. Can it be further digested to be more readable?

We have edited this section and hope it is more focused now.

p18: Did I get it right that clustering was done based on the two UMAP dimensions rather than the underlying multidimensional space (e.g., PCA)? This is generally not recommended.

We thank the reviewer for raising this important point. The reviewer is correct that in our original submission, clustering was performed based on the two UMAP dimensions. We agree that clustering solely in such a low-dimensional space can be problematic, as it may overly compress the underlying structure of the data.

To address this concern, we have now re-performed the clustering analysis using the original multidimensional feature space after applying Principal Component Analysis (PCA). Specifically, PCA was conducted on the original feature matrix (shape: $N \times 22$, where N is the number of non-redundant loops in H1-hESC or HFFc6), resulting in a transformed matrix of the same shape. We then constructed multiple k -nearest neighbor graphs using varying numbers of neighbors and applied the Leiden algorithm for community detection, with a resolution parameter set to 0.5. Consensus clusters were subsequently derived by integrating clustering results across all graphs.

As shown in the revised Figure 3b, although the updated clustering results are not identical to those obtained using the original UMAP-based approach, the same number of clusters were identified, and our main conclusions remain largely unchanged.

b Clustering of chromatin loops in H1ESC
(n = 141,365)

Figure 3b. UMAP projection and clustering of 141,365 union loops in H1-hESC, based on the ChromHMM state composition at their loop anchors. Cluster IDs and the number of loops per cluster are labeled on the UMAP.

p19, Figure 3e: It is quite hard to see differences between plots except for Pol2 ChIA-PET, and if this is the only point you're making perhaps it doesn't merit a main figure.

We thank the reviewer for this helpful suggestion. In response, we have moved the original UMAP-based plot to Extended Data Figure 3f. To improve clarity, we have replaced Figure 3e with a stacked bar plot that quantifies the proportion of loops from each cluster among those detected by each platform.

Figure 3e. Cluster composition of loops detected by each platform.

p22: I'm not sure I fully got this point: "Loops in the fourth and sixth clusters are of particular interest given that they are less enriched for RNA polymerase II, but are highly enriched for cohesin binding, which suggests that cohesin might be important in mediating enhancer-promoter loops (note that these clusters are enriched with interactions between enhancers and promoters based on Figure 3d)."

We thank the reviewer for pointing out the ambiguity in this sentence. In the revised manuscript, we have clarified the description as follows: "Enhancer-promoter loops in cluster 4 and transcriptional elongation-related loops in cluster 6 showed lower POLR2A enrichment but retained strong cohesin binding, suggesting a role for cohesin in mediating these regulatory interactions."

p22: "Interestingly, although both Hi-C and Micro-C are designed to detect all-to-all chromatin interactions without bias, loops detected with these methods are less enriched with transcription-related loops. These assays appear to excel in detecting structural CTCF-anchored loops (Figure 3f-g)." I think it's important to make a slightly more nuanced point here, given that Micro-C was used to demonstrate transcription-dependent loops and Hi-C and Capture Hi-C cohesin/CTCF-independent E-P contacts.

We thank the reviewer for the suggestion. In the revised manuscript, we have rephrased the sentence as follows:

"Hi-C and Micro-C offer a relatively unbiased view of chromatin interactions and detect both insulator- and transcription-related loops (Figure 3e,f). However, these methods appear less sensitive to certain transcription-related loops compared to Pol2 ChIA-PET and H3K4me3 PLAC-Seq, as they fail to detect key enhancer-promoter interactions, including those associated with genes essential for maintaining the embryonic stem cell state in H1ESC (Figure 3g and Supplemental Figure 4)."

We do want to make clear that our original statement is not in any way inconsistent with the published work that used Micro-C to demonstrate transcription-dependent loops (Zhang et al. 2023), or work that showed that Hi-C and Capture Hi-C can detect cohesin/CTCF-independent E-P contacts: the key point we make is that these assays are just less efficient in detecting such interactions that are typically much less frequent than cohesin/CTCF-dependent contacts.

p26-27: the SPIN framework result, as presented, is somewhat underwhelming. It would be good for the authors to explain in what way this annotation is more useful than histone mark-derived states (if they believe so) and/or what can we learn from the correlation between these two annotations. Do SPIN states have stronger associations with caRNAs than histone marks, for example? It would also be great to explain more clearly why SPIN is more useful/relevant for the 3D analyses that follow compared to other epi- and 3D genome annotations. One possibility is that this is simply a better annotation as it provides some 3D structural anchoring - but for this, it would be good to have a better idea of how confident the SPIN state assignments to various microscopic subnuclear structures are.

Thank you for the thoughtful comment. SPIN provides a complementary perspective to chromatin state annotations such as ChromHMM by integrating nuclear structure mapping data (TSA-seq and DamID) with 3D genome organization data (Hi-C). While ChromHMM captures local chromatin states based on histone modification patterns at a resolution of two hundreds base pairs, SPIN infers spatial nuclear compartmentalization patterns that are anchored to subnuclear bodies, including the nuclear lamina, nucleoli, and nuclear speckles together with interactions derived from Hi-C data at 25 kb resolution. Such spatial context cannot be captured by histone-based annotations using a hidden Markov model and is particularly relevant for understanding higher-order genome organization.

As shown in our original publication (Wang et al. Genome Biol 2021), the SPIN states are strongly associated with distinct patterns of transcriptional activity, histone modifications, replication timing, and nuclear subcompartment organization. These results demonstrate that SPIN provides a spatially informed stratification of genome functional annotation that complements chromatin states derived from histone marks. While SPIN does not incorporate imaging data directly here, we showed previously that the inferred states correspond closely to known nuclear compartments validated by microscopy-based methods (Wang et al. Genome Biol 2021) in K562.

For instance, the figure panel below shows a comparison between the identified SPIN states and examples of microscopy imaging of DNA FISH on corresponding regions. DNA FISH probe imaging data were obtained from Chen et al (J Cell Biol (2018) 217 (11): 4025–4048). This figure was listed as Figure 1e in our original publication (Wang et al. Genome Biol 2021).

The following table shows the complete comparisons of SPIN states with measurement of more DNA FISH probes provided in Chen et al.

BAC	Position (hg38)	Mean distance to speckles (um)	SPIN state
RP11-634L10	chr17:81,838,938-82,011,416	0.09	Speckle
RP11-479I13	chr6:31,726,513-31,941,166	0.11	Speckle

RP11-264N5	chr7:100,470,711-100,665,335	0.16	Speckle
RP11-302K17	chr10:102,058,243-102,216,676	0.25	Speckle
RP11-246J15	chr1:202,111,934-202,271,376	0.36	Interior_Act1
RP11-1058N17	chr18:48,801,892-48,998,008	0.47	Interior_Act1
CTD-3244P16	chr10:102,990,563-103,165,047	0.47	Interior_Act2
CTD-3106L12	chr2:24,775,316-24,986,883	0.5	Interior_Act2
CTD-2503D10	chr10:103,950,392-104,169,255	0.51	Interior_Act3
RP11-729K13	chr2:30,387,389-30,582,344	0.6	Interior_Act3
RP11-531I9	chr6:23,302,020-23,446,223	0.76	Lamina
RP11-997B19	chr17:71,701,963-71,881,247	0.81	Near_Lm2
RP11-543G21	chr1:199,421,726-199,594,860	0.92	Lamina
RP11-978O5	chr2:22,703,019-22,897,141	0.97	Lamina
RP11-1047B3	chr7:114,496,092-114,692,886	0.97	Lamina
RP11-846O11	chr18:41,032,946-41,237,107	0.98	Lamina

Furthermore, Dr. Andrew Belmont’s lab has performed extensive comparison of TSA-seq data and MERFISH data (Su et al. Cell 2020) targeting nuclear speckles, nuclear lamina, and nucleolus in HFFc6 cells (Gholamalamdari et al. Elife 2025 and Kumar, et al. Communications Biology 2025), showing strong correlation between omics readout and microscopy measurements.

Additionally, we have clarified the insights gained from the integrative analysis of SPIN and caRNA profiles: “Thus, sequence features of caRNAs are correlated with the 3D compartmentalization of their target genomic regions.”

Crucially, while the original SPIN study focused on K562 cells, in this work we extend the framework to multiple human cell types using new TSA-seq, DamID, and complementary datasets. This extension demonstrates the robustness and generalizability of SPIN as a large-scale chromatin state annotation framework for 3D genome analysis.

p28-35: Given the rise of imaging-based technologies, it would have been great to see some validation of the structural findings with direct measurements from high-resolution microscopy.

We appreciate the reviewer’s comment on the value of validating structural models with high-resolution imaging. While targeted imaging for H1-hESC and HFF cells is beyond the scope of

this project, which focuses on sequencing-based assays, we agree that imaging offers valuable complementary validation. To that end, we highlight that our 3D models have been validated against DNA MERFISH imaging data, as shown in Supplementary Figure 5 (already included in the original submission), demonstrating the validity and accuracy of our structural predictions. Additionally, we refer readers to companion papers within the 4DN consortium that show the general complementarity between genome structure generated from sequencing-based data and imaging data (Boninsegna et al., Nature Methods 2022, PMID: 35817938; Yildirim et al., Nature Struc. & Mol. Biol. 2024, PMID: 37580627). We also note, and now cite, the work by the Misteli group (Finn et al., Cell 2019, PMID: 30799036), which provides a comprehensive comparison of Hi-C and FISH data in HFF cells. We include these references in the revised manuscript to contextualize our findings and acknowledge the broader body of work integrating imaging and sequencing modalities.

In addition, from a reader's perspective, I wanted more clarity in the authors' message regarding bulk vs single cell-based modelling approaches. Which ones are "better" (given the sparsity challenges of scHi-C) and/or how do they complement each other?

We thank the reviewer for raising an important question regarding the relative roles of bulk versus single-cell modeling approaches. Concerning data types, the distinction between bulk and single-cell Hi-C is not a matter of superiority, but of suitability for different biological contexts. Bulk Hi-C, with its higher sequencing depth, enables more detailed detection of structural features—particularly in homogeneous cell populations. In contrast, studies involving complex tissues and heterogeneous cell populations demand single cell approaches to capture structural variability across cell types. We have added a section on single cell approaches in the discussion.

Concerning 3D modelling approaches, it is critical to account for the structural variability between individual cells when interpreting both bulk and single cell data. Our modeling approach leverages the high sequencing depth of bulk Hi-C to infer a population of structurally diverse single-cell models, rather than a population-average structure. This allows the analysis of genome structure heterogeneity at single cell resolution, while maintaining consistency with the bulk data. We also point to complementary published work that exemplifies the power of this approach in studying the impact of structural variability on gene expression (Zhan et al, 2025 PMID: 36824908) and chromatin compartmentalization (Zhan et al. 2025, PMID: 40408515). We have now clarified this point in the revised manuscript.

p34: Minor comment: I think the last sentence on the page has some textual issues.

We agree this sentence was poorly phrased. In the revised manuscript we have rewritten this section.

p36-43: The relationship between chromatin loops and the genome function is the part where I

felt the most that this study is trying to reinvent the wheel. A large number of comparative and perturbation studies, including using high-resolution flavours of Hi-C, have looked at this question very extensively and comprehensively. It is very important to make it clear what exactly the robust new finding(s)/concept(s) are from the analyses presented here - including for the authors' sake, as this will help protect them from the usual criticisms that large, government-funded genomics consortia typically attract, and often undeservedly. Perhaps focusing on house-keeping genes was one attempt at providing such a novel finding, but (a) if it's novel, why would others not see it before and (b) given the sparse nature of most techniques employed here, how much confidence should we put in the findings of "dynamics" (as opposed to sampling error)?

We thank the reviewer for this thoughtful and constructive comment. We fully agree that the functional relevance of chromatin loops has been extensively studied, and we appreciate the opportunity to clarify the specific contributions of our work in this context.

Regarding the observation that housekeeping genes are engaged in extensive and dynamic enhancer–promoter loops, we would like to clarify, and acknowledge, that this is not an entirely novel concept. However, compared to developmental or cell-type-specific genes, much less attention has been paid to the distal regulation of housekeeping genes, which are often assumed to be constitutively expressed and regulated solely by proximal promoter elements. That said, our aim is not to “reinvent the wheel,” but to build on prior work by providing a more comprehensive view of the regulatory landscape of housekeeping genes using high-resolution chromatin loops derived from multiple orthogonal 3D genomic assays. Specifically:

1. Earlier studies have shown that housekeeping genes can be regulated by enhancers using MPRA and STARR-seq (Nature 2014, PMID: 25517091; Nat Commun 2024, PMID: 39362902; eLife 2024, PMID: 39466837), but these studies examined a limited number of genes and were performed outside the native chromatin context.
2. A recent study (Genome Res 2023, PMID: 37884340) demonstrated that housekeeping genes can be associated with cell-type-specific enhancers in native chromatin. However, enhancer–promoter links were assigned based on linear proximity without incorporating chromatin looping, limiting the accuracy of inferred regulatory relationships.
3. Another study (Cell Reports 2023, PMID: 37182209) identified promoter–promoter loops involving housekeeping genes mediated by Ronin.

The paper mentioned in point 3 showed that promoter-promoter interactions, especially those dependent on Ronin, may play an important role in regulation of housekeeping genes. Our work explores all housekeeping more generally and suggests that promoter-enhancer interactions, that are often cell-type specific, also contribute to regulation of these genes. Notably, our results are not contradictory to the conclusions of Cell Reports 2023; in fact, additional analysis shows that a substantial fraction of enhancer–promoter loops involving housekeeping genes also connect to distal promoters, suggesting that these genes may be regulated through both enhancer–promoter and promoter–promoter interactions (Extended Data Fig. 7g).

Extended Data Figure 7g. Number of loop anchors linked to housekeeping genes, categorized by whether they contain enhancers, promoters, both, or neither.

Regarding the concern about the confidence in loop dynamics due to data sparsity: we agree that (some) individual assays may suffer from incomplete sampling. To mitigate this, we defined a union loop set by integrating multiple high-resolution chromatin conformation assays in both H1-hESC and HFFc6, which we believe alleviates the sparsity issue inherent to any single dataset. To further validate the observed loop dynamics, we performed Aggregate Peak Analysis (APA). As shown in Extended Data Figures 7h-i, H1-hESC-specific loops show clear contact enrichment in H1-hESC datasets but not in HFFc6, and vice versa. This supports the interpretation that the detected dynamic loops reflect true biological differences between the two cell types rather than sampling artifacts.

p.44-49: Again, I have reservations regarding the conceptual novelty of most of the presented findings, including the relationship with replication timing that was covered by David Gilbert in his lab's earlier work.

This section contains background information from recent 4DN publications (referenced as such), but all of the analyses shown are new. We have substantially cut back the section.

p53: The sequence-based 3D modelling part is rudimentary, particularly in light of the ever-expanding array of tools and approaches for deep learning-based 3D structure prediction. Given that the authors dedicated much effort to benchmarking genomics assays, would it make sense for them to at least attempt 3D structure prediction using a variety of models, if not benchmark them formally?

The reviewer makes an excellent suggestion. We agree that the section on sequence-to-structure prediction is rather small. We had included it as a more forward looking section, i.e., much current and future work is focused on developing and applying such methods. A benchmark would require retraining the models on a consistent dataset with the same training, validation, and test regions, requiring massive computing resources. In combination with the need to reduce the length of our manuscript, the reviewer's suggestion motivated us to move this point to the Discussion as a part of the future perspective of the 4D nucleome field, with the results in Extended Data Figure 10 rather than a main Figure. Given this strategic decision, we opted not to perform the proposed benchmark. Instead, we added a sentence stating that this is important future work: "It will be important to perform benchmarking studies to evaluate how well current and future predictive models can predict the effects of variants and decode sequence determinants of genome folding."

Referee #2 (Remarks to the Author):

In the paper by Dekker and colleagues, titled "An Integrated View of the Structure and Function of the Human 4D Nucleome", the authors provide a thorough description of the work conducted by numerous labs as part of the NIH's 4D Nucleome Project. The account is highly detailed and will likely be well-received by the global 4D nucleome research community. The manuscript is well-structured, with clearly outlined sections. The figures are appropriate, and the supplementary material and figures are extensive. Altogether, this work represents a monumental effort in delineating the structure and function of the genome.

We thank the reviewer for the supportive comments, and their very insightful comments below. We have worked hard to address all of them.

Overall, the findings presented in the manuscript are highly engaging for a researcher like myself, working in the field of 4D nucleomics. However, some sections are challenging to follow or overly repetitive for a non-specialized audience. I suggest that the authors simplify certain sections to make the content more accessible and appealing to readers who are less familiar with nuclear organization.

We thank the reviewer for this important suggestion. We have made attempts throughout the text to simplify where possible.

Similarly, some figures could be simplified without losing their ability to convey the main message. Conversely, the discussion section feels somewhat limited and could benefit from expansion to better contextualize and interpret the findings presented.

We are very grateful for the specific recommendation made by the reviewer. We have attempted to follow these, as outlined below.

We have expanded the discussion to include a single cell analysis section, and now include the forward looking section on predicting 3D folding from sequence.

Below, I outline additional suggestions for improvement beyond the two main points mentioned above:

- I recommend generating a narrative text for the final part of the introduction, where the authors outline the detailed analysis and integration of data.

We have rewritten the introduction. The bullet points have been removed and the text incorporated in the last paragraph. We now focus much stronger on the main novel aspect of the work which is the benchmarking and integration of the diverse datasets which goes much beyond the companion papers produced by the consortium. (The 32 companion papers for this work are listed in the new supplemental table).

- Figure 1 is beautifully crafted, but this is not the case for many of the subsequent figures. Some of these (particularly Figures 7 and 8) are overly cluttered, with excessive panels that obscure the message they aim to convey.

We have simplified the figures throughout the manuscript. We hope that this makes the key points of each figure more clear.

- The authors have provided a thorough introduction to the specificities and particularities of each experimental method used to interrogate the genome. This allows readers to understand how methods like SPRITE and GAM differ from other approaches. However, it would be beneficial to include a section that contextualizes these methods practicalities discussing when each is most appropriate and how their results should be interpreted. This addition could enrich the discussion section.

The reviewer brings up a very good point that was also raised by Reviewer 1. We agree that we have not stated explicitly any direct guidance on best assays. In the revised manuscript we added “user guide” in the form of a table and decision tree with our recommendations for use of specific assays for addressing specific research questions (new Supplemental Figure 12).

- The authors could speculate as to why there are large quantitative differences between the methods (page 15) in detecting AB compartments.

The reviewer brings up a very good point that was also raised by Reviewer 1. We do not have insights into this phenomenon. It may be related to the fact that contact radius for Hi-C/HindIII may be larger than for Micro-C, which would suggest compartmental interactions are less proximal than for instance cohesin-mediated looping interactions. We are aware of several modeling groups attempting to test such hypotheses.

Reviewer 1 raised a similar answer, and we provided this related response (see above):

“Any mechanistic explanations would be highly speculative, here our goal is to give readers a guide to what aspects of chromatin architecture may be enhanced with each method.

This issue again shows the value of the comparison of data obtained with different methods for the same cell samples in a single manuscript: with further modeling this may reveal new insights into why these assays differ quantitatively in their ability to detect particular features of the 4D nucleome, and this in turn may provide new insights into the molecular nature of the structures themselves. We believe this is a major contribution of this manuscript.”

- As indicated by the authors (page 23), Hi-C and Micro-C are unbiased when detecting interactions. However, they are less enriched when detecting transcription-related loops. Can the authors speculate why is that?

One factor that likely contributes to the lower sensitivity of genome-wide unbiased methods such as Hi-C and Micro-C for detecting transcription related loops as compared to CTCF/Cohesin mediated loops is that enhancer-promoter loops appear to be of much lower interaction frequency. We had noted this several years ago when we performed 4C and 5C: those methods achieve much higher sequence coverage for the targeted loci and are better able to detect enhancer - promoter loops (see for instance PMID: 30142347, Supplemental Figure S5 panel C: in that panel one can observe CTCF-CTCF loops and enhancer-promoter loops, with the latter being of much lower interaction frequency).

- Figure 4 seems to be incomplete as for the text as well as the figure itself.

We have reorganized the figure panels and streamlined the presentation. Specifically, the panels originally in Figure 4 have been integrated into the revised Figure 4 to better highlight the cohesive and integrative nature of our analysis, which combines SPIN state annotations with 3D structural modeling.

- Could the authors elaborate on the sentence indicating that in certain cell types, S[^]PIN states have a cell type-specific distribution of repressive histone marks such as H3K27me3 (page 26).

We thank the reviewer for raising this question. We found that H3K27me3 is more enriched in Interior_Repr1 states in HFFc6 but has a stronger association with Interior_Act3 and Speckle in H1-hESC. We think this observation can be supported by several existing observations or evidence. For example, H3K27me3 is primarily known to play a role in transcription repression while studies have found that in embryonic stem cells H3K27me3 and H3K4me3 can co-exist on bivalent promoters (PMID: 16570078). This observation is consistent with our observation of enrichment of H3K27me3 in more active chromatin (Interior_Act3 and Speckle).

On the other hand, recent studies have shown that H3K27me3 is enriched in a certain type of facultative lamina-associated domains (fLADs) in HFF and HCT116 (Gholamalamdari et al. 2025) and mouse fibroblast cell (Martin et al.

(<https://www.biorxiv.org/content/10.1101/2024.12.20.629719v1>). Gholamalamdari et al. found that these H3K27me3 enriched fLADs tend to have moderate nuclear speckle TSA-seq signals. The authors further suggested that this may be attributed to the flat shape of fibroblast cells as compared with the round shape of H1 cells. Therefore, our observation of the cell type-specific distribution of H3K27me3 might also stem from the different cell shape between H1 and HFFc6 cells.

- Can the authors quantify the “frequently” in the sentence “Differences in gene expression between H1-hESC and HFFc6 cells frequently correlated with notable differences in the nuclear microenvironment of genes”. Page 28.

We have revised the sentence to clarify the fraction of differentially expressed genes that show significant differences in their nuclear microenvironment, as follows: “Genes with large expression differences between H1- hESC and HFFc6 cells are often found in distinct nuclear microenvironments. s. For example, 74% of genes with a \log_2 -fold expression reduction > 9 in HFFc6 (FDR < 0.05) show significantly increased distances to nuclear speckles compared to H1-hESC (FDR < 0.05 ; Extended Data Figure 6g). .

- Following the text in the section “Gene expression” when discussing highlight expressed genes is difficult (pages 29-30). I would revisit it to clarify the points.

In response to the reviewer’s suggestion, we have revised the text to improve clarity.

- The entire text focuses on the works for 2 cell types for genome-wide analysis. However, in the section of integrative modelling of single-cell 3D genomes, the work is done in a different cell type and for a 2Mb region of the genome. Maybe an introductory text as to why would be very welcome.

The reviewer is correct that the bulk of this study is focused on two cell types. This allowed the consortium to obtain sufficient datasets that could be directly compared and integrated. The single cell modeling of loci was done with a iPSC line WTC-11, another cell line widely used by the consortium, and many others. We have added a statement to better introduce the use of WTC-11 cells.

- When analysing how SPIN states are embedded or not into different hierarchical organization of the genome (AB, TAD, etc...), could the authors speculate as to for some matchings do not agree (SPINs for silencing embedded in A and vice versa). Could it be related to the resolution of the data for the specific analysis? Other possible reasons explaining such contradictory results? (page 45).

As also pointed out by Rev.3, in the manuscript we did speculate that these discordant spin states may be genes that are poised for changes in gene expression/compartments/replication timing as we have shown previously to be common in stem cells. In fact supporting our

hypothesis, the Int rep1 SPIN state is enriched in genes that are differentially expressed during hPSC differentiation (unpublished). However, as pointed out by Reviewer 2, these can be issues of resolution, i.e., SPIN states being called at higher resolution and A and B compartments. Furthermore, there are variable numbers of each SPIN state found in the different compartments. Also, A/B states should not be viewed as binary: there are quantitative differences in the strength of compartment interactions at the sites where the different SPIN states reside. Since we have been asked to significantly reduce the paper length, and this is clearly a complex topic, we have decided to delete this statement. We agree with the Reviewer that this is a very interesting issue that warrants an in-depth future analysis.

- The section “predicting chromosome folding from sequence” seems not to be well integrated into the main text. Additionally, the results in Figure 9 as explained in the text are poor compared with other sections of the manuscript. How much it adds to the overall story as it is now written?

The reviewer makes an excellent comment. Reviewer 1 also mentioned this issue. We agree that the section on sequence-to-structure prediction is rather small. We had included it as a more forward looking section, i.e., much current and future work is focused on developing and applying such methods. Given space constraints, rather than elaborating this section, we have moved it to the forward-looking part of the Discussion, with the results moved to Extended Data Figure 10.

Other minor points.

- Add space between “comparing” and “the” (page 8)

This sentence has been rewritten as: “We started by comparing and benchmarking the most used chromatin interaction assays and integrating with assays that report sub-nuclear locus positions.”

- Figure 2. Panel b. I would add a label to the bar legend. There seems to be a misplacing in the labels for the x-axis.

We added a label to the color bar (“Pearson correlation”), and fixed the labels for the x-axis.

- Figure 2. Panel c. Move legend outside the plot. Right now, the dots in the legend could be mistakenly interpreted as data.

There is limited room to move the legend outside the plot. We gave the legend a white background to make it clear that it is not part of the plot.

- Figure 2 Panel d. I would add a label to the bar legend.

We added a label to the color bar (“Pearson correlation”).

- Lower case “f” in “(For” page 15).

Done.

Referee #3 (Remarks to the Author):

This manuscript from the 4D nucleome consortium brings together a range of different methods to study the 3D genome on two different cell lines (H1-hESCs) and HFF6c (a human foreskin cell line). The most exciting aspect of this enterprise is that different methods (Hi-C, Micro-C, ChIA-PET, HiChIP SPRITE and GAM) can be compared side-by-side, enabling the assessment of the merits of these methods. Comparison of various genome organisation assays is important since different papers make claims using different methods and it is not necessarily clear how suited various methods are for quantifying 3D genome features at different scales and which artifacts they present with. The authors extensively quantify how the methods differ in quantification of different features, however it remains somewhat vague which methods should practically be used for which purpose.

We appreciate the reviewer’s comments. The section that quantitates performance of a large set of assays, is a key advance of our integrated analysis. We agree that we have not stated explicitly any direct guidance on best assays. The two other reviewers also raised this issue. In the revised manuscript we added a “user guide” in the form of a table and decision tree (Supplemental Figure 12) with our recommendations for use of specific assays for addressing specific research questions.

This reviewer appreciates the amount of data that has been generated, unfortunately, the biological insight that is gleaned from this massive dataset is rather limited. In fact, many trivial observations (e.g. what a saddle plot is) are discussed at length, whereas in the second half of the paper many non-standard methods (e.g. iMARGI) are discussed without proper introduction. Currently, the manuscript feels very disjointed with the different sections not linked in any meaningful way. The manuscript could benefit from someone taking the lead and streamlining the text and making it into a more cohesive story.

We thank the reviewer for this important comment. We had spent significant effort to put this massive amount of data into a coherent story that would be interesting to read for a wide audience. Clearly, we need to further improve the narrative.

In response to the many very helpful comments by all three reviewers, we have now extensively revised the manuscript. We also shortened the text. We hope that the revised manuscript is more balanced in covering known and new approaches. We also made an attempt to better highlight new insights.

That said, this “flagship” paper is an integration of much work by the 4DN community, and the community has published some aspects of the work as “companion” papers (see our response to reviewer 1), and that obviously influences the novelty of some of the data and conclusions. The key advance here is that different datasets and approaches have been integrated and benchmarked.

In response to reviewer 1’s similar comment we wrote:

“This manuscript does not present new methods because any such methods have been presented in earlier work from this group and in companion papers (See new Supplemental Table listing all companion papers). Rather, as we now highlight in the revised manuscript, this manuscript summarizes the efforts of the consortium to integrate datasets obtained with a comprehensive set of genomic assays. We show how combining analysis of different datasets adds important insights. For example, and this is a major contribution to the field, we find that the number of loops will be severely underestimated when only a single assay, e.g., Micro-C, is used. Combining loops called with a set of different assays leads to detection of more loops and more different types of loops (e.g., structural loops and transcription-related loops). This manuscript delivers, to our knowledge, the most comprehensive set of loop calls to date. This will provide a rich resource for the field, including as a training set for efforts to predict long-range interaction for other cell lines. In the revised manuscript we highlight this in more detail.”

And in response to reviewer 1 we wrote this about the companion papers:

“In the revised manuscript we have worked hard towards emphasizing new methods and biological insights. We note that we failed to make clear that the consortium has indeed put together a large set of companion papers that have been reviewed and published separately. It has been the policy of the 4DN consortium that these papers were published on their own (and much faster) timeline than this Flagship manuscript so that work of trainees could be published without further delay that is intrinsic to publishing complex consortium flagship papers.

In the revised manuscript we now include a table listing all 32 companion papers. Most of these were published in Nature family journals, so that they could easily be linked when the Flagship would be finally accepted”.

In response to the specific comment on iMARGI: iMARGI (in-situ MApping of RNA–Genome Interactions) is a genome-wide profiling technology designed to systematically identify and map the physical interactions between RNA molecules and chromatin in their native nuclear environment. We have spelled this out more explicitly in the revised manuscript.

Below specific comments are presented, to improve the readability of the manuscript:

- Figure 2a: What are the line plots underneath the heatmap. Insulation score is assumed, but please indicate.

We thank the reviewer for pointing out this oversight. The reviewer is correct, the line plots underneath the heatmaps represent insulation scores. In the revised manuscript we have added labels to these plots (“Insulation”).

- Figure 2b has misaligned labels on x axis.

We thank the reviewer for pointing out this error, which is fixed in the revised manuscript.

- The authors state that “Visual inspection of these contact maps at large genomic distances show that ligation-based methods capture similar chromatin organization patterns, independently of whether contacts are mapped with Hi-C, Micro-C, GAM, SPRITE, or based on occupancy of specific proteins (CTCF, RNA Pol II, H3K4Me3)”. Based on the example plot in Supp. Fig. 1a GAM does not seem to capture similar patterns. Additionally, the correlation with GAM compartment scores vs others is really low.

The reviewer is correct: the GAM data has a relatively low correlation with other datatypes.

- Please plot Supp. Figure 1a the same way as Figure 2a: i.e. compartment level, TAD/loop level. This allows for a proper comparison of all methods.

This is a good suggestion. We have now made such plots and, in response to the reviewer’s next suggestion, have added them to main Figure 2.

- Please make Supp. Figure 1a a main figure. It is the only figure where all methods are (and can be) compared directly. This seems like very useful data for the community. It would be unfortunate if this ends up in the Supplement.

The reviewer makes an important point. The reason we had placed the H1-hESC data examples in Supplemental Figure 1 was that the quantitative differences in the methods, e.g., compartmental patterns, are so much more pronounced in differentiated cells as compared to pluripotent cells. Therefore we felt that showing the fibroblast data in the main Figure 2a was more compelling.

We completely agree with the reviewer that it is useful for the community to have all datasets in a main figure. Therefore we have added the H1-hESC data that was in Supplemental Figure 1a in the first submission, and shown the data in the same format as the HFFc6 data (compartment level, TAD/loop level).

- Extended Data Figure 2a: The color bars for the APAs differ greatly. Please set all the color bars to be consistent with the Hi-C/Micro-C (i.e. 1.1-4.5) with the exception of the ChIA-PET data.

We thank the reviewer for the suggestion and have revised the figure. In the updated figure (Extended Data Fig. 2a) below, we have standardized the color bar range to 1.1–7 for all

experimental platforms, except for GAM, which is set to 1–1.5 due to its lower interaction value range compared to the other methods. As shown, consistent with our original submission, high-confidence loops are enriched across all platforms. Additionally, for DNA SPRITE, interactions derived from clusters with fewer fragments show greater enrichment of chromatin loops compared to those from larger clusters.

- Please also generate APAs for the PolII loops identified by PolII ChIA-PET. It would be important to know whether these loops can be detected with the other methods or whether they can only be detected with ChIA-PET.

We thank the reviewer for the suggestion. We have generated APA plots for both total Pol2 ChIA-PET loops (Extended Data Fig. 2b) and loops uniquely detected by Pol2 ChIA-PET (Extended Data Fig. 2c). As shown, while the signals for these loops are much weaker in Hi-C, Micro-C, CTCF ChIA-PET, DNA SPRITE, and GAM, they are notably enriched in both Pol2 ChIA-PET and H3K4me3 PLAC-Seq.

- Adding a schematic to show that chromHMM was used at loop anchors to create UMAP in Figure 3C and Extended Figure 3B would be helpful to the reader.

In this revision, we have added a schematic illustrating how ChromHMM states at loop anchors were used to construct the feature matrix for UMAP projection, as shown in the new panel (Fig. 3a).

Figure 3a. Schematic illustrating the construction of the feature matrix used for UMAP projection of chromatin loops.

- In Figure 3, how do the authors choose 6 clusters in c, d, e, whereas in the text roughly 2 groups are mentioned, while the figure 3b shows 4 clusters?

We apologize for the confusion. In our original submission, two separate clustering analyses were performed:

1. Loop anchor clustering (old Figure 3b): In this analysis, we treated each loop anchor independently and assessed their ChromHMM state enrichments across 31 assay combinations (represented as 31 bars in the old Figure 3b). We applied hierarchical clustering to group these anchor sets, and depending on the dendrogram cutoffs, different numbers of clusters could be identified. In the manuscript text, we focused on two broad clusters that were robust and consistent across both H1-hESC and HFFc6 cell lines.

2. Loop-level clustering (old Figure 3c): Here, we jointly considered chromatin states from both anchors of each loop. To visualize the distribution of loops based on the chromatin state compositions of their anchors, we used UMAP to project loops into a 2D space, with each point representing a loop. Clustering in this space was initially performed using HDBSCAN, which identified 6 clusters. Following Reviewer #1's suggestion, we revised this by applying a community detection-based clustering algorithm directly on the multidimensional feature space, which again yielded a similar 6-cluster structure (**new Figure 3b**).

We agree that presenting both clustering results in the same section may have led to confusion. To improve clarity, we have removed the anchor-set clustering from the revision and now focus solely on the loop-level clustering, which captures finer distinctions among loop types.

- In Figure 3c, what is the point of the UMAP? If it is to quantify the separation between clusters, the authors should rather show more quantitative clustering metrics (e.g. proportions of k-nearest neighbours belonging to the same cluster; silhouette widths etc.) instead of a 2D embedding.

We thank the reviewer for the comment. We did not intend to use UMAP (old Figure 3c and current Figure 3b) to quantify the separation between loop clusters. Rather, UMAP was applied purely as a visualization tool to project high-dimensional loop features into two dimensions. Each loop is represented by a 22-dimensional feature vector (described in the Methods), and UMAP helps illustrate how loops are distributed based on the chromatin state compositions of their anchors.

- In Figure 3e, it is not clear what the colour encodes – signal strength in each loop? If so, it is hard to get any sort of quantitative insight of loop strengths in each cluster from a 2D embedding, a better alternative might be to visualize the distribution of loop strengths for each method in each cluster.

We apologize for the confusion. In Figure 3e (now Extended Data Figure 3f), the color intensity does **not** represent the signal strength of individual loops. Instead, the plot was generated as follows: for a given platform, we marked each loop detected by that platform as a red dot on the UMAP (each dot represents one loop). Loops not detected by that platform remain white, forming the background. Thus, the color intensity in each panel generally reflects the density of detected loops from that platform across the UMAP. For example, if cluster #1 appears more intensely red than cluster #2 for a given platform, it suggests that a higher proportion of loops detected by that platform fall into cluster #1 compared to cluster #2.

- Is there a more quantitative way of presenting the data in Figure 3e? The highlighted UMAP projections are not very easy to interpret in this particular case.

We thank the reviewer for the suggestion. In this revision, we have moved the original UMAP-based plot to Extended Data Figure 3f. In the main figure, we have replaced it with a stacked bar plot (see revised Figure 3e below) that quantifies the proportion of loops from each cluster among those detected by each platform.

We believe this updated visualization more clearly supports the following observations:

1. CTCF ChIA-PET predominantly captures insulator-related loops (cluster #2; see Figure 3c for cluster annotations).

2. Pol2 ChIA-PET and H3K4me3 PLAC-Seq are enriched for transcription-related loops, particularly those involving active promoters (clusters #3 and #5), but are relatively depleted for insulator-related loops.
3. H3K4me3 PLAC-Seq shows stronger enrichment for loops involving poised promoters (cluster #1) compared to Pol2 ChIA-PET.
4. Hi-C and Micro-C provide broader coverage across loop categories but tend to preferentially detect insulator-related loops, with relatively lower enrichment for transcription-related loops.

We hope this revised figure now more clearly illustrates the differences in loop detection preferences across platforms.

- In Figure 3g, transcription-related loops are not convincing. Loop calling in Pol2 ChIA-PET data in the provided example locus seems overly zealous and could easily be false positives, which are enriched overall at active gene promoters only because that is where Pol2 is localized so that is where any non-specific signal is found. See above regarding comments on APAs for Micro-C.

In this revision, we have replaced Figure 3g with a different region, which we believe is a more convincing example. Please see our response to the next comment for further details.

To demonstrate that the Pol2 ChIA-PET loops reported in this study are not simply non-specific signals at Pol2-enriched regions, we performed APA analyses across different platforms for: 1) chromatin loops uniquely detected by Pol2 ChIA-PET; 2) a set of control loops that share one anchor with a Pol2 loop but link to loci at the same genomic distance on the opposite side.

The results are shown in Extended Data Figures 2c-d. Although the Pol2 ChIA-PET loops are uniquely detected by this platform, they show notable enrichment in H3K4me3 PLAC-Seq and slight enrichment in other platforms (Micro-C, Hi-C). In contrast, random control loops exhibit

little to no enrichment across all platforms, highlighting the specificity of Pol2 ChIA-PET loops. Therefore, we conclude that the chromatin loops detected by Pol2 ChIA-PET in this study are not merely non-specific signals at Pol2 binding regions but are instead enriched in transcription-related interactions.

- Is the locus shown in Figure 3g the best representation of transcription-related loops? For example, would it be clearer to select a region where there are loops detected by both ChIA-PET and PLAC-seq with clear overlap?

We have replaced the previous example with a different region in H1-hESC cells. As shown in Figure 3g below, this region compasses the SOX2 gene, a key regulator of pluripotency in embryonic stem cells. Both Pol2 ChIA-PET and H3K4me3 PLAC-Seq detect chromatin loops linking SOX2 to two distal enhancers (marked by H3K27ac peaks and highlighted in yellow shaded bars). However, these loops are not discernible in other platforms.

Figure 3g. An example illustrating differences among platforms in detecting transcription-related loops. Contact maps are plotted at 1 kb resolution, with chromatin loops detected by each platform marked by blue circles. Loops linking the SOX2 gene to distal enhancers are indicated by black arrows, and the interacting distal enhancers are highlighted with yellow-shaded bars.

It is worth noting that while both Pol2 ChIA-PET and H3K4me3 PLAC-Seq are enriched for transcription-related loops, their interaction patterns differ significantly. Additional examples can be found in Supplementary Figure 4.

Supplementary Figure 4. Two additional examples illustrating differences among platforms in detecting transcription-related loops. Contact maps are plotted at 1 kb resolution, with chromatin loops

detected by each platform marked by blue circles. Loops linking the indicated genes (POU5F1 in the first example and NANOG in the second) to distal enhancers are indicated by black arrows, and the interacting distal enhancers are highlighted with yellow-shaded bars.

- The authors state that “Unbiased genome-wide assays such as Hi-C and Micro-C most efficiently capture CTCF/cohesin based loops, while assays targeting RNA Pol II or H3K4me3 capture more transcription-related loops”. In Ext. Data Figure 4b, it appears that clusters 3-6 have sizeable fractions of loops which are not CTCF-bound. Perhaps it would be of interest to explore the differences in epigenetic states and binding of transcription factors in CTCF-bound vs unbound loops in these categories.

We thank the reviewer for the suggestion. As suggested, we calculated the fold-enrichment scores of different ChromHMM states and transcription factors (TFs) at CTCF-bound and CTCF-unbound loop anchors within clusters 3–6 in H1ESC cells. We observed the following: (1) Although both CTCF-bound and CTCF-unbound loop anchors are enriched for similar chromatin states, CTCF-unbound anchors show slightly higher enrichment for active chromatin states (active promoters and transcriptional transition and elongation) and relative depletion of inactive states (Extended Data Fig. 4c). (2) Similarly, while both groups are enriched for many of the same TFs, the relative enrichment levels differ between them (Extended Data Fig. 4d). For example, in cluster #4, which corresponds to enhancer-promoter loops, BCL11A, CHD1, CHD7, POLR2A, and POU5F1 are preferentially enriched at CTCF-unbound loop anchors, suggesting that these factors may play a role in mediating enhancer-promoter interactions independently of CTCF in H1ESC.

- The authors state: “We further characterized loops using additional transcription factor binding data and found that loops from different clusters are enriched for distinct transcription factors (Extended Data Figure 4; Supplemental Methods). Notably, Polycomb-group (PcG) proteins, such as EZH2 and RNF2, are specifically enriched at loop anchors in the first cluster.” Polycomb group proteins are not transcription factors, please reword.

We thank the reviewer for pointing this out. Below is the reworded version of the sentence.

“We further characterized loops using additional transcription factor (TF) and chromatin regulator binding data and found different loop clusters are enriched for distinct sets of regulatory proteins (Extended Data Figs. 4a-b; Supplemental Methods). Notably, Polycomb-group (PcG) proteins, such as EZH2 and RNF2, are specifically enriched at loop anchors in the first cluster.”

- The cell types used for SPIN modelling are not clear. Supplementary Figure 4B mentions K562 comparison of previous versus joint K562 states, while the text and methods only mention SPIN modelling for H1-hESC and HFFc6. Could the authors clarify this part?

We apologize for not being clear on this. In this work, we used a modified SPIN method to perform joint calling across four cell types (H1, HFFc6, K562, and HCT116). Extended Data Figure 5b compares the SPIN states identified in K562 using the new approach with those from our previous publication (Wang et al. Genome Biology 2021). Notably, in the previous version of SPIN, we used DamID mapping data targeting a nucleolar peptide repeat (4xAP3) to indicate putative nucleolus interaction. In this work, we utilized TSA-seq mapping data targeting the granular component of the nucleolus using the antibody against MKI67IP. We have modified the Methods section to clarify this.

- It is unclear how Figure 4 fits with the rest of the manuscript. The SPIN model and its relation with histone marks and genomic structures has already been published as referenced. caRNA data analysis in this context is novel but does not provide much biological insight. The flanking box plots are confusing as they do not refer to anything. This is perhaps better suited for extended/supplementary material.

We thank the reviewer for the helpful comments. In response, we have clarified the insights gained from our integrative analysis of SPIN and caRNA profiles by explicitly stating: "Thus, sequence features of caRNAs are correlated with the 3D compartmentalization of their target genomic regions." This clarification highlights how spatial genome organization, as defined by SPIN states, relates to the localization of caRNAs and their potential regulatory roles.

Importantly, while the original SPIN study focused on K562 cells, in this work we extend the framework to multiple human cell types using newly generated TSA-seq, DamID, and other complementary datasets. This expansion demonstrates the robustness and generalizability of SPIN as a large-scale chromatin state annotation framework for 3D genome analysis.

To further emphasize the integrative nature of our approach, we have reorganized and streamlined the relevant figure panels. Specifically, the panels originally in Figure 4 have been incorporated into the original Figure 5 (now Figure 4 in the revised manuscript), presenting a more cohesive visualization of how SPIN state annotations align with 3D structural modeling.

- The reviewer appreciates the analyses of the relationship between 3D structure and gene expression presented in Figure 5 and extended Figure 5. It would be of great benefit to the reader if the authors could provide an example of class I and class II genes with corresponding Hi-C maps, genomic tracks of histone modifications, TSA-seq and Dam-ID to make it even clearer to the reader that these are inherently different in terms of their genome architectural properties.

Following the reviewer's request we now include representative examples of genes from each class (class I: LMNA and class II: FBN2 class) along with their corresponding Hi-C maps, and genomic tracks of histone modifications (H3K4me1, H3K4me3, H3K27ac), SON-TSA-seq and Lamina Dam-ID signals and RNA-seq signal tracks (Extended Figure 6n in the revised manuscript).

As expected, the Class I gene LMNA shows significantly higher SON-TSA-seq signals and lower Lamin B1 DamID signals compared to the Class II gene FBN2, reflecting its closer proximity to nuclear speckles (Extended Figure 6n in the revised manuscript). Striking differences are also observed in their Hi-C contact patterns: LMNA exhibits higher contact frequencies at short genomic distances, whereas FBN2 displays reduced local interactions but markedly stronger ultra-long-range contacts, confirming our observations for promoter-enhancer interactions.

Extended Data Figure 6n. Genome architecture surrounding the transcription start sites (TSS) of representative Class I and Class II genes. Left panel, Class I gene LMNA. Right panel, Class II gene FBN2. From top to bottom, tracks display: Hi-C contact maps at 10kb resolution (observed/expected KR normalized Hi-C contact frequencies), SON-TSA-seq signal (proximity to nuclear speckles), Lamina B1 DamID-seq (proximity to nuclear lamina), H3K4me1 (enhancer-associated mark), H3K4me3 (active promoter mark), H3K27ac (active enhancer/promoter mark), RNA-seq signal on the plus strand, RNA-seq signal on the minus strand.

To further illustrate the differences in long-range promoter interactions between the two gene classes, we have added a new figure panel (Figure 4i) in the revised manuscript. This panel presents a pile-up analysis of Hi-C contact frequencies around aligned Class I and Class II promoters, revealing that Class II promoters show enhanced ultra-long-range interactions, but reduced contact frequencies at shorter genomic distances compared to Class I promoters.

Figure 4i. Hi-C contact frequency near transcription start sites (TSS) of Class I and Class II genes, derived from Hi-C maps at 10kb resolution. Top panel, average Hi-C contact frequency profiles (10kb bins) centered on TSS of Class I and Class II genes. Middle panel, heatmap of Hi-C contact frequencies around individual Class I gene TSSs (rows: genes; columns: genomic distance from TSS). Bottom panel, heatmap of Hi-C contact frequencies around individual Class II gene TSSs (rows: genes, columns: genomic distance from TSS).

- In Figure 5c, it is not stated what NAF abbreviation stands for.

We thank the reviewer for pointing out the missing definition. We now define “NAF” as “nucleoli association frequency” in the Figure 4 caption of the revised manuscript.

- In Figure 5i and 5j, it is not clear what the colour scale encodes and what information does a 2D embedding provide.

As requested by the reviewer, we have added a color scale to Figure 5i and 5j. The t-SNE 2D embedding in Figure 5i highlights variations in structural features across genes with different expression levels. While the majority of genes within each expression class cluster in similar regions distinct from other classes, a substantial fraction overlaps with other classes, indicating shared nuclear microenvironments. We have now moved Figure 5i to the extended Figure 6 as panel “I”.

The 2D embedding in Figure 5J (now Figure 4f in the revised manuscript) demonstrates a clear separation between Class I and Class II genes based on their structural features, indicating distinct nuclear microenvironments. We have clarified this point further in the revised manuscript.

- In Figure 5j, how is it possible that the bottom quartile of the genes by SAF contains a different number of genes than the top SAF quartile? Were the genes first divided into class 1 and class 2? It should be made clearer how these subsets of genes were selected.

As suggested by the reviewer, we have clarified the selection criteria in the Methods Section. All 200 kb chromatin regions, including gene-free regions, were first divided into quartiles based on their SAF values. We then classified the top 25% most highly expressed genes into 2 classes according to the SAF quartiles of their associated regions.

- It is unclear what the biological rationale for normalizing the number of enhancers to the number of nearby genes (i.e. enhancer density) is, since enhancers are not exclusive to single genes. This could be clarified more.

We agree with the reviewer. Following the reviewer’s suggestion, we now provide our analysis in Figure 5 without normalizing enhancers to the number of nearby genes (enhancer density). We removed the fold enrichment plots of enhancer densities (Figure 5l in the previous version of the manuscript) and replaced them in the revised manuscript with violin plots showing properties of the absolute spatial enhancer density within a distance of 350 nm (Figure 4h). We have revised the text accordingly.

Figure 4h. Spatial enhancer density around transcription start sites (TSS) of class I and class II genes. Left panel. Distribution of the number of intra-chromosomal enhancers within 350 nm of each gene's TSS. Middle panel, distribution of the fraction of long-range (>1 Mb) intra-chromosomal enhancers within 350 nm of each TSS. Right panel, distribution of the number of inter-chromosomal enhancers within 350nm of each TSS.

- The analyses of gene expression in relation to nuclear positions of genes in Figure 5 and Ext. Data Figure 5 contain interesting observations. Perhaps it would make sense to implement interesting examples of genes in Ext. Data Figure 5g and 5h into the main figure and move Figure 5d and 5f-g to extended/supplementary material. This would put more focus onto biological insights one can gain from these data.

As requested we have moved Extended Figure panels 5g and 5h into revised Figure 4 (now shown as Figure 4d and e in the revised manuscript). Conversely, Figure 5a, 5b, 5c, 5d, 5e, 5g, 5i have been moved to the Extended Data Figure 6 in the revised manuscript (now as panels b, c, d, e, j, g, l). We believe that Figure panel 5f in the original manuscript should remain in the main figures, as it effectively illustrates the substantial global structural differences between the two cell types (now Figure 4b in the revised manuscript).

Revised Figure 5, now Figure 4:

Revised Extended Data Figure 6:

Added new Supplemental Figure 6:

- Regarding scHi-C imputation with Higashi and the SBS polymer model, it is not visually apparent that the insulation scores in Figure 6a are concordant between the two models. Same for the contact strength matrices in 6b. Please quantify.

We thank the reviewer for raising this concern and have quantified the similarities between the contact maps and derived insulation scores from the polymer model and Higashi imputation. The Pearson correlation between the average insulation score derived from Higashi imputation and that derived from the SBS polymer models is 0.72. For comparison, the correlation between the average insulation score from Hi-C and that from bulk Hi-C is 0.92, while the correlation between the average insulation score from the SBS polymer models and that from bulk Hi-C is 0.83. The three Higashi–SBS polymer model contact map pairs showcased in Fig. 6b have distance-stratified similarity scores of 0.69, 0.64, and 0.77, respectively.

Moreover, we also compared the bulk level contact map as shown in the following Figure 5a.

Legend Figure 5

Cell-to-cell variabilities of 3D genome features.

a. Heatmaps on the top show the merged sChI-C contact maps at a 2Mb region from chr3 imputed by Higashi or predicted by the SBS polymer model (left), as compared with the contact map from bulk Hi-C and raw contact map from

scHi-C without imputation. Insulation scores from bulk Hi-C, calculated insulation scores after Higashi imputation and SBS polymer modeling are shown at the bottom-left. The heatmap on the bottom-right shows the Spearman correlation coefficients between these contact frequency maps.

b. 3D genome structure models, raw scHi-C contact cap, and the imputed contact map from three mutually similar cells between Higashi imputation and SBS model are shown. The three Higashi–SBS polymer model contact map pairs from left to right have distance-stratified similarity scores of 0.69, 0.64, and 0.77, respectively.

c. The average normalized intensity of chromatin loop across 188 WTC-11 cells is calculated and compared by dividing loops based on their relative position within TADs and A/B compartments. The pink boxplot (left) represents the difference between loops in the same TAD and loops spanning multiple TADs. The blue boxplot (right) shows the difference between loops in the same A/B compartment and loops spanning different compartments. A representative chromatin loop near gene RABGAP1L is highlighted in the box plot on the right. The original distribution of the normalized intensity of this specific loop in each cell is shown in the box plots on the right. Loops are stratified into different groups depending on whether this loop locates within one TAD or spans TADs (top) or the A/B compartments state of two loop anchors in each single cell.

d. The heatmap shows the aggregated contact map from single-cell Hi-C data at the gene RABGAP1L locus (for cells with z-score>1.96). The circle with dashed line indicates the 450 kb loop identified by SnapHi-C.

e. KR-normalized Hi-C contact frequency from WTC11 bulk Hi-C data in the same region at gene RABGAP1L locus.

- The authors should provide Hi-C maps for the RABGAP1L loop mentioned in the boxplot of Figure 6c?

We thank the reviewer for the suggestion. We have included a panel in the new Figure 5d-4 (see above, also shown in the previous response) to show the contact map in aggregated single-cell as well as bulk Hi-C data.

- Figure 6d is mentioned in the text but not in the figures.

We apologize for the confusion. The correct text should be “the bottom-right boxplot in Figure 6c (Now Figure 5c in the revised manuscript)”.

- It is not clear how the p-values in Figure 6c are calculated and are potentially influenced by pseudoreplication if each cell is treated as an independent replicate (e.g. for scRNA data see <https://www.nature.com/articles/s41467-021-21038-1>, <https://www.nature.com/articles/s41467-022-35519-4>, <https://www.nature.com/articles/s41467-021-25960-2>).

The p-value is calculated using a t-test. We have modified the figure caption to clarify this. We agree with the reviewer that treating cells as independent samples can be potentially problematic when they originate from distinct biological replicates, due to potential pseudoreplication as shown in scRNA-seq studies involving multiple samples. However, in our study, all profiled single cells were derived from a single biological replicate (a single clone). Therefore, the assumption of independence between cells is reasonable within this context.

- The hypothesis that promoters of active genes near LADs locally carry active marks and may need to loop out of this environment for activation is very interesting (Figure 7). Given the amount of chromatin interaction data generated by the authors, would it be possible to perform quantitative analysis of the interactions of these genes? For instance, the authors could compare these to a set of control genes from similar environment to show that active genes near LADs have more of such interactions.

We thank the reviewer for the suggestion. As shown in Extended Data Figure 8d, in both H1-hESC and HFFc6, active genes (TPM > 1) located near the Lamina exhibit significantly more interactions with distal enhancers compared to control inactive genes (TPM < 1) in the same environment.

Extended Data Figure 8d. Comparison of the number of linked enhancers between active and inactive genes in the Lamina SPIN state. P values were calculated using a two-sided Mann–Whitney U test.

- Figure 8 and the related text present interesting findings on the association of SPIN states to gene expression in relation to the genomic compartment context that these SPIN states are embedded in. Particularly interesting is the finding that certain SPIN states seem to “bookmark” particular sets of genes irrespectively of which compartment they are in. The authors hypothesise that these could be hESC domains that are poised to switch on during differentiation. Perhaps it would be useful to quantify whether indeed these domains are enriched for genes that need to be switched on during differentiation to establish proper transcriptional programmes. Additionally, since this part of the paper further explores relation of nuclear positioning to gene expression, it would perhaps better fit after Figure 5 to maintain the flow and readability of the paper.

Thank you for pushing us to quantify the enrichment of differentially expressed genes with SPIN states. In fact, supporting our hypothesis, the Int rep1 SPIN state is enriched in genes that are differentially expressed during hPSC differentiation. Interestingly, the Int rep 2 SPIN state was not. However, as pointed out by Reviewer 2, these can be issues of resolution, i.e., SPIN states

being called at higher resolution and A and B compartments. Furthermore, there are variable numbers of each SPIN state found in the different compartments. Also, A/B states should not be viewed as binary: there are quantitative differences in the strength of compartment interactions at the sites where the different SPIN states reside. Since we have been asked to significantly reduce the paper length, and this is clearly a complex topic, we have decided to delete this statement. We agree with the Reviewer that this is a very interesting issue that warrants an in-depth future analysis.

For the last comment: "As for the flow between Figure 5 (current Figure 4) to Figure 8, collectively we found that moving these sections would disrupt the flow of other parts of the manuscript. Instead, we have re-written the sections referring to Figure 5 (current Figure 4) and 8 to remove redundancy, revised extensively Figure 5 (current Figure 4) and 8 and referred back to the panel in Figure 5 (current Figure 4) that is relevant to the discussion of Figure 8."

- Figures 8f-h do not seem to be referenced anywhere in the text.

We have extensively revised this figure, currently figure 7 in the revised manuscript. We have extensively revised this entire section, and hope to now have fully referenced the panels.

- What is 3DNetMod? It is not introduced in the text and does not seem to be referenced. Please explain.

We apologize for this omission. This is a previously published approach (PMID: 29334377). We fixed this in the revised manuscript.

- The data on predicting chromatin structure from DNA sequence is very exciting. Regarding the trained models, is it possible to extract the DNA motifs that these models have learned to understand which are important for establishing chromatin interactions? For example, would these models learn cell-type-specific TF motifs in different cell types (such as POU2F1::SOX2 and FOXL1::JUND in H1-hESC and HFFc6, shown in Figure 9B)?

We thank the reviewer for their enthusiasm about the section on predicting chromatin structure from sequence. In light of space constraints and comments from the other reviewers, we have moved this section to the supplement and discuss it as a forward-looking theme rather than a major section of the manuscript. As such, we have incorporated all of Reviewer 3's following suggestions, but we chose not to investigate model interpretation techniques and instead mention this as an exciting future direction.

- In Figure 9a, the Akita model seems to nicely capture the effect of the deletion in the TAL1 locus in the HFFc6 model. Does this locus show different 3D genome organisation and gene expression in hESC? If so, what does the prediction look like for the hESC model?

From interaction maps derived from the experimental Micro-C data, we observed differences in 3D genome organization at this locus, including stronger insulation of TAD boundaries upstream of the *TAL1* gene. We chose to show results from the HFFc6 model, because it has higher read depth (5.07 billion reads for HFFc6 vs. 2.79 billion reads for H1-hESC) and hence higher resolution interactions. While we did not show it, the H1-hESC model also captured the gained chromatin interactions near the *TAL1* locus following the deletion, albeit with less pronounced changes in interaction frequencies compared to the HFFc6 model. This could be real biology, but is also likely due to the differences in the training data quality (in the original Akita manuscript, model performance was correlated with read depth). Since this is now Extended Data Figure 10, we prefer not to add the H1-hESC panels, but we did add text to the figure legend stating that a similar change was seen in models from both cell lines. Regarding gene regulatory effects of the deletion, *TAL1* is not expressed in either H1-hESC or HFFc6 cells.

- In Figure 9b, the effect size appears to be tiny. If this is the best example of a prediction in hESC, this approach seems to be of limited usefulness. This should be discussed.

In our experience, effect sizes predicted by deep learning models tend to underestimate experimentally measured effects. This means that *in silico* mutagenesis is conservative (i.e., misses some true effects, while having a very low false positive rate). We added text to discuss this caveat in the Discussion: “One caveat that future work needs to address is the fact that deep learning models tend to under-estimate the true effects of genetic changes.”

- The authors state: “We found that the H1-hESC model is more sensitive to mutation of motifs for the embryonic stem cell factors POU2F1::SOX2, while the HFFc6 model is more sensitive to the fibroblast factors FOSL1::JUND.”. This is not shown or quantified anywhere, only the two examples are shown in Figure 9.

Other reviewer and editor comments pushed us to shorten the Results section and the overall length of the manuscript, as well as the figure count. To address these other comments, the analysis of motif perturbations using cell type specific Akita models was moved to the Discussion, with the results now in Extended Data Figure 10, rather than a main Figure. To respect the need to shorten, rather than lengthen, the manuscript, we did not add the quantification of model sensitivity to motifs as suggested by this reviewer comment.

Textual comments:

In the discussion, the authors state that “Looping interactions with distal enhancers is strongly correlated with gene expression”. This is too general/strong of a statement. The authors have only qualitatively shown correlation of expression with the number of loops formed between the gene and potential enhancers and the correlation with the interaction frequencies was not assessed.

We agree, and have reworded this statement in the following way: “We find a strong correlation between expression of genes and the number of loops with distal putative enhancers these genes engage in”.

“Loops in the fourth and sixth clusters are of particular interest given that they are less enriched for RNA polymerase II, but are highly enriched for cohesin binding, which suggests that cohesin might be important in mediating enhancer-promoter loops”

“Further, most loop anchors, of any type, are associated with cohesin, suggesting a general role of this loop-extrusion complex in chromatin looping.”

“For instance, we find that cohesin is enriched at a large proportion of anchors of all types of loops, suggesting that cohesin, and possibly loop extrusion, is involved in their formation.”

The use of the words “suggests” and “suggesting” suggests that the role of cohesin in loop formation and loop extrusion is still unresolved. However, many papers have shown the role of cohesin in loop formation (e.g. Rao et al. 2017; Rhodes et al. 2020; Davidson et al 2019 and many more).

The reviewer is entirely correct that roles of cohesin in (some) loop formation is well established. Yet, we chose to be careful in our language because we have not shown for these sets of loop that their formation is indeed cohesin dependent. We have reworded such statements throughout the manuscript, while maintaining caution that in many cases our results are correlative.

“However, this is not directly demonstrated through perturbation experiments, and other mechanisms most likely will also play roles.”

Not in this manuscript perhaps, but RAD21-degrons have clearly shown the role of the cohesin complex in loop formation. See examples given above.

The reviewer is correct that Rad21 is involved in many structural loops detected by Hi-C (e.g., Rao et al. PMID: 28985562) . However, in more recent work from the 4DN nucleome consortium and others, we now know many loops can form, or be maintained, without cohesin, especially between loci enriched in H3K27Ac or Polycomb (e.g., work from the Blobel lab (PMID: 39975341), and Dekker lab PMID: PMID: 39345587). That is not to say that cohesin does not play additional, possibly guiding roles at favoring specific pairings of some combinations of elements as compared to others, as the Dekker lab has argued in PMID: 39345587.

“We found that the H1-hESC model is more sensitive to mutation of motifs for the embryonic stem cell factors POU2F1::SOX2”

This is a somewhat puzzling comment. The PWM in the figure does not look like the POU2F1::SOX2 motif in JASPAR. The most commonly identified motif in pluripotent cells is POU5F1::SOX2. Of the two citations one is not referring to Oct-1/Sox2 motif and the other does, but it is a rather obscure observation. Some more explanation towards the relevance of this observation would be useful.

To quantitatively assess whether the models are more sensitive to cell-type-specific motifs, we performed *in silico* mutations of approximately 500 binding sites for POU2F1::SOX2 and FOSL1::JUND, located at TAD boundaries unique to each cell type, by replacing the motifs with random sequences. We found that mutating the FOSL1::JUND motifs resulted in significantly higher disruption scores in the HFFc6 model compared to the H1-hESC model. Additionally, the distribution of disruption scores from the HFFc6 model exhibited a longer right tail than that of the H1-hESC model. In contrast, while the disruption scores were not significantly different between models when mutating the POU2F1::SOX2 motifs, the H1-hESC model still displayed a distribution with a longer tail compared to the HFFc6 model. The lack of significant differences in disruption scores may be attributed to the differences in sequencing depth of the Micro-C data used for model training.

Regarding the PWM in the figure, it is generated from the importance score of the motif at this genomic location. There is a JASPAR match to POU2F1::SOX2, and the motif we plotted shows what the model is attending to at that location. Although the importance values do not strongly resemble the canonical POU2F1::SOX2 motif from JASPAR, we do observe a general pattern consistent with the composite motif structure. This may reflect a context-specific version of the motif. The figure legend (now and Extended Data Figure 10) was modified to clarify this point.

Response to reviewers

Referee #1 (Remarks to the Author):

The authors have done a good job addressing my specific comments. Whether these revisions have sufficiently elevated the paper above the level of the initial submission remains an open question. To me personally, it still feels somewhat underwhelming for a consortium and a journal of this calibre. A key challenge may lie in presenting high-impact, novel insights that go significantly beyond those already reported in the previously published consortium papers, which are now helpfully listed in Supplemental Table 1.

What might work better for me in this context would be a stronger emphasis on synthesizing the results of those earlier papers, while incorporating new high-level analyses such as benchmarking, rather than striving to present this paper as a set of entirely novel findings. However, I defer to the editors on whether this approach aligns with their expectations.

Response: We thank the reviewer for their time to review this expansive work. We appreciate the additional comments from this reviewer. This is often an issue that is raised with this type of paper. In contrast to other consortia, we did not try to get the flagship and companion papers published all on the same day, as that would have unnecessarily delayed publication of dozens of papers. That would have hurt our young trainees and their progression through their PhD and post-doctoral studies. Therefore, we had the associated primary papers appear over the last years (see Supplementary Table 1). This does mean that the current flagship paper appears “late” and less novel. We do emphasize that there are important new resources described in the current flagship: we present the most comprehensive loop set to date for widely used cell lines. This was only possible by integrating data from a range of assays and has not been done in any companion paper. Second, we extensively benchmarked widely used assays that will guide future studies. Third, we have integrated different data modalities to derive the most comprehensive 3D models of the human genome, allowing the examination of the nuclear environment of all human genes, including cell-to-cell variation in that environment.

Specific comments:

1) I could not find a reference to Supplemental Table 1 in the main text.

Response: We have now added reference to Supplemental Table 1.

2) The penultimate sentence in the abstract is missing a verb and is likely truncated.

Response. The sentence contained a verb (“established”), but for clarity we replaced it with “showed”.

3) Supplemental Figure 12:

Panel (a): It’s a bit odd that global methods such as Hi-C and Micro-C are shown as equally suitable for detecting cCRE interactions as enrichment-based methods. After all, if this were the case, there would be no need to develop enrichment-based methods such as HiChIP in the first place. Unless the authors have strong opinions on that topic, I would suggest giving two stars to Hi-C and Micro-C for global cCRE detection rather than three.

Response. The reviewer is entirely correct. We have followed their advice and have given Hi-C and Micro-C two stars for global cCRE looping detection.

Panel (b): Capture-C and Capture Micro-C seem to be missing from the decision tree entirely. I’d also suggest adding some colour/framing to the diagrams so they are easier to use.

Response. Capture based methods were in fact included. We have now added the terms Capture-Hi-C and Capture Micro-C to make this more explicit.

4) Supplemental Figure 13: The figures and text are low-resolution and appear blurred. I also note that they are based on Capture Hi-C data, which is not referred to in the Supplemental Figure 12 at all.

Response: We checked the figure and believe it is of appropriate resolution. We can edit this once more if the editor feels this is needed.

As outlined in our response to the previous comment, capture methods were included in the decision tree. We have now added the terms Capture-Hi-C and Capture Micro-C to make this more explicit.

Referee #2 (Remarks to the Author):

The authors have addressed all my initial concerns. Thanks.

Response. We thank the reviewer for their time to review this expansive work.

Referee #3 (Remarks to the Author):

The reviewers have responded to my comments. They have made a new figure (Supplemental Figure 12) to guide the choice of method for 3/4D genome analysis, unfortunately this figure was not included in the submission, so this could not be assessed.

Response. We are sorry that this reviewer did not find Supplementary Figure 12, which in fact was included. The other reviewers did find this figure and had some suggestions to improve the figure. We have edited the figure accordingly.